# Diversity of Leaf Katydids of Odzala-Kokoua National Park, Republic of Congo, Central Tropical Africa (Insecta: Orthoptera: Tettigoniidae)

**DOI:** 10.3390/insects16030241

**Published:** 2025-02-25

**Authors:** Bruno Massa

**Affiliations:** IUCN SSC Orthoptera, Department of Agricultural, Food and Forest Sciences, University of Palermo, Viale delle Scienze, 13, 90128 Palermo, Italy; bruno.massa@people.unipa.it

**Keywords:** central and west Africa, forests, taxonomy, new species, list of species, species richness, conservation

## Abstract

Central tropical Africa is one of the richest areas in animal species in the world; regrettably, much of the primary African forest has been lost, and very probably many species became extinct before they had been discovered, mainly among insects. However, unknown taxa still live in these areas, demonstrating the present species richness and the urgence of conservation. This paper deals with the study of the Insecta Orthoptera Tettigoniidae found in Odzala-Kokoua National Park, Republic of Congo. This group of insects contains phytophagous, saprophagous, as well as predatory species, and occurs mainly within the vegetation characterizing tropical forests, being a good example of biological indicators. They were collected during three expeditions carried out by the African Natural History Research Trust (UK) and studied by the present author. The results show the high diversity of species that may be found in tropical Africa, mainly in the protected areas; a certain number of unknown species still lives in these forested areas, and a high endemism rate (16.0%) has been found in this wide Congolese area and conservation actions are requested for them.

## 1. Introduction

The Guinean Forests of west-central tropical Africa are critically endangered, and only few of the old primary forests survive. Main tropical Guinean forests lie in the countries of Liberia, Ivory Coast, Ghana, Togo, Benin, Nigeria, Equatorial Guinea, and São Tomé and Principe, and eastwards to the Sanaga River in Cameroon [1,2,3]. The forests of Gabon, the southern part of Central African Republic and Congo, show a certain faunistic continuity with the West African Guinean forests; see [3,4]. Of the original extent of Guinean forest of 620,314 km^2^, only 93,047 (15%) remain [5]. In many of these forests, some biodiversity hotspots have been recognized, mainly based on botanical aspects, but later also on zoological distinctive peculiarities, like the species richness and the number of endemic taxa. The Guinean Forest hotspots contain a rich and unique faunal assemblage, many species are endemic and tend to have highly restricted ranges within the forest. Biodiversity hotspots (covering just 16% of the Earth’s land) contain about 32% of all humans. Presently, the major threat is deforestation due to commercial logging and slash-and-burn agriculture, both of which are prevalent in all tropical African forest countries [6].

Much of the wide tropical African region is little explored from the entomological point of view, and we know that insects are the most abundant species living on the Earth. Thus, any information on the species richness and the endemistic occurrence of selected insect groups will certainly be used for conservation purposes.

The African Natural History Research Trust (ANHRT, Leominster, UK) in April, September, and November 2024 organized three research expeditions to Odzala-Kokoua National Park, Republic of Congo, in partnership with host institutions and government bodies. These expeditions followed previous ones in the Nouabalé-Ndoki National Park, Republic of Congo, also organized by the ANHRT (see Massa [4] for results on Tettigoniidae). The ANHRT sent on loan to the present author the three lots of specimens of Orthoptera Tettigoniidae collected during the entomological expeditions of 2024; this paper deals mainly with the study of this material.

## 2. Materials and Methods

### 2.1. Study Area (Figure 1)

The Odzala-Kokoua National Park (Republic of the Congo, 0.8° N, 14.9° E) was first established in 1935, and declared a Biosphere Reserve in 1977, with an official Presidential designation in 2001. It is 13,500 km^2^ wide and preserves old-growth rainforests and also savanna ecosystems. The park is presently managed by African Parks in partnership with the Congolese government, with the collaboration of the United States Fish and Wildlife Service, Wildlife Conservation Society, and World Wide Fund for Nature. Traditional problems in the park are poaching of elephants for the ivory market and mineral extraction. Odzala-Kokoua has approximately 100 mammal, 440 bird, and 4500 plant species, and one of Africa’s most diverse primate populations; 20,000 gorillas were estimated, 5000 of which were regrettably killed in 2005 by Ebola virus disease. The park hosts the highest densities of western lowland gorilla and chimpanzee presently recorded in central Africa. In addition, 9600 elephants were estimated in 2014, and a very high number of reptiles, fish, and insects, many of which endemic (source: https://en.wikipedia.org/wiki/Odzala-Kokoua_National_Park, accessed on 12 December 2024).

**Figure 1 insects-16-00241-f001:**
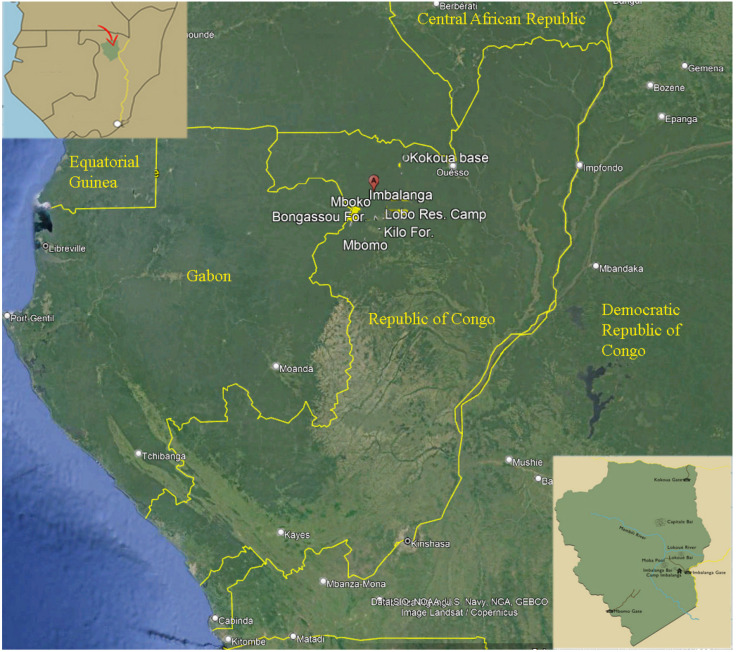
Central tropical Africa showing the borders of the Republic of Congo; in the inset above is the position of the Odzala-Kokoua National Park, in the inset below are the 13,500 km^2^ borders of Odzala-Kokoua National Park. Modified from Google Earth.

### 2.2. Study Sites

Imbalanga Camp (540 m), 00°45′47″ N, 15°15′39″ E (Figure 2)

Open-canopy Marantaceae forest; swamp forest; baï (swampy forest clearing). Imbalanga Camp is located in the northern-central region of Odzala-Kokoua National Park; it is characterized by an open-canopy rainforest (as part the Northwestern Congolian lowland rainforest ecoregion), with dense Marantaceae understorey vegetation covering the lowland rainforest floor. The camp is home to a baï, a distinct swampy forest clearing where various wildlife species gather, as well as a swamp forest along the streams.


**Figure 2 insects-16-00241-f002:**
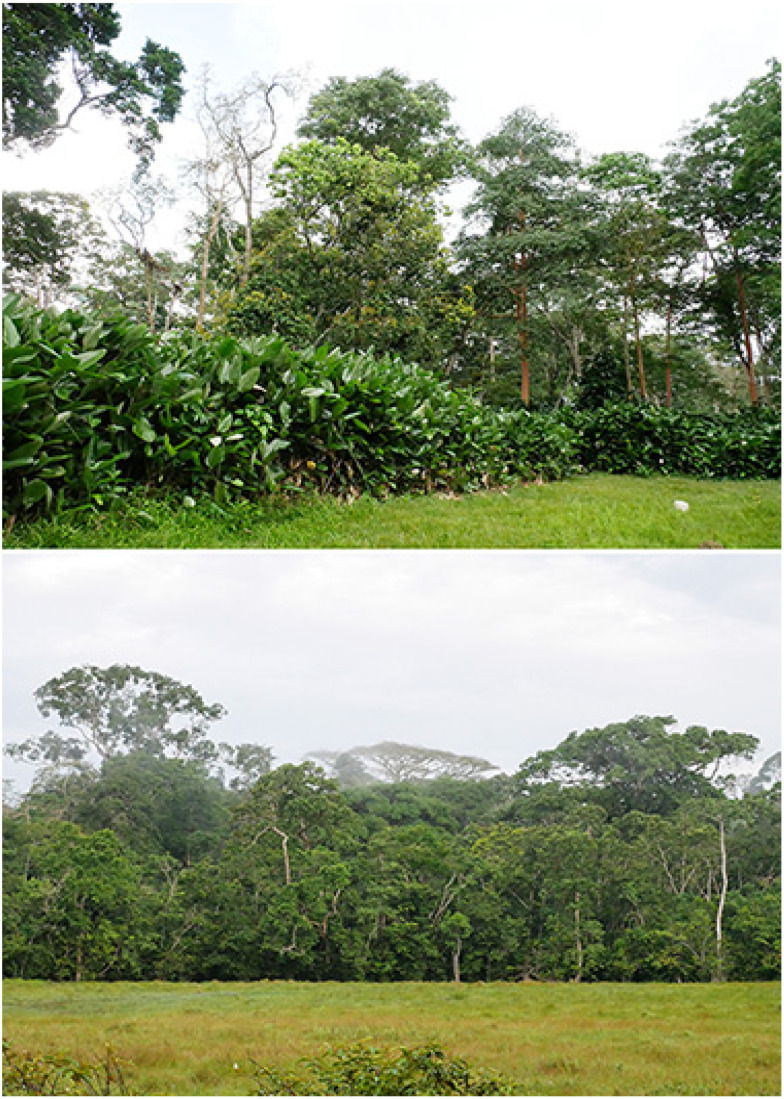
Imbalanga camp. (**Above**): the open-canopy Marantaceae forest; (**below**): baï, with swampy forest clearing. Photos: Molly Bashford.

Kokoua Base (540 m), 01°28′39″ N, 15°16′42″ E (Figure 3)

Open-canopy Marantaceae forest; stream/gallery forest. Kokoua Base is situated in the northern sector of Odzala-Kokoua National Park, sharing ecological similarities with Imbalanga Camp but at a higher elevation. The area is similarly covered by an open-canopy rainforest with widespread Marantaceae understorey; however, the canopy is more closed, and the forest consists of larger number of older trees than in Imbalanga with patches of Marantaceae-free and more diverse understorey vegetation. In close proximity to the base, there is a nearby stream and gallery forest with interrupted Marantaceae cover of the forest floor.

**Figure 3 insects-16-00241-f003:**
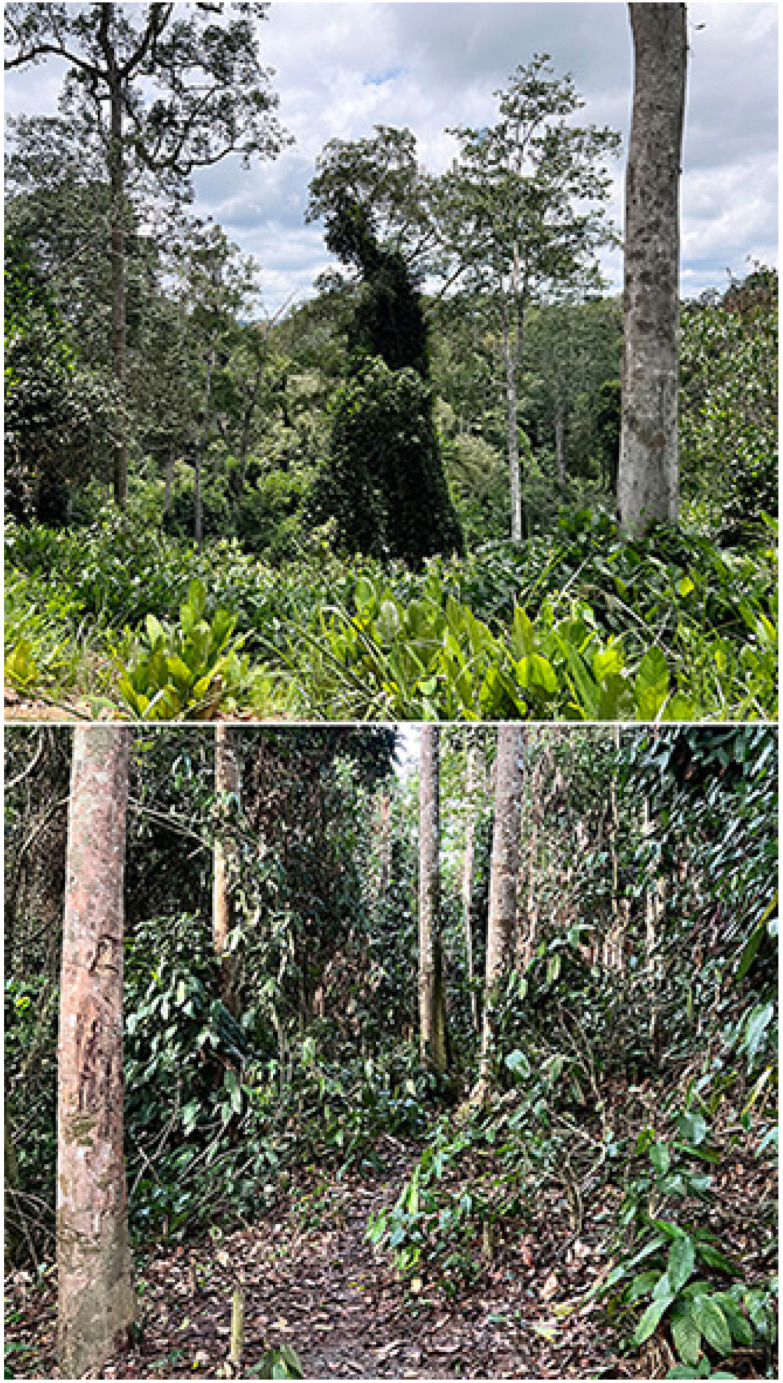
Kokoua base. (**Above**): open-canopy Marantaceae forest; (**below**): stream/gallery forest. Photos: Molly Bashford.

Moba (380 m), 00°48′52″ N, 15°05′15″ E (Figure 4)

Primary forest/closed canopy; riverine vegetation (gallery forest); baï (swampy or water-covered forest clearing). Moba is situated in proximity to the Mambili River. This area of the park primarily consists of closed-canopy rainforest with diverse vegetation, including minimal presence of Marantaceae species. Additionally, two large baïs can be found in the vicinity; one of them has a direct connection to the river, and the depth of water coverage is determined by the river’s water level, while the other baï is situated at a higher elevation suffused by narrow watercourses.

**Figure 4 insects-16-00241-f004:**
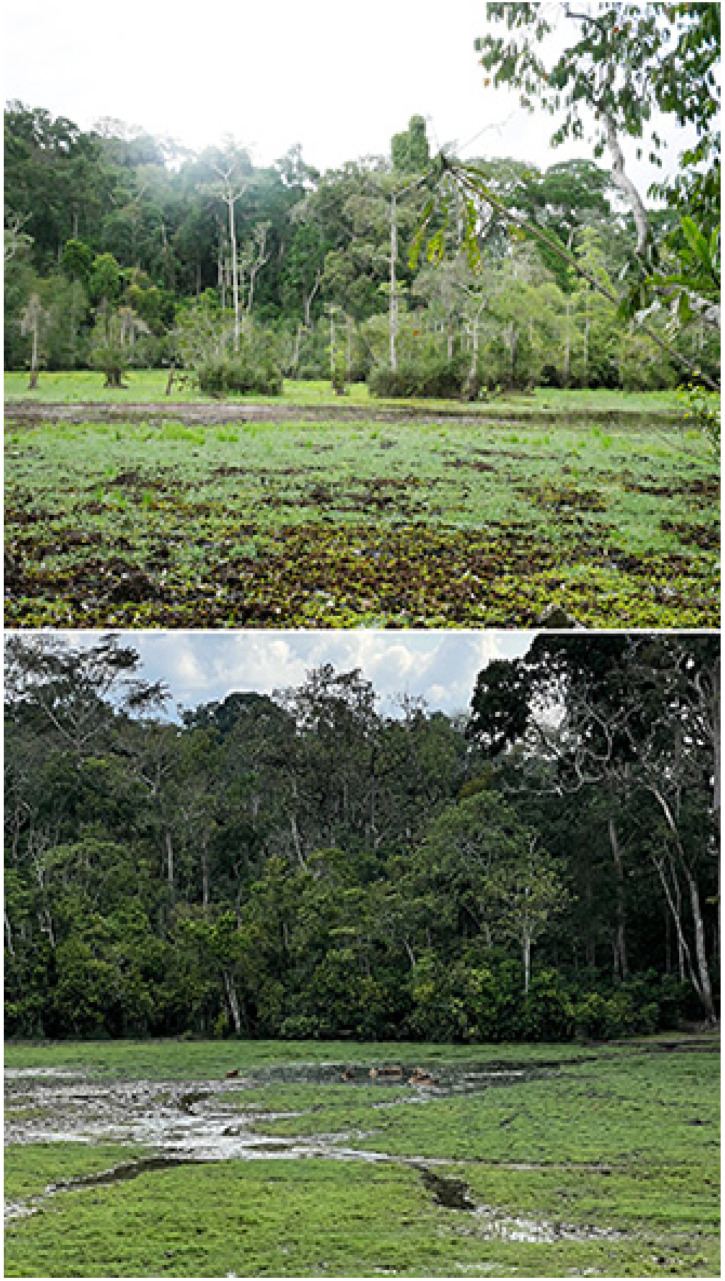
Moba. (**Above**): primary forest/closed canopy; (**below**): baï, with swampy or water-covered forest clearing. Photos: Molly Bashford.

Mboko (379 m), 00°37′04.79″ N, 14°24′27.32″ E (Figure 5)

Wet savanna; savanna-forest mosaics; riverine vegetation (gallery forest); primary forest (closed canopy). Located at the ecotone between wet grassland savanna and tropical lowland rainforest of Odzala-Kokoua National Park, the camp is characterized by savanna-forest mosaics. This unique landscape features a blend of hygrophilous grasslands, drier thickets, and closed-canopy rainforests without Marantaceae understorey vegetation. Nearby, gallery forests line the banks of smaller river systems, while patches of dense, humid primary forests with closed canopies add to the area’s ecological diversity.

**Figure 5 insects-16-00241-f005:**
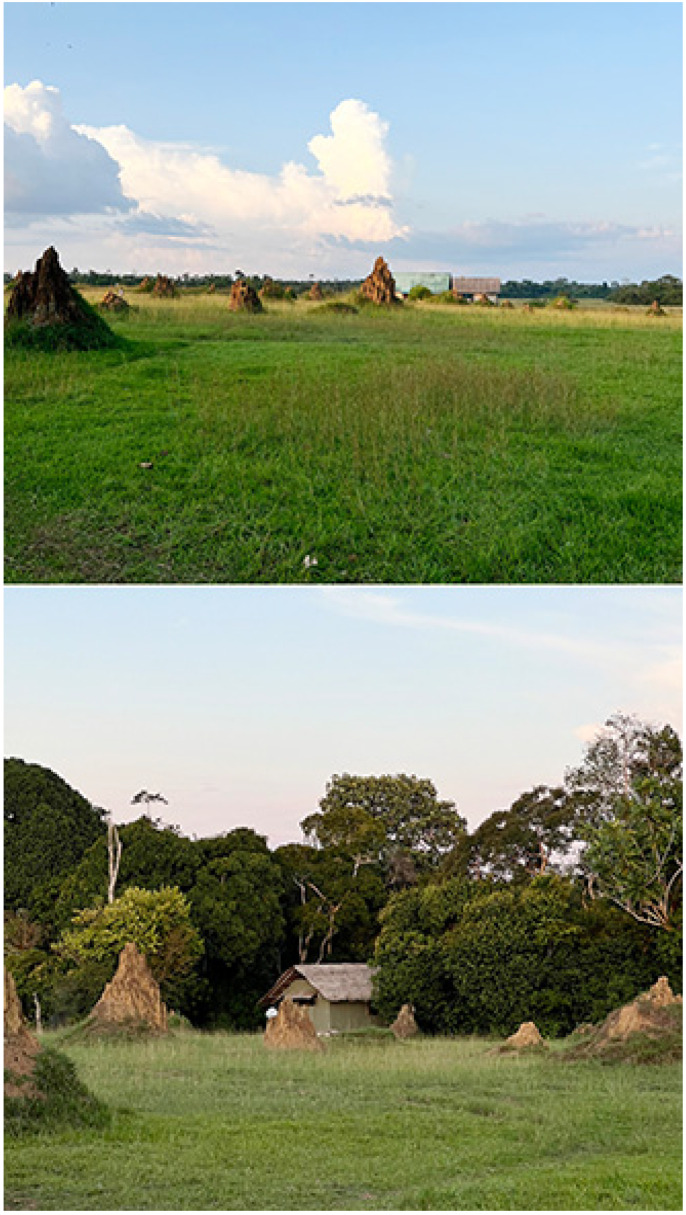
Mboko. (**Above**): wet savanna; (**below**): primary forest, closed canopy. Photos: Molly Bashford.

Lobo Research Camp (390 m), 00°35′04″ N, 14°53′12″ E (Figure 6)

Wet shrub savanna; Dambo (Dembe) (hygrophilous grassland with swampy patches); thicket; savanna-forest mosaics; gallery forest; stream. Located 10 km away from Mboko Tourist Camp, Lobo Research Camp is situated within the heart of Odzala-Kokoua National Park’s savanna ecosystem. The camp is enveloped by expansive wet shrub savanna and hygrophilous grassland (Dambo or Dembe). Within a short walking distance from the camp, a stream and gallery forest provide additional habitat variation. The savanna is characterized by its tall, sharp grasses and is home to various plant species. Kilo Forest (00°31′37″ N, 14°52′20″ E, 500 m) and Bongassou Forest near Lubo (00°32′50″ N, 14°51′47″ E, 400 m) are not many km from the Lobo Res. Camp.

**Figure 6 insects-16-00241-f006:**
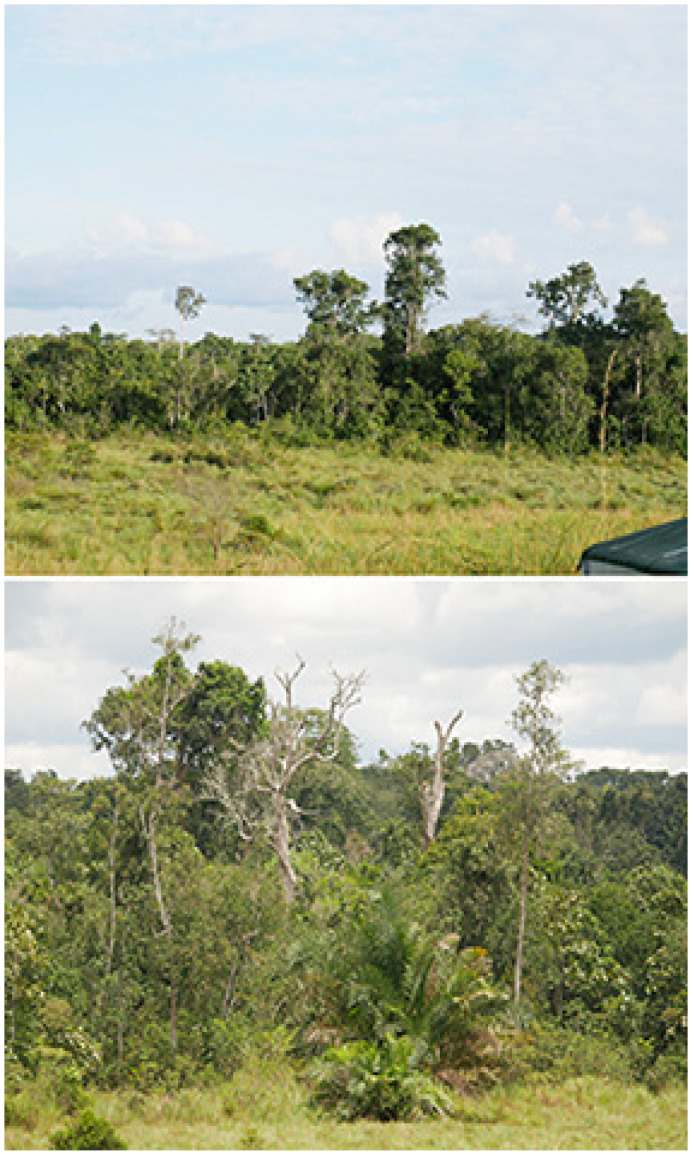
Lobo Research Camp. (**Above**): wet shrub savanna; (**below**): thicket. Photos: Molly Bashford.

Mbomo Headquarters (540 m), 00°26′13″ N, 14°42′01″ E (Figure 7)

Open-canopy Marantaceae forest; secondary/disturbed forest; village/farmland. Situated at the primary access point of Odzala-Kokoua National Park, Mbomo headquarters is surrounded by a mosaic of diverse habitats. The surrounding landscape includes the village of Mbomo and various agricultural lands. This region of the park showcases an open-canopy and consists of a disturbed secondary forest, where Marantaceae vegetation predominantly covers the rainforest floor.

**Figure 7 insects-16-00241-f007:**
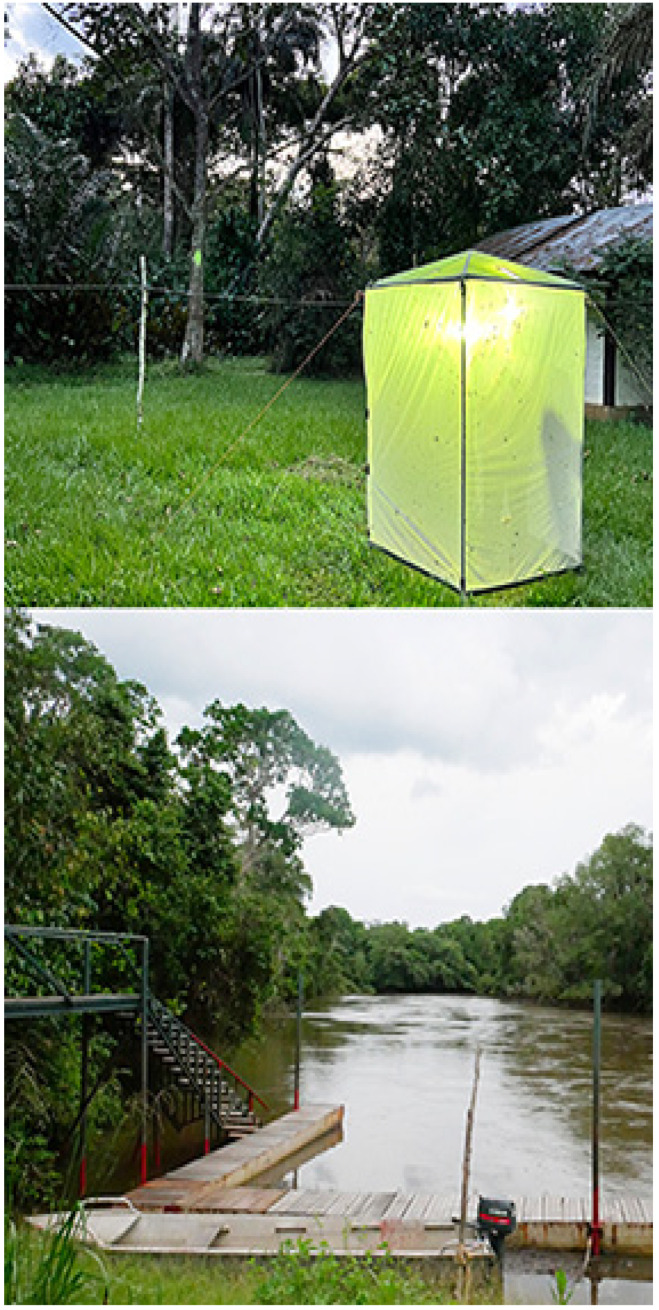
Mbomo Headquarters. (**Above**): the active MV light trap in open-canopy Marantaceae forest; (**below**): Moba-Mambili river. Photos: Molly Bashford.

### 2.3. Insect Sampling

Although the main focus of the ANHRT was Lepidoptera, other insect orders were sampled, including Coleoptera, Hemiptera, Diptera, Hymenoptera, Orthoptera, and Mantodea. Three main methods to collect Orthoptera were used. (a) Hand collecting: grasshoppers and katydids were collected using nets whenever they were spotted; (b) light trap: nocturnal insects were attracted to a specially engineered white tent containing a bright mercury vapour bulb where they could be collected. The ANHRT-designed Edwards’ Trap was placed under the bulb collecting the smallest insects which would otherwise be missed. As different insects are attracted to different wavelengths of light, smaller ultraviolet lights were also used, in combination with an automatic bucket trap; (c) a few insects were captured inside traps baited with carrion.

### 2.4. Study of Specimens

Selected specimens or their parts were photographed with a Nikon Coolpix 4500 digital camera (Tokyo, Japan), mounted on a Wild M3 Stereomicroscope and with an Olympus Stylus TG-5 Tough (Tokyo, Japan; cf. Mertens et al. [7]). Photographs were integrated using the freeware CombineZP [8]. Mounted specimens were measured with a digital calliper (precision 0.01 mm); the following measurements were taken (in mm): body length: dorsal length from the head to the apex of the abdomen; pronotum length and height; tegmina: length and depth of tegmina; hind femora length; ovipositor: maximum length, subgenital plate included. Comparisons of specimens have been carried out with private and public collections.

## 3. Results

The most fruitful method to collect Orthoptera was the light trap, using different kinds of lamps; it has been shown to be very attractive for most Orthoptera Ensifera species. Concerning some general results, below are reported the collecting days and collected specimens of Tettigoniidae in the three different expeditions: 5–22.IV.2024 [18 days, 392 specimens (284♂, 108♀), 21.8 specimens per day], 5.IX–1.X.2024 [27 days, 1371 specimens (986♂, 385♀), 46.2 specimens per day], 14–30.XI.2024 [17 days, 266 specimens (147♂, 119♀), 15.6 specimens per day].

All the specimens have been examined to observe the presence/absence of the spermatophore; indeed in bushcrickets, the male transfers an external spermatophore composed of a sperm-containing ampulla and a gelatinous, protein-rich spermatophylax (e.g., Eberhard and Lehmann [9]). The presence of the spermatophore at the abdomen apex of males indicates the mating season and is relevant to understand the phenology of these insects. Thus, in many cases, this information is carefully recorded.

The list of all Tettigoniidae collected is reported and commented on, excluding only the subfamily Meconematinae Burmeister, 1838.

Family Tettigoniidae Krauss, 1902Subfamily Pseudophyllinae Burmeister, 1838Tribe Pseudophyllini Burmeister, 1838*Liocentrum aduncum* Karsch, 1891Karsch, 1891, Berlin Ent. Z. 36(1): 88; type locality: Barombi Station (Cameroon) and Sierra Leone; depository: MfN, Berlin (syntypes).Material examined. Republic of Congo, Odzala-Kokoua NP, Kokoua base 5–13.IX.2024, MV light trap, M. Bashford, G. Lászlo, M. Talani, A. Volynkin (3♂, 2♀); Mbomo Headquarters 28.IX–1.X.2024, MV light trap, M. Bashford, G. László, A. Volynkin (1♂, 1♀) (ANHRT).Total. September: 4♂, 3♀.Distribution. Widespread in west-central tropical Africa [2].

*Opisthodicrus cochlearistylus* Karsch, 1891Karsch, 1891. Berlin Ent. Z. 36(1): 87; type locality: Barombi Station and Cinchoxo (Cameroon); depository: MfN, Berlin (♂, ♀ syntypes).Material examined. Republic of Congo, Odzala-Kokoua NP, Imbalanga Camp, 5–9.IV.2024, MV Light trap, M. Bashford, G. László, M. Talani, S. Yaba Ngouma (1♀); Kokoua base 5–13.IX.2024, MV light trap, M. Bashford, G. Lászlo, M. Talani, A. Volynkin (5♂, 19♀); Mbomo Headquarters 28.IX–1.X.2024, MV light trap, M. Bashford, G. László, A. Volynkin (2♂, 1♀); Lobo Res. Camp 22–30.XI.2024, MV light trap, M. Bashford, I. Elliott, A. Kirk-Spriggs (1♀) (ANHRT).Total. April: 1♀; September: 7♂, 20♀; November: 1♀.Distribution. Widespread in west-central tropical Africa.

*Oxyaspis congensis* Brunner von Wattenwyl, 1895Brunner von Wattenwyl, 1895. Verh. der Zoologisch-Botanischen Gesellsch. Wien 45: 34; type locality: Congo; depository: RBINS, Bruxelles (♀ holotype).Material examined. Republic of Congo, Odzala-Kokoua NP, Imbalanga Camp 5–9.IV.2024, MV Light trap, M. Bashford, G. László, M. Talani, S. Yaba Ngouma (1♀) (ANHRT).Total. April: 1♀.Distribution. Uncommon in west-central tropical Africa.

*Zabalius* aff. *albifasciatus* (Karsch, 1896)Karsch, 1896. Stett. Entomol. Z. 57: 347 (*Mataeus albifasciatus*); type locality: lolodorf (Cameroon); depository: MfN, Berlin (♂, ♀ syntypes).Material examined. Republic of Congo, Odzala-Kokoua NP, Imbalanga Camp 5–9.IV.2024, MV Light trap, M. Bashford, G. László, M. Talani, S. Yaba Ngouma (1♂) (ANHRT).Remarks. This specimen is similar to *Z. albifasciatus*, but it lacks the whitish band between the eyes, also lacking in the specimen collected in the Nouabalé-Ndoki National Park (Republic of Congo) [4]; thus, it has been tentatively identified.Total. April: 1♂.Distribution. *Z. albifasciatus* is presently known only from its type material, collected in Cameroon, and from the specimens collected in the Sangha Reserve (Central African Republic) and the Nouabalé-Ndoki National Park (Republic of Congo).

*Zabalius lineolatus* (Stål, 1873)Stål, 1873, Ofv. K. Vetensk. Akad. Forh. 30(4): 48; type locality: Cameroon (?); depository: NHRS, Stockholm (♀ holotype).Material examined. Republic of Congo, Odzala-Kokoua NP, Lobo Res. Camp 20–27.IX.2024, MV light trap, M. Bashford, G. László, A. Volynkin (1♂); Mbomo Headquarters 28.IX–1.X.2024, MV light trap, M. Bashford, G. László, A. Volynkin (1♂); Imbalanga camp 14–18.XI.2024, MV light trap, M. Bashford, I. Elliott, A. Kirk-Spriggs (1♂, 1♀); Lobo Res. Camp 22–23.XI.2024, MV light trap, M. Bashford, I. Elliott, A. Kirk-Spriggs (3♂) (ANHRT).Total. September: 2♂; November: 4♂, 1♀.Distribution. Widespread in west-central tropical Africa.

Tribe Pleminiini Brunner von Wattenwyl, 1895*Lichenochrus crassipes* Karsch, 1890Karsch, 1890, Entom. Nachricht. 16: 269; type locality: Barombi Station (Cameroon); depository: MfN, Berlin (♀ holotype, missing).Material examined. Republic of Congo, Odzala-Kokoua NP, Mbomo Headquarters 28.IX–1.X.2024, MV light trap, M. Bashford, G. László, A. Volynkin (1♂) (ANHRT).Total. September: 1♂.Distribution. Known from Cameroon and Guinea [2].

*Adapantus* (*Adapantus*) *longipennis* Beier, 1954Beier, 1954, Revision der Pseudophyllinen 356; type locality: Kribi (Cameroon); depository: MZPW, Warsaw (♂ holotype).Material examined. Republic of Congo, Odzala-Kokoua NP, Kokoua base 5–13.IX.2024, MV light trap, M. Bashford, G. Lászlo, M. Talani, A. Volynkin (1♂) (ANHRT).Total. September: 1♂.Distribution. Previously known from Cameroon and Republic of Congo [2].

*Habrocomes piotri* n. sp. (Figure 8)urn:lsid:zoobank.org:act:8937EB2C-1EC4-4528-92C9-EEC9A37BB62A

Material examined. Republic of Congo, Odzala-Kokoua NP, Mbomo Headquarters (540 m) 00°26′13″ N, 14°42′01″ E, 28.IX–1.X.2024, MV light trap, M. Bashford, G. László, A. Volynkin (♀ holotype) (ANHRT).

Description. Female (Figure 8A,B). *Colour*. Small, brownish with cream spots and ivory base of ovipositor; antennae with black and white rings. *Head*. Forehead not or only slightly transverse, mostly dark. Fastigium verticis triangular, concave, triangular, weakly furrowed, just protruding the edges of the antenna pits. Antennae very long, the scapus with terminal spine. *Thorax*. Pronotum only very indistinctly granulated, flat, the anterior margin protruding and merging almost straight into the lateral lobes, with a median tubercle; the transverse grooves rather shallow, the posterior one clearly located behind the centre and hardly angled; lateral lobes about twice as long as they are high, the posterior margin narrowly groin-shaped with two lateral denticles, right-angled anterior and only slightly obtuse-angled posterior corners (Figure 8C,D). Meso- and metasternum transverse, flat, the latter rather narrowed posteriorly, its pits closer together and short, curved, anteriorly concave. Prosternum with two spines. *Legs*. Long and strong, short ciliated. Fore femora slightly compressed, with four spines on the ventral inner edge. Fore tibiae with ventral double row of seven spines, dorsally unarmed. Dorsal margin of mid femora unarmed, ventral outer margin with three robust spines. Middle tibiae broadened in the basal half, on the dorsal inner edge with a small, inconspicuous spine in front of the centre, and a sub-basal spine similar to a small denticle. Ventral margin with a double row of four spines. Hind femora with 6 stout spines on outer ventral margin (Figure 8E), hind tibiae with 12 big flat spines on inner dorsal margina, 9 smaller spines on outer margin; ventral margin with double row of 9 spines. Genicular lobes with long and pointed, wide, protruding terminal spines, only the outer lobes of the middle femora rounded. *Tegmina*. Very short, reaching the first abdominal tergite, alae hidden below and shorter than tegmina (Figure 8C,D). *Abdomen*. Dorsal posterior margins of tergites with a small swelling. Subgenital plate triangular, terminal just incised (Figure 8E). Ovipositor strong, rather broad, with apical folds, the dorsal margin with a very shallow and indistinct median kink, straight and crenulated behind it.

Measurements (mm). Body length: 29.0; length of pronotum: 6.2; height of pronotum: 2.5; length of tegmina: 5.0; length of fore femora: 10.0; width of fore femora: 1.3; length of mid femora: 8.0; length of hind femora: 19.0; ovipositor: 15.6.

Male. Unknown

Diagnosis. Small species, unique for its tegmina reduced to two scales. Compared to other species of the genus [10], the pronotum is not barely saddled, the legs are less densely hairy than other species of the genus, and the tegmina are very reduced.

Etymology. This species is dedicated with high esteem to Piotr Naskrecki, an authority on Pseudophyllinae from tropical Africa; he is credited with identifying the genus of the present Pleminiini.

Remarks. Within the tribe Pleminiini, there are no species with such small wings; in other known species within the tribe, at most, the wings reach the length of the apex of the hind femora. The systematics of the group of Pleminiini with a flat pronotum, not saddled, is largely based on the presence or absence of small dorsal spines on the mid-tibia. The genus *Habrocomes* Karsch, 1890 is characterized by the presence of only a small, inconspicuous dorsal spine on the mid-tibia, and the inner margin of hind tibiae has conspicuous big spines. This species is a case of an extreme reduction in tegmina and evidently can only move by walking through forest vegetation. The known species in the genus [*Habrocomes lanosus* Karsch, 1891, *H. marmoratus* (Bólivar, 1906) and *H. personatus* (Sjöstedt, 1902)] have developed wings, more or less as long as the abdomen [11].

**Figure 8 insects-16-00241-f008:**
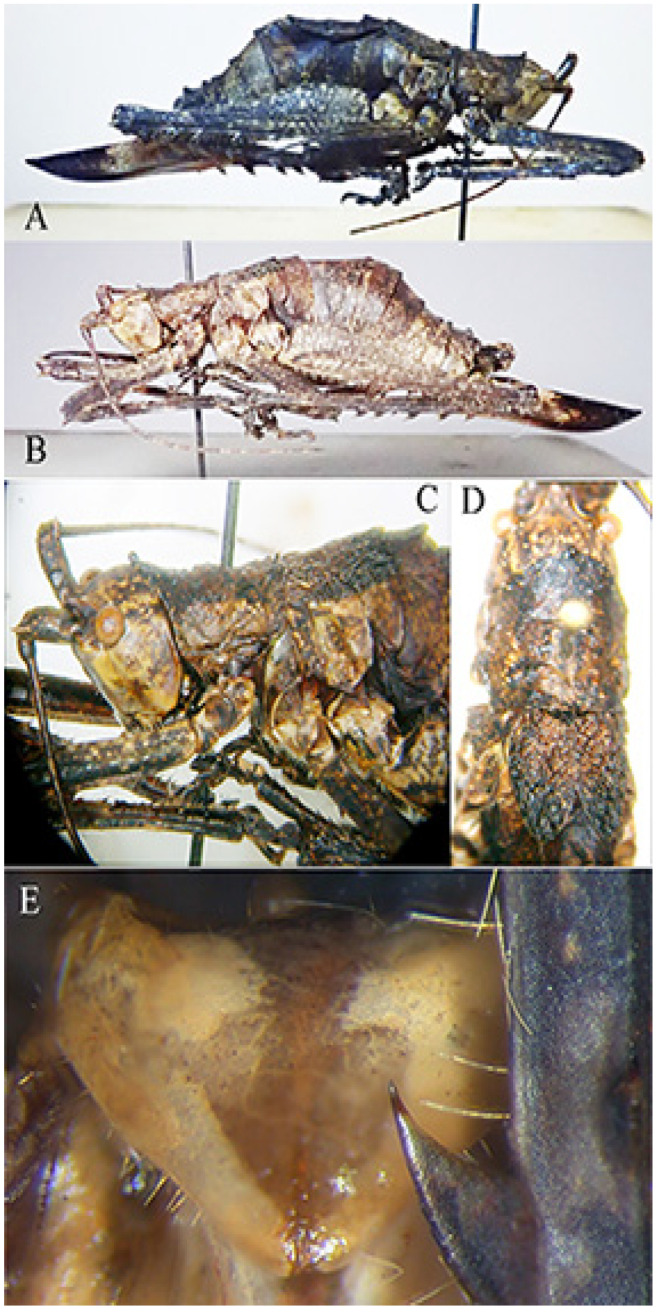
*Habrocomes piotri* n. sp. (**A**): right lateral view; (**B**): left lateral view; (**C**): left lateral view of head, pronotum, and metanotum; (**D**): dorsal view of head, pronotum, and tegmina; (**E**): female subgenital plate and detail of one spine of hind tibiae.

*Tympanocompus acclivis* Karsch, 1891Karsch, 1891, Berlin Ent. Z. 36(1): 108; type locality: Barombi Station (Cameroon); depository: MfN, Berlin (♂, ♀ syntypes).Material examined. Republic of Congo, Odzala-Kokoua NP, Camp Imbalanga 14–18.XI.2024, MV light trap, M. Bashford, I. Elliott, A. Kirk-Spriggs (1♂) (ANHRT).Distribution. Widespread in west-central tropical Africa.

Tribe Cymatomerini Brunner von Wattenwyl, 1895*Cymatomera argillata* Karsch, 1891Karsch, 1891, Berlin Ent. Z. 36(1): 98; type locality: Congo; depository: MZPW, Warsaw (♀ holotype).Material examined. Republic of Congo, Odzala-Kokoua NP, Lobo Res. Camp 13–18.IV.2024, MV Light trap, M. Bashford, G. László, M. Talani, S. Yaba Ngouma (1♂); Mbomo Headquarters 28.IX–1.X.2024, MV light trap, M. Bashford, G. László, A. Volynkin (1♂, 1♀) (ANHRT).Distribution. Widespread in west-central tropical Africa.

Tribe Phyllomimini Brunner von Wattenwyl, 1895*Tomias* (*Tomias*) *stenopterus* Karsch, 1891Karsch, 1891, Berlin Ent. Z. 36(1): 90; type locality: Cameroon; depository: MZPW, Warsaw (♀ holotype).Material examined. Republic of Congo, Odzala-Kokoua NP, Kokoua base 5–13.IX.2024, MV light trap, M. Bashford, G. Lászlo, M. Talani, A. Volynkin (1♂) (ANHRT).Distribution. Reported from west-central tropical Africa [2].

*Stenampyx annulicornis* Karsch, 1891Karsch, 1891, Berlin Ent. Z. 36(1): 93; type locality: Barombi Station (Cameroon); depository: MfN (♀ syntypus).Material examined. Republic of Congo, Odzala-Kokoua NP, Imbalanga Camp 5–9.IV.2024, MV Light trap, M. Bashford, G. László, M. Talani, S. Yaba Ngouma (4♂, 4♀); Kokoua base 5–13.IX.2024, MV light trap, M. Bashford, G. Lászlo, M. Talani, A. Volynkin (2♂); Mbomo Headquarters 28.IX–1.X.2024, MV light trap, M. Bashford, G. László, A. Volynkin (6♂, 8♀) (ANHRT).Total. April: 4♂, 4♀; September: 8♂, 8♀.Distribution. Widespread in west-central tropical Africa.

Subfamily Conocephalinae Burmeister, 1838Tribe Conocephalini Burmeister, 1838*Conocephalus* (*Conocephalus*) *conocephalus* (Linnaeus, 1767)Linnaeus, 1767, Systema Naturae 1, pt. 2: 696 [*Gryllus* (*Tettigonia*) *conocephalus*]; type locality: Africa; type lost.Material examined. Republic of Congo, Odzala-Kokoua NP, Lobo Res. Camp 16–17.IV.2024, Lepiled light trap, M. Bashford, G. László, M. Talani, S. Yaba Ngouma (1♂); Camp Imbalanga 14–18.XI.2024, MV light trap, M. Bashford, I. Elliott, A. Kirk-Spriggs (3♀); Imbalanga bai 15–17.XI.2024, Lepiled light trap M. Bashford, I. Elliott, A. Kirk-Spriggs (1♀); Mboko 18–21.XI.2024 MV light trap, M. Bashford, I. Elliott, A. Kirk-Spriggs (5♂, 7♀); Lobo Res. Camp 22–23.XI.2024, MV light trap, M. Bashford, I. Elliott, A. Kirk-Spriggs (2♂, 5♀); Lobo Res. Camp 23–27.XI.2024, Actinic light trap M. Bashford, I. Elliott, A. Kirk-Spriggs (1♀); Lobo Res. Camp 23–25.XI.2024, Lepiled light trap, M. Bashford, I. Elliott, A. Kirk-Spriggs (1♂, 2♀) (ANHRT).Total. April: 1♂; November: 8♂, 19♀.Distribution. Widespread in Africa and south Europe.

*Conocephalus* (*Anisoptera*) *mollyae* n. sp. (Figure 9 and Figure 10)urn:lsid:zoobank.org:act:DE453BE8-7679-4D90-9FCA-AB4F0E0BFF3D

Material examined. Republic of Congo, Odzala-Kokoua NP, Imbalanga Camp, 00°45′47″ N, 15°15′39″ E, 5–11.IV.2024, Actinic Light trap, M. Bashford, G. László, M. Talani, S. Yaba Ngouma (♂ holotype in ANHRT).

Description. Male (Figure 9A). *Colour*. Light brown with two black lateral bands on the head, pronotum, and abdomen; two black spots on fore and hind dorsal margin of pronotum. Abdominal tergites yellowish with a central dark stripe, antennae of the same colour as the body. Tegmina brownish, styli black. *Head*. Fastigium compressed, antennae exceeding the apex of femora, eyes round and prominently projecting. *Thorax*. Pronotum short, longer than tegmina, with anterior and posterior margins straight, lower lateral margins rounded, a small incision on the posterior margin (Figure 9B). Prosternum with two thin spines. *Legs*. Fore coxae armed with an evident spine. Fore tibiae with five spines on each lower margin; middle tibiae and hind tibiae with 2–3 apical spines on each lower margin. Hind tibiae with four big spurs. Femora unarmed. *Tegmina*. Exceeding the posterior margin of the second tergite (Figure 9A,B). *Abdomen*. Supragenital plate with a central concavity; cerci slender, long, straight with narrow, sinuous apical tips and two inner thin spines, apically pointed (Figure 9C), similar to those of *Thyridorhoptrum carbonarium* (Redtenbacher, 1891). Subgenital plate narrow and concave, styli pointed.

Female. Unknown.

Measurements (mm). Body length: 14.3; length of pronotum: 3.2; height of pronotum: 2.3; length of tegmina: 2.8; length of hind femora: 14.3.

Diagnosis. *Conocephalus mollyae* n. sp. is characterized by two small spines on the sternum; thus, it belongs to the subgenus *Anisoptera* Latreille, 1829; also, the subgenus *Chloroxiphidion* Hebard, 1922 has two spines on the sternum, but its dorsal pair of spurs are greatly reduced, not larger than the spines of the dorsal margins of the caudal tibiae and only differentiated from them by having a basal socket [12]. It is a small micropterous species, easily distinguished by the two slender inner spines on the thin and long cerci.

Distribution. Presently known only from the Imbalanga Camp, Odzala-Kokoua National Park (Republic of Congo).

Etymology. *C. mollyae* n. sp. is named after Molly Bashford, Administrator of the Museum of the African Natural History Research Trust, and leader of 2024 expeditions to the Odzala-Kokoua National Park in the Republic of Congo.

Affinities. Among the micropterous taxa belonging to the subgenus *Anisoptera,* the following species have been described from tropical Africa: *Conocephalus maputensis* Massa, 2022 (Mozambique) (tegmina reaching the apical margin of the first abdominal tergite, cerci robust, short, curved inwards, apically blunt; a first small pair of teeth in the inner side before the middle, and a second much stouter and longer pair of teeth before the apex of the cerci) (Figure 9D,E); *C. bechuanensis* (Péringuey, 1916) (Botswana, Bechuanaland, Southern Africa) (holotype male, not female, as reported by Péringuey [13], and pointed out by Uvarov [14], who depicted the cerci) (cerci with only one inner pre-apical spine, tegmina reaching only the apical margin of the first abdominal tergite) (Figure 9F); *C. caudalis* (Walker, 1869) (KwaZulu-Natal, Southern Africa), whose male was described by Redtenbacher [15] as *Xiphidium natalense* (later synonymized with *C. caudalis*) (inner spines of cerci similar, but the apex of the cerci much longer than in *C. mollyae* n. sp.) (Figure 9F); *C. peringueyi* Uvarov, 1928 (Cape Province) (pronotum profile different, cerci with an apical incurved spine and 2–3 teeth at its base) (Figure 9F); *C. rhodesianus* (Péringuey, 1916) (South Tropical Africa, Zimbabwe) (tegmina reaching the base of the fifth abdominal tergite, cerci with only one inner stout spine) (Figure 10A,B); *C. meadowsae* Harz, 1970 (Togo) (tegmina reaching the third abdominal tergite, cerci with only one inner stout spine) (Figure 10C,D). *C. iris* (Serville, 1838) (from Madagascar to most of central-southern tropical Africa), which has brachypterous as well as holopterous forms, has been excluded, with its last tergite very notable and protruding [16].

**Figure 9 insects-16-00241-f009:**
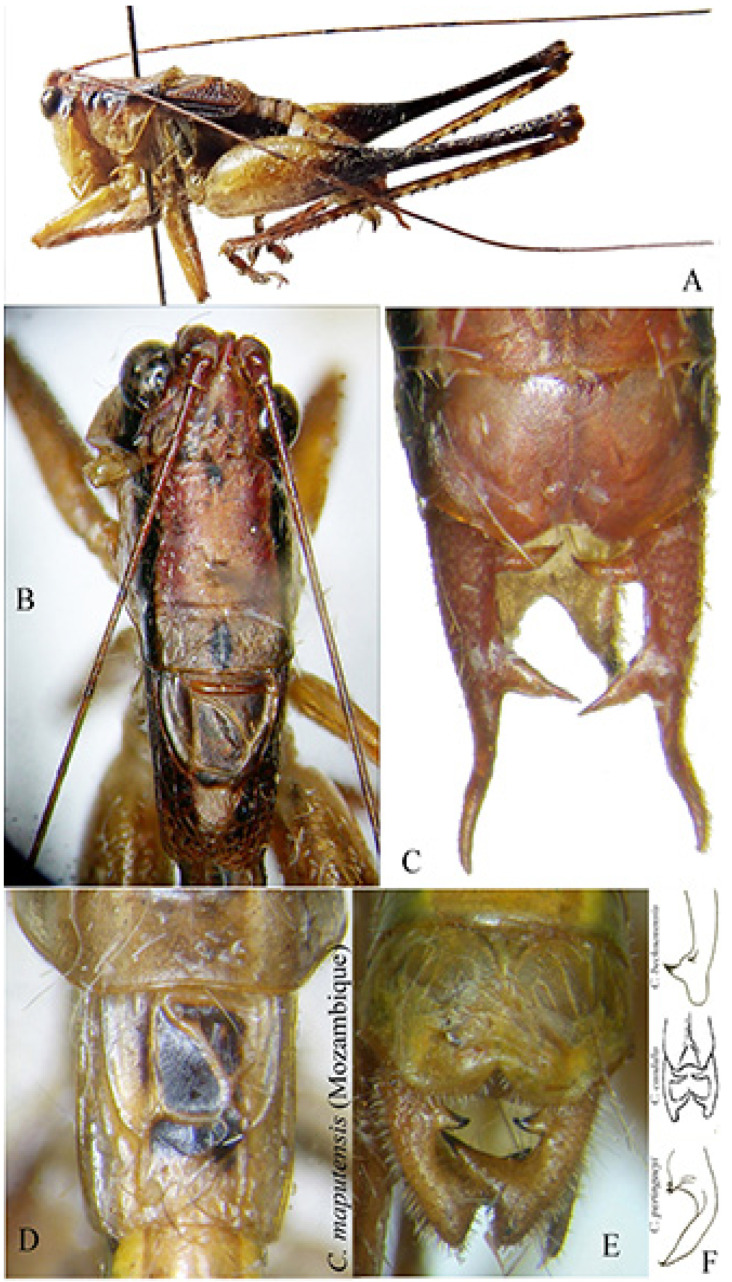
*Conocephalus* (*Anisoptera*) *mollyae* n. sp. (**A**): habitus of male; (**B**): dorsal view of head, pronotum, and tegmina; (**C**): abdomen apex with cerci and subgenital plate; (**D**,**E**): dorsal view of tegmina and cerci of *C.* (*Anisoptera*) *maputensis* from Mozambique; (**F**): cerci of *C. bechuanensis* (above), *C. caudalis* (middle), and *C. peringueyi* (after Uvarov [14]).

**Figure 10 insects-16-00241-f010:**
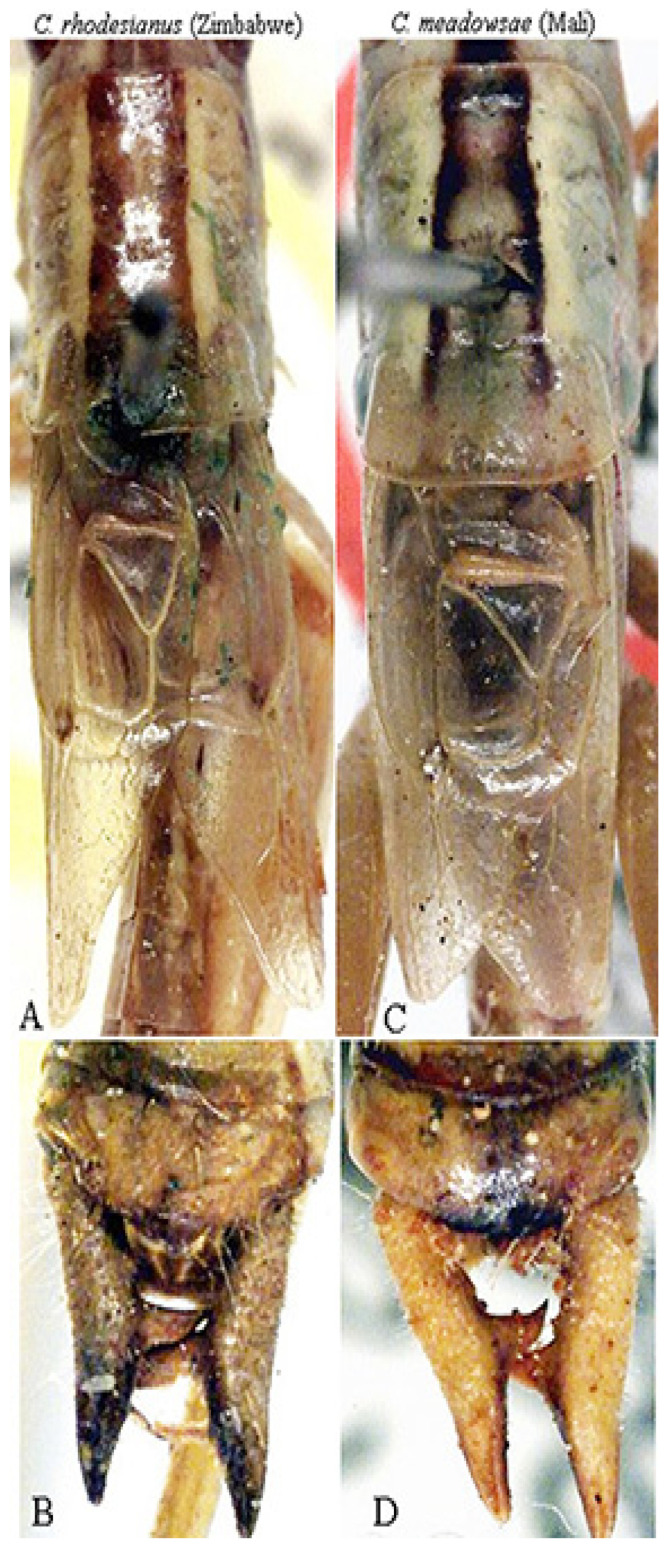
(**A**,**B**): dorsal view of pronotum and tegmina (**A**) and cerci (**B**) of male syntype *C. rhodesianus* from Zimbawe; (**C**,**D**): dorsal view of pronotum and tegmina (**C**) and cerci (**D**) of male holotype *C. meadowsae* from Mali, both preserved at NHM, London.

*Conocephalus* (*Anisoptera*) *maculatus* (Le Guillou, 1841)Le Guillou, 1841, Revue et Magasin de Zoologie 294; type locality: West Malaysia; depository: NHM, London (♂ neotype).

Material examined. Republic of Congo, Odzala-Kokoua NP, Kokoua base 5–13.IX.2024, MV light trap, M. Bashford, G. Lászlo, M. Talani, A. Volynkin (2♀); Lobo Res. Camp 20–27.IX.2024, MV light trap, M. Bashford, G. László, A. Volynkin (2♂, 5♀); Lobo Res. Camp 21–22.IX.2024, Actinic light trap, M. Bashford, G. László, M. Talani, A. Volynkin (1♀); Lobo Res. Camp 23–26.XI.2024, Malaise trap M. Bashford, I. Elliott, A. Kirk-Spriggs (1♀); Lobo Res. Camp 22–23.XI.2024, MV light trap, M. Bashford, I. Elliott, A. Kirk-Spriggs (1♀); Lobo Res. Camp 23–25.XI.2024, Lepiled light trap, M. Bashford, I. Elliott, A. Kirk-Spriggs (1♂) (ANHRT).

Total. September: 2♂, 8♀; November: 1♂, 2♀.Distribution. Widespread in Africa and Asia.

*Ruspolia* spp.

Material examined. Republic of Congo, Odzala-Kokoua NP, Imbalanga Camp, 5–9.IV.2024, MV Light trap, M. Bashford, G. László, M. Talani, S. Yaba Ngouma (2♂, 4♀); Mbomo Headquarters, 19–22.IV.2024, MV Light trap, M. Bashford, G. László, M. Talani, S. Yaba Ngouma (1♂, 1♀); Mbomo Headquarters 5–11.IV.2024, Actinic Light trap, M. Bashford, G. László, M. Talani, S. Yaba Ngouma (2♂); Lobo Res. Camp 12–19.IV.2024, Actinic Light trap, M. Bashford, G. László, M. Talani, S. Yaba Ngouma (1♀); Lobo Res. Camp 13–18.IV.2024, MV Light trap, M. Bashford, G. László, M. Talani, S. Yaba Ngouma (1♀); Kokoua base 5–13.IX.2024, MV light trap, M. Bashford, G. Lászlo, M. Talani, A. Volynkin (2♂, 1♀); Kokoua base 11–13.IX.2024, Lepiled light trap, M. Bashford, G. Lászlo, M. Talani, A. Volynkin, S. Yaba Ngouma (1♂); Moba 17.IX.2024, General collection, M. Bashford, G. Lászlo, M. Talani, A. Volynkin, S. Yaba Ngouma (1♂); Lobo Res. Camp 20–27.IX.2024, MV light trap, M. Bashford, G. László, A. Volynkin (3♂, 5♀); Lobo Res. Camp 21–22.IX.2024, Actinic light trap, M. Bashford, G. László, M. Talani, A. Volynkin (1♂); Lobo Res. Camp 20–27.IX.2024, MV light trap, M. Bashford, G. László, A. Volynkin (4♂, 2♀); Lango bai (373 m) 00°36′03.83″ N, 14°56′02.21″ E, 24–25.IX.2024, Actinic Light trap, M. Bashford, G. László, A. Volynkin, S. Yaba Ngouma (1♂); Mbomo Headquarters 28.IX–1.X.2024, MV light trap, M. Bashford, G. László, A. Volynkin (1♂, 2♀); Camp Imbalanga 14–18.XI.2024, MV light trap, M. Bashford, I. Elliott, A. Kirk-Spriggs (1♂, 7♀); Mboko 18–21.XI.2024 MV light trap, M. Bashford, I. Elliott, A. Kirk-Spriggs (4♂, 6♀); Lobo Res. Camp 22–30.XI.2024, MV light trap, M. Bashford, I. Elliott, A. Kirk-Spriggs (4♂, 18♀); Bengassou near Lobo 24–28.XI.2024, Malaise trap, M. Bashford, I. Elliott, A. Kirk-Spriggs (2♀); Lekoli river, near Mboko 23–24.XI.2024, Lepiled light trap M. Bashford, I. Elliott, A. Kirk-Spriggs (1♂); Bongassou Forest near Lobo 26–28.XI.2024, Lepiled light trap M. Bashford, I. Elliott, A. Kirk-Spriggs (1♂, 1♀); Lobo Res. Camp 23–26.XI.2024, Malaise trap M. Bashford, I. Elliott, A. Kirk-Spriggs (1♂, 1♂ nymph, 2♀); Lobo Res. Camp 23–25.XI.2024, Lepiled light trap, M. Bashford, I. Elliott, A. Kirk-Spriggs (1♂, 1♀); Lobo Res. Camp 23–27.XI.2024, Actinic light trap M. Bashford, I. Elliott, A. Kirk-Spriggs (1♂, 1♀) (ANHRT).

Total. April: 5♂, 7♀; September: 14♂, 10♀; November: 14♂, 1♂ nymph, 38♀.

Remarks. Even if the genus has been revised by Bailey [17], some difficulties remain for the female and sometimes also for the male identification; thus, I prefer to leave all the specimens as unidentified.

Tribe Copiphorini Karny, 1912*Plastocorypha nigrifrons* (Redtenbacher, 1891)Redtenbacher, 1891, Verh. der Zoologisch-Botanischen Gesellsch. Wien 41:,368; type locality: Cameroon; depository: MZPW, Warsaw (♀ holotype).

Material examined. Republic of Congo; Odzala-Kokoua NP; Mbomo Headquarters 28.IX–1.X.2024; MV light trap; M. Bashford; G. László; A. Volynkin (1♀) (ANHRT)

Total. September: 1♀.Distribution. Uncommon in west-central tropical Africa.

Subfamily Phaneropterinae Burmeister, 1838Tribe Phaneropterini Burmeister, 1838*Phaneroptera sparsa* Stål, 1857Stål, 1857[1856]. Ofv. K. Vetensk. Akad. Forh. 13: 170; type locality: South Africa; depository: NHRS (♀ holotype).

Material examined. Republic of Congo, Odzala-Kokoua NP, Imbalanga Camp 5–9.IV.2024, MV Light trap, M. Bashford, G. László, M. Talani, S. Yaba Ngouma (1♂); Moba, 11–12.IV.2024, Actinic Light trap, M. Bashford, G. László, M. Talani, S. Yaba Ngouma (1♂); Mbomo Headquarters 19–22.IV.2024, MV Light trap, M. Bashford, G. László, M. Talani, S. Yaba Ngouma (2♂, 1♀); Kokoua base 5–13.IX.2024, MV light trap, M. Bashford, G. László, M. Talani, A. Volynkin (3♂, 2♀); Kokoua base 11–13.IX.2024, Lepiled light trap, M. Bashford, G. Lászlo, M. Talani, A. Volynkin, S. Yaba Ngouma (3♀); Lobo Res. Camp 20–27.IX.2024, MV light trap, M. Bashford, G. László, A. Volynkin (1♂, 2♀); Lobo Res. Camp 21–22.IX.2024, Actinic light trap, M. Bashford, G. László, M. Talani, A. Volynkin (1♂); Lobo Res. Camp 25–26.IX.2024, Lepiled light trap, M. Bashford, G. László, A. Volynkin, S. Yaba Ngouma (2♂, 1♀); Mbomo Headquarters 28.IX–1.X.2024, MV light trap, M. Bashford, G. László, A. Volynkin (1♀); Mbomo Headquarters 28.IX–1.X.2024, Lepiled light trap, M. Bashford, G. László, M. Talani, A. Volynkin, S. Yaba Ngouma (1♂); Lobo Res. Camp 23–26.XI.2024, Malaise trap M. Bashford, I. Elliott, A. Kirk-Spriggs (3♀); Lobo Res. Camp 23–25.XI.2024, Lepiled light trap, M. Bashford, I. Elliott, A. Kirk-Spriggs (1♂, 1♀) (ANHRT).

Total. April: 4♂, 1♀; September: 8♂, 9♀; November: 1♂, 4♀.Distribution. Widespread in Africa to Arabian peninsula and Iran.

*Dannfeltia nana* Sjöstedt, 1902Sjöstedt, 1902, Bihang Kungl. Svenska Vet. Akad. Handl. 27(3):,19; type locality: Dannfelt (Democratic Republic of Congo); depository: NHRS, Stockholm (♂ holotypus).Material examined. Republic of Congo, Odzala-Kokoua NP, Kokoua base 5–13.IX.2024, MV light trap, M. Bashford, G. László, M. Talani, A. Volynkin (1♂); Lobo Res. Camp 20–27.IX.2024, MV light trap, M. Bashford, G. László, A. Volynkin (2♂); Lobo Res. Camp 21–22.IX.2024, Actinic light trap, M. Bashford, G. László, M. Talani, A. Volynkin (1♂); Lobo Res. camp 23–25.XI.2024, Lepiled light trap, M. Bashford, I. Elliott, A. Kirk-Spriggs (1♂); Lobo Res. camp 23–27.XI.2024, Actinic light trap, M. Bashford, I. Elliott, A. Kirk-Spriggs (1♂) (ANHRT).Total. September: 4♂; November: 2♂.Distribution. Previously known from Democratic Republic of Congo, Gabon, Central African Republic and Ivory Coast [18].

*Pleothrix conradti* (Bólivar, 1906)Bolívar, 1906, Mem. Soc. espan. Hist. nat. 1: 329 (*Pyrrhicia*); type locality: Cameron: depository: MNCN, Madrid (♂ lectotype).Material examined. Republic of Congo, Odzala-Kokoua NP, Mbomo Headquarters 28.IX–1.X.2024, MV light trap, M. Bashford, G. László, A. Volynkin (1♀) (ANHRT).Distribution. Rare species, previously known only from the typical series (2♂, 2♀ from Cameroon), discovered 118 years after its description in the Odzala-Kokoua National Park (Republic of Congo).

*Phanreticula* n. gen. (Figure 11 and Figure 12)urn:lsid:zoobank.org:act:555A24AC-3E6E-481B-B751-D88673F0DFD8Type species: *Phanreticula fenestrata* n. sp., here designated.

Diagnosis. Characterized by fastigium of vertex just wider than first antennal segment, pronotum slightly saddle-shaped, long legs, fore coxae unarmed, open tympana, and tegmina evidently reticulate.

Description. General habitus. Head and antennae. Fastigium of vertex as wide as first antennal segment, not contiguous with fastigium of frons, sulcate above. Eyes oval, prominent, face smooth, wider than high with lateral keels. Thorax. Pronotum smooth, slightly saddle-shaped, as high as it is long, with a well-developed humeral excision, lateral lobes widely rounded. Legs. Long, fore coxae unarmed, fore and mid femora unarmed, hind femora armed, fore and mid-tibiae ventrally armed. Open tympana on inner and on outer margins of fore tibiae. Tegmina translucent, evidently reticulate, narrow, well developed, shorter than hind wings. Styli absent.

Etymology. The name *Phanreticula* means that it is a Phaneropterinae characterized by evident reticulate veinlets on the tegmina.

Remarks. Neither the keys of Chopard [19], in which the author makes the division between genera with armed and unarmed coxae, nor those of Ragge [20] on the genera with fore tibiae provided with open tympana, made it possible to arrive at the identification of the genus.

*Phanreticula fenestrata* n. sp. (Figure 11 and Figure 12)urn:lsid:zoobank.org:act:57DC6FEC-32B8-4E38-A10B-4D7C6D93B80E

Material examined. Republic of Congo, Odzala-Kokoua NP, Kokoua base (540 m) 01°28′39″ N, 15°16′42″ E, 5–13.IX.2024, MV light trap, M. Bashford, G. Lászlo, M. Talani, A. Volynkin (♂ holotypus) (ANHRT).

Description. Male (Figure 11A). *General habitus and colour*. Medium-sized species, brownish with some darker spots on tegmina. *Head and antennae*. Antennae thin. Fastigium of vertex sulcate above, as wide as scapus of antenna, meeting equally broad fastigium of frons along a horizontal line; face with frontogenal carinae; eyes elongate and oval (Figure 11B and Figure 12E). *Thorax*. Pronotum slightly saddle-shaped, as long as it is high, with evident latero-posterior margin, smooth disc, anterior margin incurved, posterior margin rounded (Figure 11B,C). *Wings*. Both pairs of wings fully developed; tegmina rounded at tips, 5.0 times as long as broad. Stridulatory area characterized by a transparent surface (2.7 × 0.3 mm) between two veinlets (Figure 11D). Stridulatory file arched, consisting of ca. 90 evenly spaced teeth (Figure 12A). *Legs*. Fore coxa unarmed. Fore femora dorsally and ventrally unarmed. Mid femora with two outer ventral spines. Hind femora with eight ventral outer and inner spines. Fore tibiae with open tympanum on both sides, dorsal margin furrowed, with ventral double row of four spines; mid-tibiae with ventral double row of seven spines. Hind tibiae with double row of 12 spines on ventral side and many on dorsal side; three slender sclerotized spurs on each side. *Abdomen*. Last abdominal tergite unmodified, straight. Cerci stout, apically narrowing and incurved with a pointed tip. Subgenital plate narrow with two small apical tips, similar to styli (Figure 12B–D).

Female. Unknown.

Measurements (mm). Males. Body length: 19.6; pronotum length: 3.5; pronotum height: 3.5; length of tegmina: 29.0; width of tegmina: 5.8; length of hind femur: 17.5.

Diagnosis. *Phanreticula fenestrata* gen. and sp. n. is very characteristic for the absence of a spine on the fore coxae, the presence of a transparent window on tegmina, which are evidently reticulate, the pronotum slightly saddle-shaped, as long as it is high, and the fore tibiae with open tympana on both sides.

Distribution. Presently known only from the Odzala-Kokoua National Park (Republic of Congo).

Etymology. The specific name of *Phanreticula fenestrata* gen. and sp. n. derives from the window (in Latin *fenestra*) present on the tegmina.

Remarks. This taxon is very peculiar for its absence of spines on fore coxae and for the shape of the pronotum. It is difficult to determine the tribe to which it belongs, but tentatively it is placed among the Phaneropterini.

**Figure 11 insects-16-00241-f011:**
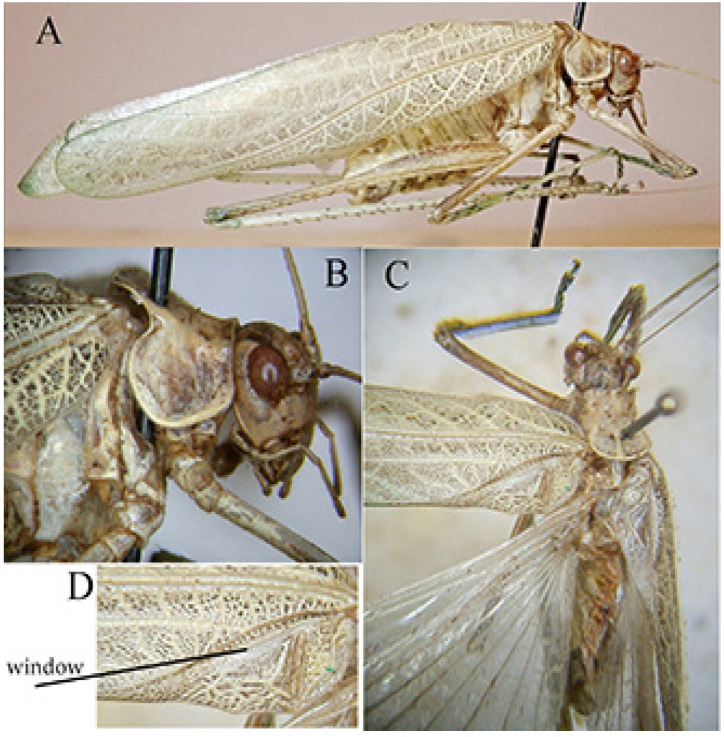
*Phanreticula fenestrata* n. gen. n. sp. (**A**): male habitus in lateral view; (**B**): lateral view of head and pronotum; (**C**): dorsal view of head, pronotum, and stridulatory area of tegmina; (**D**): detailed view of transparent window on left tegmen.

**Figure 12 insects-16-00241-f012:**
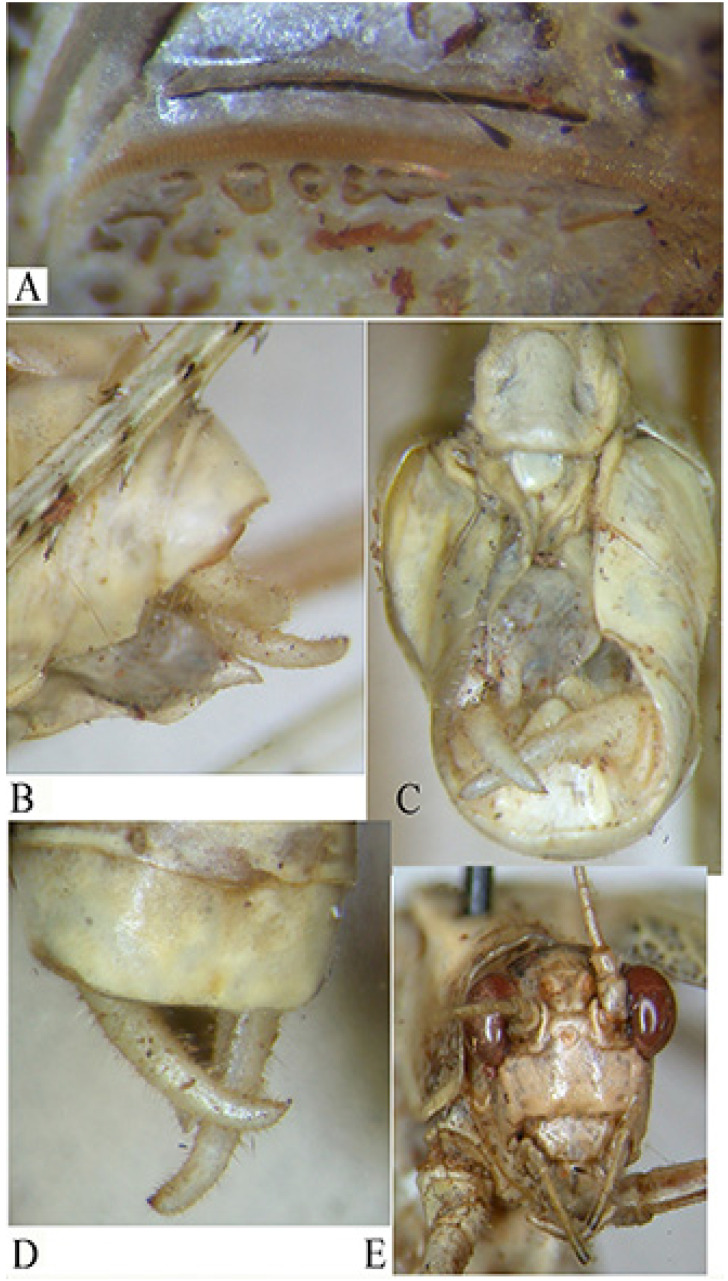
*Phanreticula fenestrata* n. gen. n. sp. (**A**): stridulatory file on underside of left tegmen of male; (**B**): lateral view of last tergites and cerci; (**C**): ventral view of subgenital plate and cerci; (**D**): dorsal view of cerci; (**E**): frontal view of male head.

Tribe Catoptropterigini Massa, 2016Genus *Paraeulioptera* Massa, 2020Considerations on the systematic position

When the genus *Paraeulioptera* has been described [21], it has been considered belonging to the tribe Phaneropterini Burmeister, 1838, related to *Phaneroptera* Serville, 1831 and *Eulioptera* Ragge, 1956. However, the shape of the ovipositor is much reduced (Figure 13A) and completely different from any species of the tribe Phaneropterini; indeed, Phaneropterini have the ovipositor flattened laterally, specialized for the oviposition within the leaf surface or other tissues of plants. Other tribes of Phaneropterinae have a reduced ovipositor, such as Otiaphysini Karsch, 1889, Phlaurocentrini Karsch, 1889, and Myllocentrini Massa, 2023, but only in Catoptropterigini Massa, 2016 is the ovipositor is lacking teeth, slightly flattened laterally, and just swollen, and its valves are covered by dense bristles (probably sensory) (Figure 13B,D), such as in the females of *Catoptropteryx*. Thus, the characteristics of its ovipositor make it possible to establish that *Paraeulioptera emitflesti* belongs to the tribe Catoptropterigini. The shape of the ovipositor could indicate that the eggs are not inserted between the layers of the leaf epidermis. The small size of the valves argues in favour of the use of crevices between cracks of tree bark as oviposition sites (see Huxley [22]).

**Figure 13 insects-16-00241-f013:**
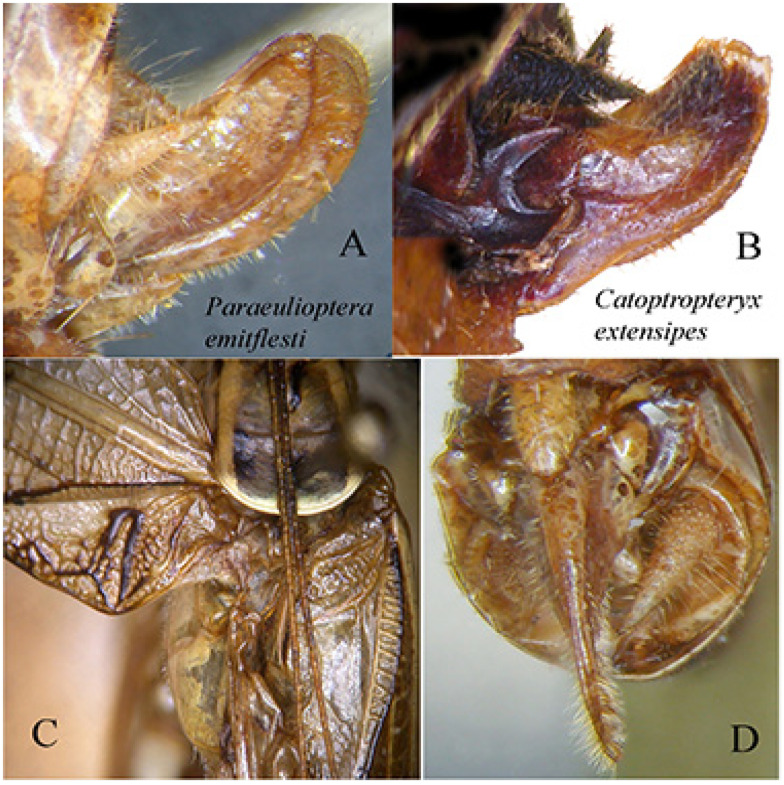
(**A**): lateral view of ovipositor of *Paraeulioptera emitflesti*; (**B**): lateral view of ovipositor of *Catoptropteryx extensipes*; (**C**): dorsal view of pronotum and stridulatory area of *Paraeulioptera emitflesti*; (**D**): ventral view of ovipositor of *Paraeulioptera emitflesti*.

*Paraeulioptera emitflesti* Massa, 2020 (Figure 13 and Figure 14)Massa, 2020, in: Massa, Annoyer, Perez, Danflous & Duvot, Zootaxa 4780(3): 409; type locality: Dzanga-Sangha Special Reserve (Central African Republic); depository: BMPC, Palermo (♂ holotypus).

Material examined. Republic of Congo, Odzala-Kokoua NP, Imbalanga Camp 5–9.IV.2024, MV Light trap, M. Bashford, G. László, M. Talani, S. Yaba Ngouma (1♂, 1♀); Kokoua base 5–13.IX.2024, MV light trap, M. Bashford, G. Lászlo, M. Talani, A. Volynkin (3♂, 1♀) (ANHRT).

Total. April: 1♂, 1♀; September: 3♂, 1♀.

Remarks. These specimens include four males and two females of one species known only from the holotype male. The characteristics of them are here reported (Figure 13A,C,D and Figure 14A,B).

Males (only differences from the original description) (Figure 14A). Antennae longer than body with tegmina, pronotum greyish bordered with cream, as high as it is long.

Female (Figure 14B). Same characteristics as the male, very small triangular subgenital plate, cerci conical and pointed, ovipositor very reduced (Figure 13A,D).

Phenology. It seems to be active as an adult in April and in September.

Measurements (mm; in parenthesis measures of the holotype). Males. Body length: 17.0–20.6 (16.7); length of pronotum: 3.2–4.0 (3.2); pronotum depth: 3.6–4.0 (3.7); length of fore femora: 4.8–6.0 (4.9); length of mid femora: 7.1–8.0 (7.3); length of hind femora: 20.5–22.4 (20.8); length of tegmina: 30.0–31.0 (30.5); width of tegmina: 4.8–5.1 (4.9); Females: body length: 23.2–23.3; length of pronotum: 4.1–4.2; pronotum depth: 4.1–4.2; length of fore femora: 5.5; length of mid femora: 8.0–8.1; length of hind femora: 21.8–22.9; length of tegmina: 30.8–31.2; width of tegmina: 5.2–5.3; length of ovipositor: 4.0–4.1.

**Figure 14 insects-16-00241-f014:**
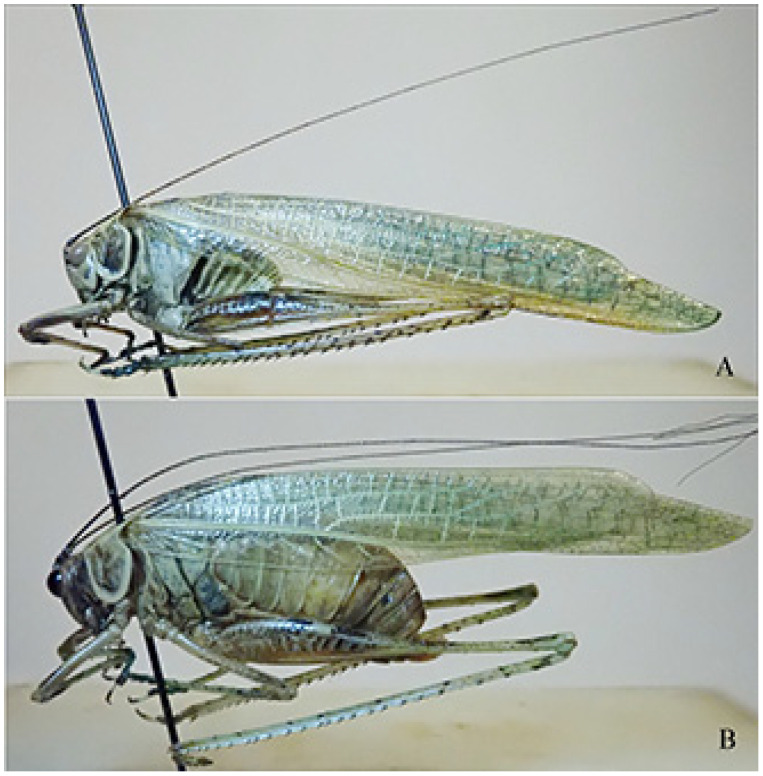
*Paraeulioptera emitflesti*. (**A**): male habitus in lateral view; (**B**): female habitus in lateral view.

*Griffinipteryx mukonja* (Griffini, 1908) (Figure 15)

Material examined. Republic of Congo, Odzala-Kokoua NP, Mbomo Headquarters 28.IX–1.X.2024, MV light trap, M. Bashford, G. László, A. Volynkin (1♂) (ANHRT).

Remarks. The present male, whose colours made it possible to identify it as *Griffinipteryx mukonja* (Griffini 1908), is the first male known of this rare species, described by Griffini [23] as *Polichne mukonja* for the female sex (Figure 15A). More recently, Massa [24] highlighted that this species does not belong to the genus *Polichne* Stål, 1874, of Australia and Papua New Guinea, and established the tropical African genus *Griffinipteryx* for it.

Description of male. *Colour*. Head, dorsal pronotum, and lateral abdomen cream, dorsal abdomen blackish; a black line on lateral pronotum continuing along the fore margin of tegmina and finishing at their middle. Hind part of tegmina dark brown, stridulatory area cream with two black lines. Hind femora black, hind tibiae black with two white rings (Figure 15B,C). *Legs*. Fore coxae armed, fore femora unarmed, fore tibiae with 4 spines on inner and on outer ventral margins, mid femora unarmed, mid-tibiae with 4 spines on inner and outer ventral margins, hind femora with 5 spines on ventral margins, hind tibiae with 7–8 spines on outer and inner ventral margins. *Tegmina*. Stridulatory area just raised. Stridulatory file with ca. 50 evenly spaced teeth (Figure 15D). *Abdomen*. Cerci long, widely incurved (Figure 15E). Subgenital plate narrow with a small emargination, styles very short.

Measurements (mm). Male. Total length: 19.4; length of pronotum: 5.3; height of pronotum: 4.6; length of hind femora: 25.5; length of tegmina: 29.2; width of tegmina: 4.2.

Distribution. This species was known from three females from Cameroon; presently, its distribution also covers the Republic of Congo.

**Figure 15 insects-16-00241-f015:**
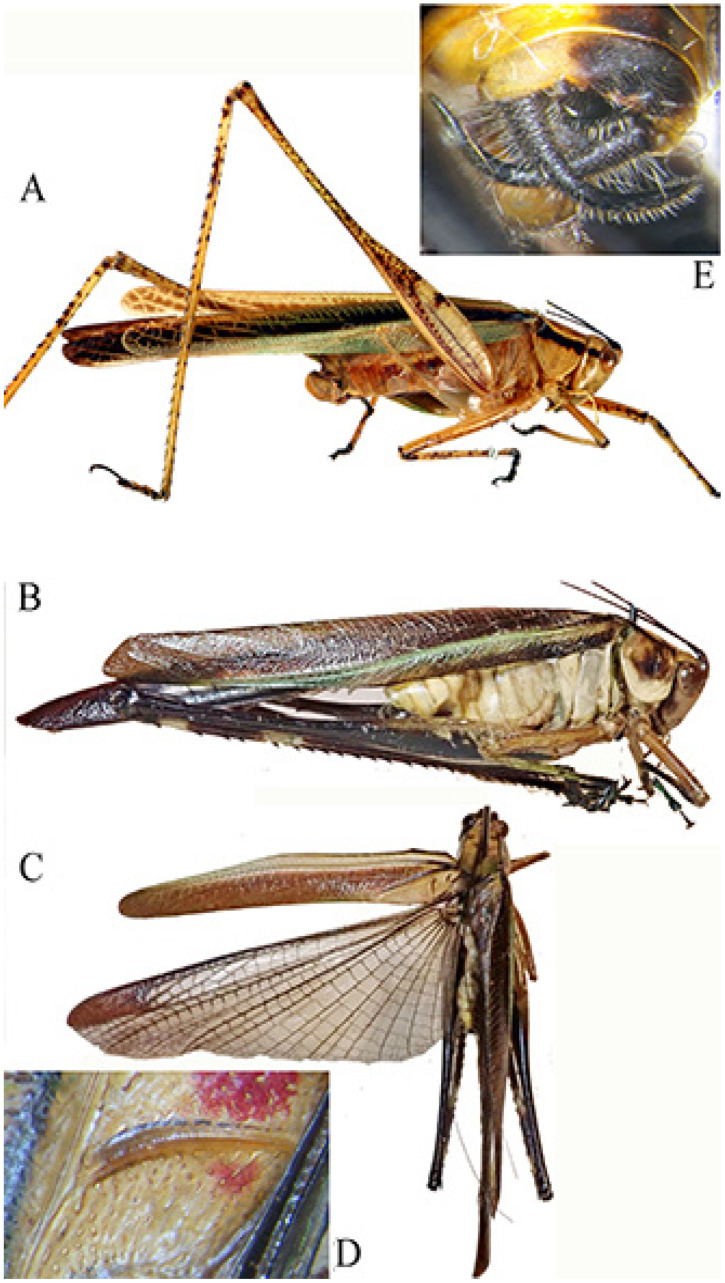
*Griffinipteryx mukonja* (Griffini, 1908). (**A**): lateral view of holotype female preserved at RBINS, Bruxelles (photo by Jérôme Constant); (**B**): male here described in lateral view; (**C**): same in dosal view; (**D**): stridulatory file on underside of left tegmen; (**E**): cerci in dorsal view.

*Catoptropteryx extensipes* Karsch, 1896Karsch, 1896, Stett. Entomol. Z. 57: 242–359; type locality: Lolodorf (Cameroon); depository: MfN (♂ holotypus).Material examined. Republic of Congo, Odzala-Kokoua NP, Lobo Res. Camp 13–18.IV.2024, MV Light trap, M. Bashford, G. László, M. Talani, S. Yaba Ngouma (2♂); Kokoua base 5–13.IX.2024, MV light trap, M. Bashford, G. Lászlo, M. Talani, A. Volynkin (5♂); Kokoua base 11–13.IX.2024, Lepiled light trap, M. Bashford, G. Lászlo, M. Talani, A. Volynkin, S. Yaba Ngouma (1♂); Kokoua Base 6–13.IX.2024, Actinic Light trap, M. Bashford, G. László, M. Talani, A. Volynkin, S. Yaba Ngouma (1♀); Mboko 18–21.XI.2024 MV light trap, M. Bashford, I. Elliott, A. Kirk-Spriggs (1♂) (ANHRT).Total. April: 2♂; September: 6♂, 1♀; November: 1♂.Distribution. Uncommon, but widespread in west-central tropical Africa.

*Catoptropteryx punctulata* (Karsch, 1890)Karsch, 1890, Entom. Nachricht. 16: 260; type locality: Kribi (Cameroon); depository: MfN (♀ holotypus).Material examined. Republic of Congo, Odzala-Kokoua NP, Imbalanga Camp 5–9.IV.2024, MV Light trap, M. Bashford, G. László, M. Talani, S. Yaba Ngouma (18♂, 7♀); Imbalanga Camp 7–8.IV.2024, Lepiled light trap, M. Bashford, G. László, M. Talani, S. Yaba Ngouma (2♂, 1♀); Mbomo Headquarters 19–22.IV.2024, MV Light trap, M. Bashford, G. László, M. Talani, S. Yaba Ngouma (2♂, 2♀); Mbomo Headquarters 5–11.IV.2024, Actinic Light trap, M. Bashford, G. László, M. Talani, S. Yaba Ngouma (2♂, 2♀); Imbalanga Camp 4–5.IX.2024, Actinic Light trap, M. Bashford, G. László, M. Talani, A. Volynkin (5♂, 1♀); Kokoua base 5–13.IX.2024, MV light trap, M. Bashford, G. Lászlo, M. Talani, (43♂, 22♀); Kokoua Base 6–13.IX.2024, Actinic Light trap, M. Bashford, G. László, M. Talani, A. Volynkin, S. Yaba Ngouma (1♂); Imbalanga Camp 13–20.IX.2024, MV Light trap, M. Bashford, G. László, M. Talani, A. Volynkin (2♂, 2♀); Lobo Res. Camp 20–27.IX.2024, MV light trap, M. Bashford, G. László, A. Volynkin (1♀); Kilo Forest 23–24.IX.2024, Lepiled Light trap, M. Bashford, G. László, M. Talani, A. Volynkin, S. Yaba Ngouma (1♂); Bongassou Forest near Lubo 23–26.IX.2024, Actinic Light trap, M. Bashford, G. László, M. Talani, A. Volynkin, S. Yaba Ngouma (1♂, 2♀); Lobo Res. Camp 25–26.IX.2024, Lepiled Light trap, M. Bashford, G. László, M. Talani, A. Volynkin, S. Yaba Ngouma (1♂); Lobo Res. Camp 25–26.IX.2024, Lepiled light trap, M. Bashford, G. László, A. Volynkin, S. Yaba Ngouma (3♂); Mbomo Headquarters 28.IX–1.X.2024, MV light trap, M. Bashford, G. László, A. Volynkin (12♂, 5♀); Mbomo Headquarters 28.IX–1.X.2024, Lepiled light trap, M. Bashford, G. László, M. Talani, A. Volynkin, S. Yaba Ngouma (4♂, 3♀); Camp Imbalanga 14–18.XI.2024, MV light trap, M. Bashford, I. Elliott, A. Kirk-Spriggs (4♂, 3♀); Camp Imbalanga 14–15 and 17–18.XI.2024, Lepiled light trap, M. Bashford, I. Elliott, A. Kirk-Spriggs (1♀); Mboko 18–21.XI.2024 MV light trap, M. Bashford, I. Elliott, A. Kirk-Spriggs (5♂, 4♀); Mboko 18–21.XI.2024, Lepiled light trap, M. Bashford, I. Elliott, A. Kirk-Spriggs (2♀); Lobo Res. Camp 23–27.XI.2024, Actinic light trap M. Bashford, I. Elliott, A. Kirk-Spriggs (1♀); Lobo Res. Camp 23–25.XI.2024, Lepiled light trap, M. Bashford, I. Elliott, A. Kirk-Spriggs (2♂, 1♀) (ANHRT).Total. April: 24♂, 12♀; September: 75♂, 38♀; November: 11♂, 12♀.Phenology. Only three males collected in September had a spermatophore.Distribution. Common and widespread in tropical Africa.Remarks. As reported by Huxley (1970) the black spots on tegmina characteristic of this species are not always discerned; many specimens from the Republic of Congo show completely dark tegmina and it is impossible to detect the spots; however, the stridulatory file and shape of cerci match those of *C. punctulata* (see Huxley [22]).

*Catoptropteryx capreola* Karsch, 1896Karsch, 1896, Stett. Entomol. Z. 57: 332, 335; type locality: Lolodorf (Cameroon); depository: MfN (♂ holotypus).

Material examined. Republic of Congo, Odzala-Kokoua NP, Imbalanga Camp 5–9.IV.2024, MV Light trap, M. Bashford, G. László, M. Talani, S. Yaba Ngouma (15♂, 5♀); Imbalanga Camp 7–8.IV.2024, Lepiled light trap, M. Bashford, G. László, M. Talani, S. Yaba Ngouma (3♂); Moba 11–12.IV.2024, Actinic Light trap, M. Bashford, G. László, M. Talani, S. Yaba Ngouma (2♂, 3♀); Mbomo Headquarters 19–22.IV.2024, MV Light trap, M. Bashford, G. László, M. Talani, S. Yaba Ngouma (2♂, 1♀); Mbomo Headquarters 5–11.IV.2024, Actinic Light trap, M. Bashford, G. László, M. Talani, S. Yaba Ngouma (4♂, 2♀); Lobo Res. Camp 16–17.IV.2024, Lepiled light trap, M. Bashford, G. László, M. Talani, S. Yaba Ngouma (1♂); Lobo Res. Camp 12–19.IV.2024, Actinic Light trap, M. Bashford, G. László, M. Talani, S. Yaba Ngouma (1♂, 1♀); Mbomo Headquarters 19–21.IV.2024, Actinic Light trap, M. Bashford, G. László, M. Talani, S. Yaba Ngouma (1♀); Imbalanga Camp 4–5.IX.2024, Actinic Light trap, M. Bashford, G. László, M. Talani, A. Volynkin (2♂, 6♀); Kokoua base 5–13.IX.2024, MV light trap, M. Bashford, G. László, M. Talani, A. Volynkin (35♂, 20♀); Kokoua base 11–13.IX.2024, Lepiled light trap, M. Bashford, G. Lászlo, M. Talani, A. Volynkin, S. Yaba Ngouma (2♂, 1♀); Camp Imbalanga 13–19.IX.2024, Actinic Light trap, M. Bashford, G. László, A. Volynkin, S. Yaba Ngouma (1♀); Imbalanga Camp 13–20.IX.2024, MV Light trap, M. Bashford, G. László, M. Talani, A. Volynkin (2♀); Lobo Res. Camp 20–27.IX.2024, MV light trap, M. Bashford, G. László, A. Volynkin (7♂, 6♀); Lekoli river near Mboko 21–23.IX.2024, Lepiled Light trap, M. Bashford, G. László, M. Talani, A. Volynkin, S. Yaba Ngouma (1♂, 2♀); Lekoli river near Mboko 22–23.IX.2024, Actinic Light trap, M. Bashford, G. László, M. Talani, A. Volynkin, S. Yaba Ngouma (2♂); Bongassou Forest near Lubo 23–26.IX.2024, Actinic Light trap, M. Bashford, G. László, M. Talani, A. Volynkin, S. Yaba Ngouma (2♂, 1♀); Lobo Res. Camp 25–26.IX.2024, Lepiled Light trap, M. Bashford, G. László, M. Talani, A. Volynkin, S. Yaba Ngouma (1♂); Kokoua Base 6–13.IX.2024, Actinic Light trap, M. Bashford, G. László, M. Talani, A. Volynkin, S. Yaba Ngouma (1♀); Kilo Forest 23–24.IX.2024, Lepiled Light trap, M. Bashford, G. László, M. Talani, A. Volynkin, S. Yaba Ngouma (1♂); Mbomo Headquarters 28–30.IX.2024, Actinic light trap, M. Bashford, G. László, M. Talani, A. Volynkin, S. Yaba Ngouma (1♂); Mbomo Headquarters 28.IX–1.X.2024, MV light trap, M. Bashford, G. László, A. Volynkin (10♂, 3♀); Mbomo Headquarters 28.IX–1.X.2024, Lepiled light trap, M. Bashford, G. László, M. Talani, A. Volynkin, S. Yaba Ngouma (8♂, 2♀); Camp Imbalanga 14–18.XI.2024, MV light trap, M. Bashford, I. Elliott, A. Kirk-Spriggs (10♂, 12♀); Camp Imbalanga 14–15 and 17–18.XI.2024, Lepiled light trap, M. Bashford, I. Elliott, A. Kirk-Spriggs (1♂, 1♀); Mboko 18–21.XI.2024, Lepiled light trap, M. Bashford, I. Elliott, A. Kirk-Spriggs (1♂); Mboko 18–21.XI.2024 MV light trap, M. Bashford, I. Elliott, A. Kirk-Spriggs (11♂, 5♀); Mboko 18–21.XI.2024, Actinic light trap M. Bashford, I. Elliott, A. Kirk-Spriggs (1♀); Lobo Res. Camp 22–30.XI.2024, MV light trap, M. Bashford, I. Elliott, A. Kirk-Spriggs (4♂); Lobo Res. Camp 23–27.XI.2024, Actinic light trap M. Bashford, I. Elliott, A. Kirk-Spriggs (1♀); Lobo Res. Camp 23–25.XI.2024, Lepiled light trap, M. Bashford, I. Elliott, A. Kirk-Spriggs (3♂) (ANHRT).

Total. April: 28♂, 12♀; September: 72♂, 44♀; November: 30♂, 20♀.Phenology. Only two males collected in September had a spermatophore.Distribution. Very common and widespread in west-central tropical Africa.

*Catoptropteryx naevia* Huxley, 1970Huxley, 1970, Bull. Br. Mus. (Nat. Hist.) Ent. 24(5): 153; type locality: Marshall Territory (Liberia); depository: ANSP Philadelphia (♂ holotypus).Material examined. Republic of Congo, Odzala-Kokoua NP, Imbalanga Camp 5–9.IV.2024, MV Light trap, M. Bashford, G. László, M. Talani, S. Yaba Ngouma (1♂); Lobo Res. Camp 16–17.IV.2024, Lepiled light trap, M. Bashford, G. László, M. Talani, S. Yaba Ngouma (1♂); Kokoua base 5–13.IX.2024, MV light trap, M. Bashford, G. László, M. Talani, A. Volynkin (2♀); Mbomo Headquarters 28.IX–1.X.2024, MV light trap, M. Bashford, G. László, A. Volynkin (1♂) (ANHRT).Total. April: 2♂; September: 1♂, 2♀.Phenology. One male collected in September had a spermatophore.Distribution. Uncommon in tropical Africa, from western countries to Zambia.

*Catoptropteryx nana* Huxley, 1970Huxley, 1970, Bull. Br. Mus. (Nat. Hist.) Ent. 24(5): 147 (*Catoptropteryx nanus*); type locality: Efulen (Cameroon); depository: ANSP Philadelphia (♀ holotypus).Material examined. Republic of Congo, Odzala-Kokoua NP, Mbomo Headquarters 10–22.IV.2024, MV Light trap, M. Bashford, G. László, M. Talani, S. Yaba Ngouma (3♂, 1♀); Imbalanga Camp 13–20.IX.2024, MV Light trap, M. Bashford, G. László, M. Talani, A. Volynkin (1♀); Lekoli river near Mboko 21–23.IX.2024, Lepiled Light trap, M. Bashford, G. László, M. Talani, A. Volynkin, S. Yaba Ngouma (1♀); Lobo Res. Camp 20–27.IX.2024, MV light trap, M. Bashford, G. László, A. Volynkin (2♀); Mbomo Headquarters 28.IX–1.X.2024, MV light trap, M. Bashford, G. László, A. Volynkin (1♂) (ANHRT).Total. April: 3♂, 1♀; September: 1♂, 4♀.Distribution. Uncommon in west and central tropical African countries.

*Catoptropteryx apicalis* Bolívar, 1893Bolívar, 1893, Ann. Soc. ent. Fr. 62: 177 (*Caedicia apicalis*); type locality: Assinie (Ivory Coast); depository: MNCN (♀ holotypus).Material examined. Republic of Congo, Odzala-Kokoua NP, Imbalanga Camp 5–9.IV.2024, MV Light trap, M. Bashford, G. László, M. Talani, S. Yaba Ngouma (13♂, 25♀); Imbalanga Camp 7–8.IV.2024, Lepiled light trap, M. Bashford, G. László, M. Talani, S. Yaba Ngouma (2♂, 4♀); Mbomo Headquarters 19–22.IV.2024, MV Light trap, M. Bashford, G. László, M. Talani, S. Yaba Ngouma (4♀); Mbomo Headquarters 5–11.IV.2024, Actinic Light trap, M. Bashford, G. László, M. Talani, S. Yaba Ngouma (2♂, 2♀); Mbomo Headquarters 19–21.IV.2024, Actinic Light trap, M. Bashford, G. László, M. Talani, S. Yaba Ngouma (1♀); Imbalanga Camp, 4–5.IX.2024, Actinic Light trap, M. Bashford, G. László, M. Talani, A. Volynkin (2♂); Kokoua base 5–13.IX.2024, MV light trap, M. Bashford, G. László, M. Talani, A. Volynkin (19♂, 36♀); Imbalanga Camp 13–20.IX.2024, MV Light trap, M. Bashford, G. László, M. Talani, A. Volynkin (1♀); Lekoli river near Mboko 21–23.IX.2024, Lepiled Light trap, M. Bashford, G. László, M. Talani, A. Volynkin, S. Yaba Ngouma (2♂, 1♀); Lobo Res. Camp 20–27.IX.2024, MV light trap, M. Bashford, G. László, A. Volynkin (2♂, 1♀); Lobo Res. Camp 25–26.IX.2024, Lepiled Light trap, M. Bashford, G. László, M. Talani, A. Volynkin, S. Yaba Ngouma (2♀); Mbomo Headquarters 28.IX–1.X.2024, MV light trap, M. Bashford, G. László, A. Volynkin (22♂, 17♀); Mbomo Headquarters 28.IX–1.X.2024, Lepiled light trap, M. Bashford, G. László, M. Talani, A. Volynkin, S. Yaba Ngouma (13♂, 6♀); Camp Imbalanga 14–18.XI.2024, MV light trap, M. Bashford, I. Elliott, A. Kirk-Spriggs (1♀); Mboko 18–21.XI.2024 MV light trap, M. Bashford, I. Elliott, A. Kirk-Spriggs (1♀); Mboko 18–21.XI.2024, Lepiled light trap, M. Bashford, I. Elliott, A. Kirk-Spriggs (1♀); Bongassou Forest near Lobo 26–28.XI.2024, Lepiled light trap M. Bashford, I. Elliott, A. Kirk-Spriggs (1♀) (ANHRT).Total. April: 17♂, 36♀; September: 60♂, 63♀; November: 4♀.Distribution. Common and widespread in tropical Africa.

*Catoptropteryx guttatipes* Karsch, 1890Karsch, 1890, Entom. Nachricht. 16(23): 362; type locality: Barombi Station (Cameroon); depository: MfN (♂ holotypus).Material examined. Republic of Congo, Odzala-Kokoua NP, Imbalanga Camp 5–9.IV.2024, MV Light trap, M. Bashford, G. László, M. Talani, S. Yaba Ngouma (4♂, 1♀); Imbalanga Camp 7–8.IV.2024, Lepiled light trap, M. Bashford, G. László, M. Talani, S. Yaba Ngouma (2♂); Moba 11–12.IV.2024, Actinic Light trap, M. Bashford, G. László, M. Talani, S. Yaba Ngouma (1♂); Mbomo Headquarters 19–22.IV.2024, MV Light trap, M. Bashford, G. László, M. Talani, S. Yaba Ngouma (1♂); Mbomo Headquarters 5–11.IV.2024, Actinic Light trap, M. Bashford, G. László, M. Talani, S. Yaba Ngouma (6♂); Lobo Res. Camp 16–17.IV.2024, Lepiled light trap), M. Bashford, G. László, M. Talani, S. Yaba Ngouma (1♂); Imbalanga Camp 5–11.IV.2024, Actinic Light trap, M. Bashford, G. László, M. Talani, S. Yaba Ngouma (2♂, 2♀); Lobo Res. Camp 13–18.IV.2024, MV Light trap, M. Bashford, G. László, M. Talani, S. Yaba Ngouma (2♂); Imbalanga Camp 4–5.IX.2024, Actinic Light trap, M. Bashford, G. László, M. Talani, A. Volynkin (2♂, 1♀); Imbalanga Camp 13–20.IX.2024, MV Light trap, M. Bashford, G. László, M. Talani, A. Volynkin (1♂); Mbomo Headquarters 21–22.IV.2024, MV Light trap, M. Bashford, G. László, M. Talani, S. Yaba Ngouma (2♂); Kokoua base 5–13.IX.2024, MV light trap, M. Bashford, G. László, M. Talani, A. Volynkin (22♂, 4♀); Kokoua base 11–13.IX.2024, Lepiled light trap, M. Bashford, G. Lászlo, M. Talani, A. Volynkin, S. Yaba Ngouma (2♂); Lekoli river near Mboko 21–23.IX.2024, Lepiled Light trap, M. Bashford, G. László, M. Talani, A. Volynkin, S. Yaba Ngouma (1♂); Lekoli river near Mboko 22–23.IX.2024, Actinic Light trap, M. Bashford, G. László, M. Talani, A. Volynkin, S. Yaba Ngouma (2♂); Kilo Forest 23–24.IX.2024, Lepiled Light trap, M. Bashford, G. László, M. Talani, A. Volynkin, S. Yaba Ngouma (3♂); Bongassou Forest near Lubo 23–26.IX.2024, Actinic Light trap, M. Bashford, G. László, M. Talani, A. Volynkin, S. Yaba Ngouma (2♂); Lobo Res. Camp 25–26.IX.2024, Lepiled light trap, M. Bashford, G. László, A. Volynkin, S. Yaba Ngouma (1♂); Mbomo Headquarters 28–30.IX.2024, Actinic light trap, M. Bashford, G. László, M. Talani, A. Volynkin, S. Yaba Ngouma (3♂, 4♀); Mbomo Headquarters 28.IX–1.X.2024, MV light trap, M. Bashford, G. László, A. Volynkin (9♂, 1♀); Mbomo Headquarters 28.IX–1.X.2024, MV light trap, M. Bashford, G. László, A. Volynkin (20♂, 7♀); Mbomo Headquarters 28.IX–1.X.2024, Lepiled light trap, M. Bashford, G. László, M. Talani, A. Volynkin, S. Yaba Ngouma (6♂, 2♀); Camp Imbalanga 14–18.XI.2024, MV light trap, M. Bashford, I. Elliott, A. Kirk-Spriggs (1♂, 1♀); Mboko 18–21.XI.2024 MV light trap, M. Bashford, I. Elliott, A. Kirk-Spriggs (1♂); Bongassou Forest near Lobo 26–28.XI.2024, Lepiled light trap M. Bashford, I. Elliott, A. Kirk-Spriggs (1♂); Camp Imbalanga 14–15 and 17–18.XI.2024, Lepiled light trap, M. Bashford, I. Elliott, A. Kirk-Spriggs (1♀) (ANHRT).Total. April: 21♂, 4♀; September: 76♂, 19♀; November: 3♂, 2♀.Phenology. One male collected in September had a spermatophore.Distribution. Mainly widespread in central tropical Africa.

*Catoptropteryx ambigua* Huxley, 1970Huxley, 1970, Bull. Br. Mus. (Nat. Hist.) Ent. 24(5): 159; type locality: Bwamba, Ntandi (Uganda); depository: NHM (♀ holotypus).Material examined. Republic of Congo, Odzala-Kokoua NP, Imbalanga Camp 4–5.IX.2024, Actinic Light trap, M. Bashford, G. László, M. Talani, A. Volynkin (1♀); Kokoua base 5–13.IX.2024, MV light trap, M. Bashford, G. Lászlo, M. Talani, A. Volynkin (1♂); Mbomo Headquarters 28–30.IX.2024, Actinic light trap, M. Bashford, G. László, M. Talani, A. Volynkin, S. Yaba Ngouma (1♂) (ANHRT).Total. April: 1♀; September: 2♂.Distribution. Common in west-central tropical Africa.

*Catoptropteryx occidentalis* Huxley, 1970Huxley, 1970, Bull. Br. Mus. (Nat. Hist.) Ent. 24(5): 164; type locality: Bomi Hills (Liberia); depository: NHM (♀ holotypus).Material examined. Republic of Congo, Odzala-Kokoua NP, Imbalanga Camp 5–9.IV.2024, MV Light trap, M. Bashford, G. László, M. Talani, S. Yaba Ngouma (8♂); Imbalanga Camp 7–8.IV.2024, Lepiled light trap, M. Bashford, G. László, M. Talani, S. Yaba Ngouma (6♂); Mbomo Headquarters 19–22.IV.2024, MV Light trap, M. Bashford, G. László, M. Talani, S. Yaba Ngouma (1♂, 1♀); Mbomo Headquarters 5–11.IV.2024, Actinic Light trap, M. Bashford, G. László, M. Talani, S. Yaba Ngouma (1♂); Kokoua base 5–13.IX.2024, MV light trap, M. Bashford, G. Lászlo, M. Talani, A. Volynkin (8♂, 6♀); Mbomo Headquarters 28.IX–1.X.2024, MV light trap, M. Bashford, G. László, A. Volynkin (10♂, 2♀); Kilo Forest 23–24.IX.2024, Lepiled Light trap, M. Bashford, G. László, M. Talani, A. Volynkin, S. Yaba Ngouma (5♂, 3♀); Lobo Res. Camp 25–26.IX.2024, Lepiled Light trap, M. Bashford, G. László, M. Talani, A. Volynkin, S. Yaba Ngouma (1♂); Mbomo Headquarters 28.IX–1.X.2024, Lepiled light trap, M. Bashford, G. László, M. Talani, A. Volynkin, S. Yaba Ngouma (1♂, 2♀) (ANHRT).Total. April: 16♂, 1♀; September: 25♂, 13♀.Phenology. Only two males collected in September had a spermatophore.Distribution. Uncommon in west-central tropical Africa.

Tribe Amblycoryphini Brunner von Wattenwyl, 1878*Plangia deminuta* Griffini, 1908Griffini, 1908. Mem. Soc. entom. Belgique, Bruxelles 15: 222; type locality: Mukonje Farm (Cameroon); depository: RBINS, Bruxelles (♂,♀ syntypes).Material examined. Republic of Congo, Odzala-Kokoua NP, Imbalanga Camp 5–9.IV.2024, MV Light trap, M. Bashford, G. László, M. Talani, S. Yaba Ngouma (1♀); Moba 11–12.IV.2024, Actinic Light trap, M. Bashford, G. László, M. Talani, S. Yaba Ngouma (4♀); Kokoua base 5–13.IX.2024, MV light trap, M. Bashford, G. László, M. Talani, A. Volynkin (3♀); Moba 19–20.IX.2024, Lepiled light trap, M. Bashford, G. Lászlo, M. Talani, A. Volynkin (1♂, 1♀); Kilo Forest 23–24.IX.2024, Lepiled light trap, M. Bashford, G. László, M. Talani, A. Volynkin, S. Yaba Ngouma (2♂, 3♀); Mbomo Headquarters 28.IX–1.X.2024, MV light trap, M. Bashford, G. László, A. Volynkin (1♂, 8♀); Mbomo Headquarters 28.IX–1.X.2024, general collection, M. Bashford, G. László, M. Talani, A. Volynkin, S. Yaba Ngouma (1♀); Mbomo Headquarters 28.IX–1.X.2024, Lepiled light trap, M. Bashford, G. László, M. Talani, S. Yaba Ngouma (1♀) (ANHRT).Total. April: 5♀; September: 3♂, 9♀.Distribution. Widespread in west-central tropical Africa.

*Plangia astylata* Massa, 2021Massa, 2021, Zootaxa 4974(3): 424; type locality: Dzanga-Sangha Special Reserve (Central African Republic); depository: BMPC, Palermo (♂ holotype).Material examined. Republic of Congo, Odzala-Kokoua NP, Mbomo Headquarters 28.IX–1.X.2024, MV light trap, M. Bashford, G. László, A. Volynkin (1♂) (ANHRT).Total. September: 1♂.Distribution. Known only from the Dzanga-Ndoki National Park and Dzanga-Sangha Special Reserve in the Central African Republic.

*Plangia nebulosa* Krasch, 1890Karsch, 1890, Entom. Nachricht. 16(23): 366; type locality: Barombi Station (Cameroon); depository: MfN, Berlin (♂ holotypus).Material examined. Republic of Congo, Odzala-Kokoua NP, Lobo Res. Camp 20–27.IX.2024, MV light trap, M. Bashford, G. László, A. Volynkin (1♂); Mboko 18–21.XI.2024, MV light trap, M. Bashford, I. Elliott, A. Kirk-Spriggs (1♂) (ANHRT).Total. September: 1♂; November: 1♂.Distribution. Widespread in west-central tropical Africa.

*Pseudoplangia laminifera* (Karsch, 1896)Karsch, 1896, Stett. Entomol. Z. 57: 343; type locality: Lolodorf (Cameroon); depository: MfN, Berlin (♀ holotype).Material examined. Republic of Congo, Odzala-Kokoua NP, Mbomo Headquarters 28.IX–1.X.2024, MV light trap, M. Bashford, G. László, A. Volynkin (1♀); Mboko 18–21.XI.2024, MV light trap, M. Bashford, I. Elliott, A. Kirk-Spriggs (5♂, 1♀) (ANHRT).Total. September: 1♀; November: 5♂, 1♀.Phenology. Two males collected in November had a spermatophore.Distribution. Uncommon in central tropical Africa; also recorded from the Nouabalé-Ndoki National Park in the Republic of Congo [4].

*Monteiroa nigricauda* Ragge, 1980Ragge, 1980. Bull. Br. Mus. (Nat. Hist.) Ent. 40(2): 178; type locality: Toumodi (Ivory Coast); depository: MNHN (♂ holotype).Material examined. Republic of Congo, Odzala-Kokoua NP, Kokoua base 5–13.IX.2024, MV light trap, M. Bashford, G. Lászlo, M. Talani, A. Volynkin (6♀) (ANHRT).Distribution. Uncommon in west-central tropical Africa.

*Corycomima camerata* (Karsch, 1889)Karsch, 1889[1888], Berlin Ent. Z. 32: 457; type locality: Barombi Station (Cameroon); depository: MfN, Berlin (♀ holotype).Material examined. Republic of Congo, Odzala-Kokoua NP, Kokoua base 5–13.IX.2024, MV light trap, M. Bashford, G. Lászlo, M. Talani, A. Volynkin (1♀) (ANHRT).Distribution. Previously known from Cameroon, Central African Republic, Democratic Republic of Congo, Uganda [20,21].

Genus *Eurycorypha* Stål, 1873At least six undescribed species of the genus *Eurycorypha* are present in the Sangha Trinational Protected Area; in the present paper, I am only listing them with a number beside them, as a complex revision of the genus is underway with Claudia Hemp and the current number of species (approx. 50) will be significantly increased. We therefore defer the description of the new central African species to the revision in progress.

*Eurycorypha canaliculata* Karsch, 1890Karsch, 1890, Entom. Nachricht. 16: 261; type locality: Kribi (Cameroon); depository: MfN, Berlin (♂ holotype).Material examined. Republic of Congo, Odzala-Kokoua NP, Lobo Res. Camp 13–18.IV.2024, MV Light trap, M. Bashford, G. László, M. Talani, S. Yaba Ngouma (1♀); Kokoua base 5–13.IX.2024, MV light trap, M. Bashford, G. Lászlo, M. Talani, A. Volynkin (2♂); Mbomo Headquarters 28.IX–1.X.2024, MV light trap, M. Bashford, G. László, A. Volynkin (3♀) (ANHRT).Total. April: 1♀; September: 2♂, 3♀.Distribution. Widespread in central tropical Africa.

*Eurycorypha montana* Sjöstedt, 1902Sjöstedt, 1902, Bihang Kungl. Svenska Vet. Akad. Handl. 27(3): 1–45, pls. I-IV; type locality: Mapanja (Cameroon); depository: NHRS, Stockholm (♀ holotype).Material examined. Republic of Congo, Odzala-Kokoua NP, Imbalanga Camp, 5–9.IV.2024, MV Light trap, M. Bashford, G. László, M. Talani, S. Yaba Ngouma (1♀); Imbalanga Camp 5–11.IV.2024, Actinic Light trap, M. Bashford, G. László, M. Talani, S. Yaba Ngouma (1♂); Kokoua base 5–13.IX.2024, MV light trap, M. Bashford, G. Lászlo, M. Talani, A. Volynkin (1♂, 1♀); Lobo Res. Camp 20–27.IX.2024, MV light trap, M. Bashford, G. László, A. Volynkin (1♂) (ANHRT).Total. April: 1♂, 1♀; September: 2♂, 1♀.Distribution. Probably uncommon in west-central tropical Africa.

*Eurycorypha ornatipes* Karsch, 1890Karsch, 1890, Entom. Nachricht. 16: 260; type locality: Kribi (Cameroon); depository: MfN, Berlin (♂ holotype).Material examined. Republic of Congo, Odzala-Kokoua NP, Imbalanga Camp 5–9.IV.2024, MV Light trap, M. Bashford, G. László, M. Talani, S. Yaba Ngouma (1♂, 1♀); Imbalanga Camp 4–5.IX.2024, Actinic Light trap, M. Bashford, G. László, M. Talani, A. Volynkin (1♂); Kokoua base 5–13.IX.2024, MV light trap, M. Bashford, G. Lászlo, M. Talani, A. Volynkin (17♂, 7♀); Imbalanga Camp 13–20.IX.2024, MV Light trap, M. Bashford, G. László, M. Talani, A. Volynkin (1♂); Mbomo Headquarters 28.IX–1.X.2024, MV light trap, M. Bashford, G. László, A. Volynkin (1♀); Lobo Res. Camp 22–30.XI.2024, MV light trap, M. Bashford, I. Elliott, A. Kirk-Spriggs (1♀) (ANHRT).Total. April: 1♂, 1♀; September: 19♂, 8♀; November: 1♀.Distribution. Widespread in west-central tropical Africa.

*Pseudoeurycorypha civilettorum* Massa, 2023Massa, 2023, Zootaxa 5331(1): 40–43; type locality: Ndoki National Park (Central African Republic); depository: BMPC, Palermo (♂ holotype).Material examined. Republic of Congo, Odzala-Kokoua NP, Mbomo Headquarters 28–30.IX.2024, Actinic light trap, M. Bashford, G. László, M. Talani, S. Yaba Ngouma (1♀); Mbomo Headquarters 28.IX–1.X.2024, MV light trap, M. Bashford, G. László, A. Volynkin (1♂) (ANHRT).Total. September: 1♂, 1♀.Distribution. Known from Dzanga-Ndoki National Park (Central African Republic), Nouabalé-Ndoki National Park (Republic of Congo), and Taï National Park (Ivory Coast) [4].

Tribe Tylopsidini Brunner von Wattenwyl, 1878*Tylopsis irregularis* Karsch, 1893Karsch, 1893, Berlin Ent. Z. 38: 130; type locality: Bismarcksburg (Togo); depository: MfN (♂ lectotype).Material examined. Republic of Congo, Odzala-Kokoua NP, Lobo Res. Camp 20–27.IX.2024, MV light trap, M. Bashford, G. László, A. Volynkin (2♂, 1♀); Lobo Res. Camp 25–26.IX.2024, Lepiled light trap, M. Bashford, G. László, A. Volynkin, S. Yaba Ngouma (1♂); Mboko 18–21.XI.2024 MV light trap, M. Bashford, I. Elliott, A. Kirk-Spriggs (1♂); Lobo Res. Camp 23–26.XI.2024, Malaise trap M. Bashford, I. Elliott, A. Kirk-Spriggs (1♀) (ANHRT).Total. September: 3♂, 1♀; November: 1♂, 1♀.Distribution. Widespread in tropical Africa.

Tribe Phlaurocentrini Karsch, 1889Genus *Phlaurocentrum* Karsch, 1889

General remarks on identification. This genus has been revised by Ragge [25]; later, Massa [4,26] added other species, showing the peculiar apical structures of the abdomen of males, in which titillators are usually hidden or poorly visible; however, they are of considerable diagnostic importance. In addition, in this genus, there are characteristics that should be considered before attempting species identification, such as whether the stridulatory area is bright yellow or not, or whether the appearance of the tegmina is slender (narrow tegmina: *P. latevittatum*, *P. lobatum*, *P. paratuberosum*) or robust (broad tegmina). Lastly, the shape of the supragenital plate can be of two basic types: with lateral reliefs and a central concavity or with a lamellar shape with several variations. Females which are alone, without males, are more difficult to identify, but the subgenital plate allows for the identification of most species. However, since the apex of the abdomen of females is often not perfectly preserved, it is not always possible to obtain a close look at the subgenital plate.

*Phlaurocentrum armatum* n. sp. (Figure 16)urn:lsid:zoobank.org:act:2CDDC9B3-91BA-4912-8D10-D53DE75C2E05

Material examined. Republic of Congo, Odzala-Kokoua NP, Mbomo Headquarters (540 m) 00°26′13″ N, 14°42′01″ E, 28.IX–1.X.2024, MV light trap, M. Bashford, G. László, A. Volynkin (♂ holotypus) (ANHRT).

Diagnosis. *P. armatum* n. sp. is well characterized by its wide brown tegmina (Figure 16A). The characteristics differentiating *P. armatum* n. sp. from other *Phlaurocentrum* species are the following: male 10th tergite ending with two raised keels medially concave with toothed apices (Figure 16B), cerci long and bent at right angles (Figure 16B,C), titillators ivory coloured, short and flat, with toothed brown margins and a preapical inner spine (Figure 16B,C).

Description. Male. *Colour*. Brown, tegmina with black spots; upper part of the head and pronotum darker than the rest of the body. Legs with brown and cream bands. Antennae brown. Stridulatory area of the male yellow. *Legs*. Long, fore coxae armed, fore femora with 7 spines on inner ventral margin, fore tibiae with 8 spines on inner and on outer ventral margins, mid femora with 6 spines on inner ventral margin, mid-tibiae with 8 spines on inner ventral and 9 on outer ventral margin, hind femora with 6 spines on ventral margins, hind tibiae with 10–12 spines on outer and 5–7 on inner ventral margin. *Tegmina*. Stridulatory area raised and stout. *Abdomen*: 10th tergite ending with two raised keels medially concave with toothed apices. Cerci long, bent at right angle, club-shaped apex. Subgenital plate narrow with a small emargination, styles quite long. Titillators short and flat, with toothed margins, provided by a wide preapical inner spine.

Measurements (mm). Male. Total length: 21.0; length of pronotum: 4.8; height of pronotum: 4.0; length of hind femora: 24.1; length of tegmina: 35.0; width of tegmina: 10.4.

Etymology. Named after the armed titillators.

Distribution. Presently known only from the Odzala-Kokoua National Park (Republic of Congo).

Affinities. At least three species have similar abdomen apex to *P. armatum* n. sp.: *P. dentatum*, *P. tuberosum,* and *P. paratuberosum*. *P. dentatum* differs by supragenital plate with two protruding toothed apices (Figure 16D), while *P. tuberosum*, widespread in central tropical Africa, differs does so by its supragenital plate with four raised toothed plates and flat upcurved titillators (Figure 16E) and *P. paratuberosum* by its narrow tegmina and peculiar male titillators, ending with a long fine apex (Figure 16F) (see also Table 1 in Massa [4]).

**Figure 16 insects-16-00241-f016:**
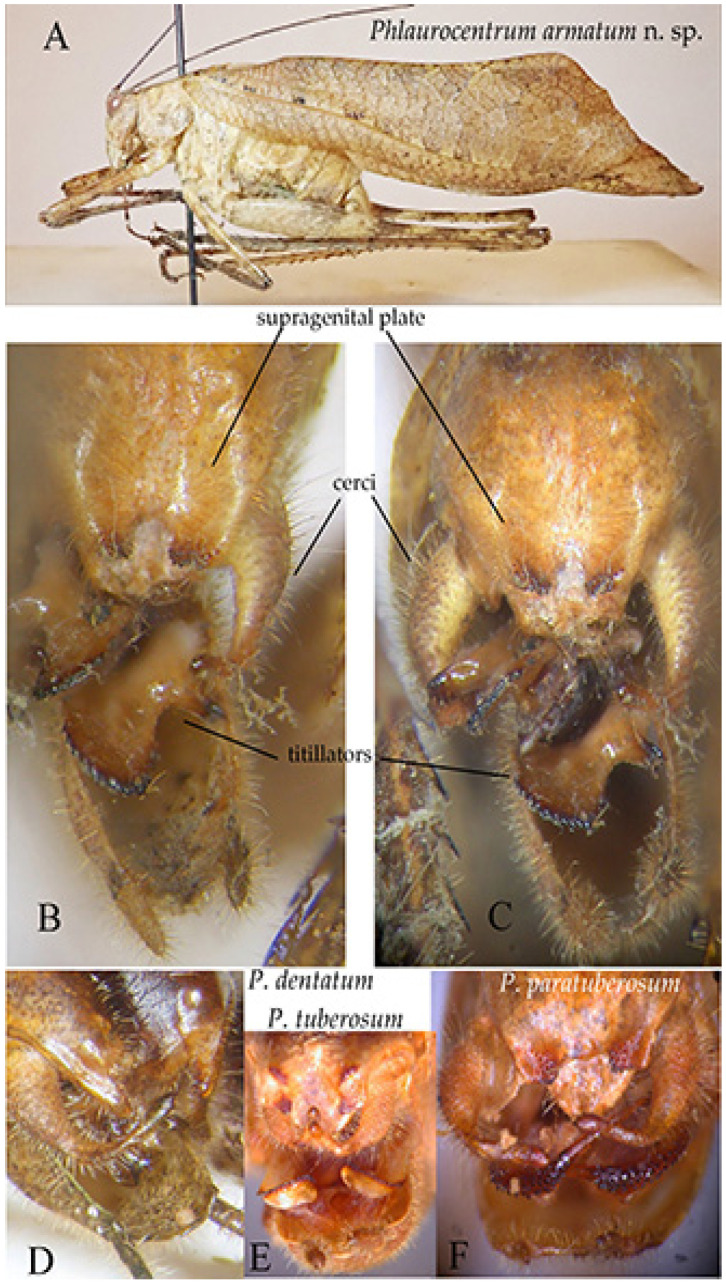
*Phlaurocentrum armatum* n. sp. (**A**): habitus in lateral view; (**B**,**C**): two dorsal views of last tergites showing supragenital plate, cerci, titillators, and subgenital plate; (**D**): genitalia of *Phlauroecentrum dentatum*; (**E**): genitalia of *P. tuberosum*; (**F**): genitalia of *P. paratuberosum*.

*Phlaurocentrum tuberosum* Ragge, 1962Ragge, 1962, Bull. Br. Mus. (Nat. Hist.) Ent. 13: 8; type locality: Mabira Forest (Uganda); depository: NHM (♂ holotype). Material examined. Republic of Congo, Odzala-Kokoua NP, Kokoua base 5–13.IX.2024, MV light trap, M. Bashford, G. László, M. Talani, A. Volynkin (2♂, 1♀); Kokoua base 6–12.IX.2024, carrion bait, M. Bashford, G. Lászlo, M. Talani, A. Volynkin, S. Yaba Ngouma (1♂, 1♀); Mbomo Headquarters 28.IX–1.X.2024, MV light trap, M. Bashford, G. László, A. Volynkin (2♂) (ANHRT).Total. September: 5♂, 2♀.Distribution. Previously known from Uganda, Democratic Republic of Congo, Cameroon, Central African Republic and Gabon [18,25].

*Phlaurocentrum paratuberosum* Massa, 2013Massa, 2013, J. Orth. Res. 22(2): 134; type locality: Dzanga-Ndoki National Park (Central African Republic); depository: BMPC, Palermo (♂ holotype). Material examined. Republic of Congo, Odzala-Kokoua NP, Kokoua base 6–13.IX.2024, Actinic light trap, M. Bashford, G. Lászlo, M. Talani, A. Volynkin, S. Yaba Ngouma (1♂) (ANHRT).Distribution. Known from Central African Republic and Republic of Congo (Nouabalé-Ndoki National Park) [4].

*Phlaurocentrum dentatum* Massa, 2023 (Figure 17A)Massa, 2023, Zootaxa 5331 (1): 30; type locality: Nouabalé-Ndoki National Park (Republic of Congo); depository: ANHRT (♂ holotype). Material examined. Republic of Congo, Odzala-Kokoua NP, Kokoua base 5–13.IX.2024, MV light trap, M. Bashford, G. László, M. Talani, A. Volynkin (♂); Kilo Forest 23–24.IX.2024, Lepiled Light trap, M. Bashford, G. László, M. Talani, A. Volynkin, S. Yaba Ngouma (1♀) (ANHRT).Total. September: 1♂, 1♀.

Description of the unknown female. *Colour*. Brown, upper part of the head and pronotum darker than the rest of the body, antennae brown, legs brownish. *Legs*: long, fore coxae armed, fore femora with 6 spines on the inner ventral margin, fore tibiae with 6 spines on inner and on outer ventral margins, mid femora with 5 spines on inner ventral margin, mid-tibiae with 6 spines on inner and on outer ventral margins, hind femora with 6 spines on ventral margins, hind tibiae with 10 spines on outer and 6 on inner ventral margin. *Abdomen*. Cerci stout and pointed. Subgenital plate with two lateral flat apices and a square central emargination (Figure 17A).

Measurements (mm). Female. Total length: 20.7; length of pronotum: 4.0; height of pronotum: 4.0; length of hind femora: 25.0; length of tegmina: 32.0; width of tegmina: 5.0; ovipositor: 2.0.

Distribution. It has been described from the Nouabalé-Ndoki National Park and at present has also been recorded from Odzala-Kokoua National Park, Republic of Congo.

**Figure 17 insects-16-00241-f017:**
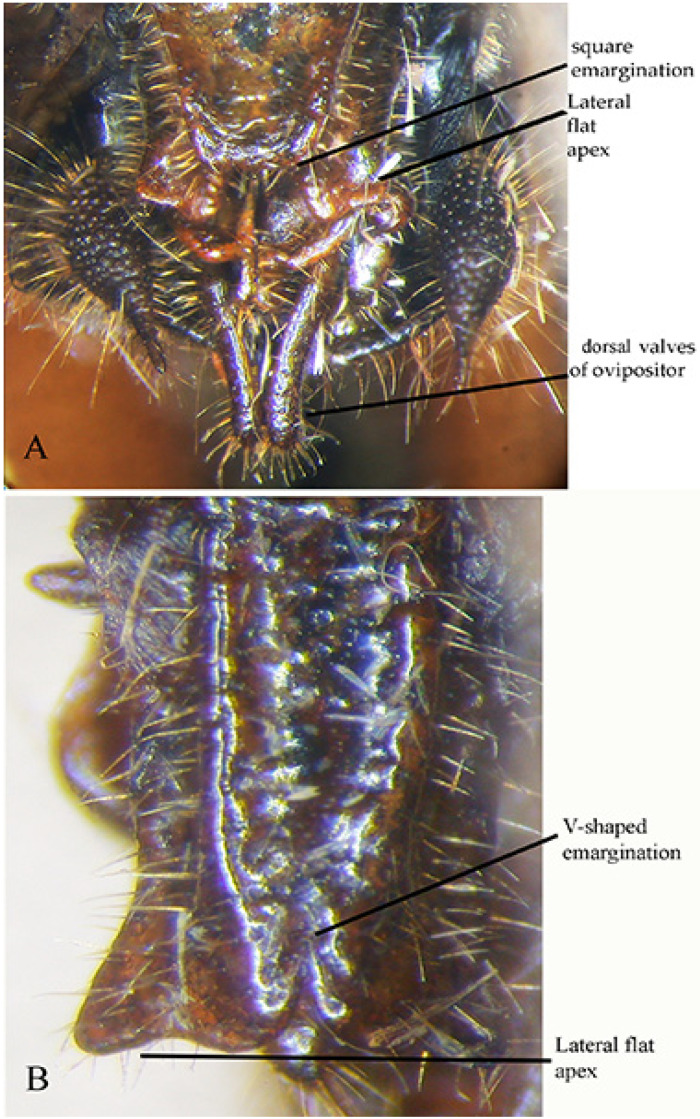
(**A**). *Phlaurocentrum dentatum* Massa, 2023, female subgenital plate; (**B**). *Phlaurocentrum mecopodoides* Karsch, 1892, female subgenital plate.

*Phlaurocentrum mecopodoides* Karsch, 1892 (Figure 17B)Karsch, 1892[1891], Berlin Ent. Z. 36(2): 321, 322; type locality: Barombi Station (Cameroon); depository: MfN, Berlin (♂ holotypus).Material examined. Republic of Congo, Odzala-Kokoua NP, Mbomo Headquarters 28.IX–1.X.2024, MV light trap, M. Bashford, G. László, A. Volynkin (1♂); Mboko 18–21.XI.2024, MV light trap, M. Bashford, I. Elliott, A. Kirk-Spriggs (1♀) (ANHRT).Total. September: 1♂; November: 1♀.Remarks. The characteristic subgenital plate of the female is shown in Figure 17B; central emargination is v-shaped, and the margins are flat and laterally expanded.Distribution. Uncommon in central tropical Africa.

*Phlaurocentrum latevittatum* Karsch, 1889Karsch, 1889[1888], Berlin Ent. Z. 32: 446; type locality: Kuako to Kimpoki (Democratic Republic of Congo); depository: MfN, Berlin (♀ holotypus).Material examined. Republic of Congo, Odzala-Kokoua NP, Mbomo Headquarters 28.IX–1.X.2024, MV light trap, M. Bashford, G. László, A. Volynkin (1♂) (ANHRT).Distribution. Democratic Republic of Congo, Cameroon, Equatorial Guinea, Central African Republic, Gabon, and Republic of Congo (Nouabalé-Ndoki National Park) [4].

*Buettneria maculiceps* Karsch, 1889Karsch, 1889[1888], Berlin Ent. Z. 32: 445; type locality: between Kwako and Kimpokon (Democratic Republic of Congo); depository: MfN, Berlin (♀ holotype).Material examined. Republic of Congo, Odzala-Kokoua NP, Mbomo Headquarters 28.IX–1.X.2024, MV light trap, M. Bashford, G. László, A. Volynkin (1♀); Imbalanga camp 14–18.XI.2024, MV light trap, M. Bashford, I. Elliott, A. Kirk-Spriggs (1♂) (ANHRT).Total. September: 1♀; November: 1♂.Distribution. Uncommon species, previously known from Democratic Republic of Congo, Cameroon, and Nouabalé-Ndoki National Park in the Republic of Congo [4].

*Leiodontocercus vicii* Massa, 2020Massa, 2020, ZooKeys 951: 52; type locality: Dzanga-Ndoki National Park (Central African Republic); depository: BMPC, Palermo (♂ holotypus).Material examined. Republic of Congo, Odzala-Kokoua NP, Imbalanga Camp 5–9.IV.2024, MV Light trap, M. Bashford, G. László, M. Talani, S. Yaba Ngouma (5♂); Imbalanga Camp 5–11.IV.2024, Actinic Light trap, M. Bashford, G. László, M. Talani, S. Yaba Ngouma (2♀); Imbalanga Camp 7–8.IV.2024, Lepiled light trap, M. Bashford, G. László, M. Talani, S. Yaba Ngouma (1♂); Kilo Forest (500 m), 00°31′37″ N, 14°52′20″ E, 23–24.IX.2024, Lepiled Light trap, M. Bashford, G. László, M. Talani, A. Volynkin, S. Yaba Ngouma (1♂, 1♀); Bongassou Forest near Lubo (400 m) 00°32′50″ N, 14°51′47″ E, 23–26.IX.2024, Actinic Light trap, M. Bashford, G. László, M. Talani, A. Volynkin, S. Yaba Ngouma (2♂); Mbomo Headquarters 28.IX–1.X.2024, Lepiled light trap, M. Bashford, G. László, M. Talani, S. Yaba Ngouma (2♂); Mbomo Headquarters 28.IX–1.X.2024, MV light trap, M. Bashford, G. László, A. Volynkin (9♂, 5♀); Imbalanga camp 14–18.XI.2024, MV light trap, M. Bashford, I. Elliott, A. Kirk-Spriggs (1♂) (ANHRT).Total. April: 6♂, 2♀; September: 14♂, 6♀; November: 1♂.Distribution. Previously known from Dzanga-Ndoki National Park (Central African Republic); the present localitis are about 200–220 km from the type locality.

*Leiodontocercus condylus* Ragge, 1962Ragge, 1962. Bull. Br. Mus. (Nat. Hist.) Ent. 13: 15; type locality: Kibali-Ituri, Yindi (Democratic Republic of Congo); depository: MRAC (♂ holotype).Material examined. Republic of Congo, Odzala-Kokoua NP, Kokoua base 5–13.IX.2024, MV light trap, M. Bashford, G. László, M. Talani, A. Volynkin (2♂, 4♀); Lobo Res. Camp 21–22.IX.2024, Actinic light trap, M. Bashford, G. László, M. Talani, A. Volynkin (1♂) (ANHRT).Total. September: 3♂, 4♀.Distribution. Previously known from Democratic Republic of Congo and Central African Republic.

Tribe Holochlorini Brunner von Wattenwyl, 1878*Arantia (Arantia) quinquemaculata* Hemp & Massa, 2017Hemp & Massa, 2017, Zootaxa 4362(4): 466; type locality: Dzanga-Ndoki National Park (Central African Republic); depository: MSNG (♂ holotypus).Material examined. Republic of Congo, Odzala-Kokoua NP, Imbalanga Camp 5–9.IV.2024, MV Light trap, M. Bashford, G. László, M. Talani, S. Yaba Ngouma (1♂); Kokoua base 5–13.IX.2024, MV light trap, M. Bashford, G. László, M. Talani, A. Volynkin (11♂) (ANHRT).Total. April: 1♂; September: 11♂.Distribution. It is fairly widespread in west-central Africa.

*Arantia (Arantia) simplicinervis* Karsch, 1889Karsch, 1889, Berlin Ent. Z., 32: 433, 434, 438; type locality: Chinchoxo and Kuako (Cameroon); depository: MfN (♂ syntypes).Material examined. Republic of Congo, Odzala-Kokoua NP, Imbalanga bai 6–9.IV.2024, Actinic Light trap, M. Bashford, G. László, M. Talani, S. Yaba Ngouma (1♂); Kokoua base 5–13.IX.2024, MV light trap, M. Bashford, G. László, M. Talani, A. Volynkin (1♂); Lobo Res. Camp 20–27.IX.2024, MV light trap, M. Bashford, G. László, A. Volynkin (1♂); Mbomo Headquarters 28.IX–1.X.2024, Lepiled light trap, M. Bashford, G. László, M. Talani, S. Yaba Ngouma (2♂); Imbalanga camp 14–18.XI.2024, MV light trap, M. Bashford, I. Elliott, A. Kirk-Spriggs (1♂) (ANHRT).Total. April: 1♂; September: 4♂; November: 1♂.Distribution. More common and more widespread than *A. quinquemaculata* in west-central tropical Africa.

*Arantia (Arantia) marginata* Massa, 2021Massa, 2021, Ann. Soc. ent. Fr. 57(1): 59; type locality: Mts. de Cristal (Gabon); depository: ANHRT, Hereford (♂ holotypus).Material examined. Republic of Congo, Odzala-Kokoua NP, Imbalanga Camp 5–9.IV.2024, MV Light trap, M. Bashford, G. László, M. Talani, S. Yaba Ngouma (16♂, 1♀); Imbalanga Camp 7–8.IV.2024, Lepiled light trap, M. Bashford, G. László, M. Talani, S. Yaba Ngouma (1♂); Moba 11–12.IV.2024, Actinic Light trap, M. Bashford, G. László, M. Talani, S. Yaba Ngouma (1♂); Mbomo Headquarters 19–22.IV.2024, MV Light trap, M. Bashford, G. László, M. Talani, S. Yaba Ngouma (1♂); Mbomo Headquarters 5–11.IV.2024, Actinic Light trap, M. Bashford, G. László, M. Talani, S. Yaba Ngouma (4♂); Lobo Res. Camp 16–17.IV.2024, Lepiled light trap, M. Bashford, G. László, M. Talani, S. Yaba Ngouma (1♂); Kokoua base 5–13.IX.2024, MV light trap, M. Bashford, G. Lászlo, M. Talani, A. Volynkin (51♂, 2♀); Imbalanga Camp 13–20.IX.2024, MV Light trap, M. Bashford, G. László, M. Talani, A. Volynkin (2♂); Lobo Res. Camp 20–27.IX.2024, MV light trap, M. Bashford, G. László, A. Volynkin (4♂); Lekoli river near Mboko 21–23.IX.2024, Lepiled Light trap, M. Bashford, G. László, M. Talani, A. Volynkin, S. Yaba Ngouma (1♂); Kilo Forest 23–24.IX.2024, Lepiled Light trap, M. Bashford, G. László, M. Talani, A. Volynkin, S. Yaba Ngouma (1♂); Mbomo Headquarters 28.IX–1.X.2024, MV light trap, M. Bashford, G. László, A. Volynkin (7♂); Mbomo Headquarters 28.IX–1.X.2024, Lepiled light trap, M. Bashford, G. László, M. Talani, A. Volynkin, S. Yaba Ngouma (2♂); Camp Imbalanga 14–18.XI.2024, MV light trap, M. Bashford, I. Elliott, A. Kirk-Spriggs (2♂); Lobo Res. Camp 22–30.XI.2024, MV light trap, M. Bashford, I. Elliott, A. Kirk-Spriggs (1♂); Lobo Res. Camp 23–25.XI.2024, Lepiled light trap, M. Bashford, I. Elliott, A. Kirk-Spriggs (1♂) (ANHRT).Total. April: 24♂, 1♀; September: 68♂, 2♀; November: 4♂.Phenology. Twenty-two males (32.3%) collected in September and one in November had a spermatophore.

Distribution. It has so far been confused with *A. rectifolia*, and thus it is difficult to establish its distribution, which, however, covers west-central tropical Africa.

*Arantia (Euarantia) rectifolia* Brunner von Wattenwyl, 1878Brunner von Wattenwyl, 1878, Monographie der Phaneropteriden 137; type locality: Bioko, Fernando Poo; depository: NMW, Vienna (♂ holotypus).Material examined. Republic of Congo, Odzala-Kokoua NP, Imbalanga Camp 5–9.IV.2024, MV Light trap, M. Bashford, G. László, M. Talani, S. Yaba Ngouma (1♀); Imbalanga Camp 4–5.IX.2024, Actinic Light trap, M. Bashford, G. László, M. Talani, A. Volynkin (1♂); Kokoua base 5–13.IX.2024, MV light trap, M. Bashford, G. Lászlo, M. Talani, A. Volynkin (5♂); Lobo Res. Camp 20–27.IX.2024, MV light trap, M. Bashford, G. László, A. Volynkin (1♀); Mbomo Headquarters 28.IX–1.X.2024, MV light trap, M. Bashford, G. László, A. Volynkin (1♀); Mbomo Headquarters 28.IX–1.X.2024, Lepiled light trap, M. Bashford, G. László, M. Talani, A. Volynkin, S. Yaba Ngouma (1♂, 2♀); Camp Imbalanga 14–18.XI.2024, MV light trap, M. Bashford, I. Elliott, A. Kirk-Spriggs (1♀) (ANHRT).Total. April: 1♀; September: 7♂, 4♀; November: 1♀.Distribution. Common and widespread in west-central tropical Africa.

*Arantia* (*Euarantia*) *retinervis* Karsch, 1889Karsch, 1889, Berlin Ent. Z., 32: 433, 437; type locality: Cameroon and Sierra Leone; depository: MfN (♂ syntypi).Material examined. Republic of Congo, Odzala-Kokoua NP, Imbalanga Camp 5–9.IV.2024, MV Light trap, M. Bashford, G. László, M. Talani, S. Yaba Ngouma (1♂); Imbalanga Camp 5–11.IV.2024, Actinic Light trap, M. Bashford, G. László, M. Talani, S. Yaba Ngouma (1♂); Moba 11–12.IV.2024, Actinic Light trap, M. Bashford, G. László, M. Talani, S. Yaba Ngouma (1♂); Kokoua base 5–13.IX.2024, MV light trap, M. Bashford, G. Lászlo, M. Talani, A. Volynkin (10♂); Kokoua base 6–13.IX.2024, Actinic light trap, M. Bashford, G. Lászlo, M. Talani, A. Volynkin, S. Yaba Ngouma (1♂); Kokoua base 11–13.IX.2024, Lepiled light trap, M. Bashford, G. Lászlo, M. Talani, A. Volynkin, S. Yaba Ngouma (1♂); Imbalanga Camp 13–20.IX.2024, MV Light trap, M. Bashford, G. László, M. Talani, A. Volynkin (1♂); Bongassou Forest near Lubo 23–26.IX.2024, Actinic Light trap, M. Bashford, G. László, M. Talani, A. Volynkin, S. Yaba Ngouma (1♂); Mbomo Headquarters 28–30.IX.2024, Actinic light trap, M. Bashford, G. László, M. Talani, A. Volynkin, S. Yaba Ngouma (2♂); Mbomo Headquarters 28.IX–1.X.2024, MV light trap, M. Bashford, G. László, A. Volynkin (1♂); Camp Imbalanga 14–18.XI.2024, MV light trap, M. Bashford, I. Elliott, A. Kirk-Spriggs (1♂) (ANHRT).Total. April: 3♂; September: 17♂; November: 1♂.Distribution. Uncommon in west-central tropical Africa.

*Arantia (Euarantia) regina* Karsch, 1889Karsch, 1889, Berlin Ent. Z., 32: 433, 434; type locality: Sibange Farm (Gabon); depository: MfN (♀ holotypus).Material examined. Republic of Congo, Odzala-Kokoua NP, Lobo Res. Camp 13–18.IV.2024, MV Light trap, M. Bashford, G. László, M. Talani, S. Yaba Ngouma (1♀); Kokoua base 5–13.IX.2024, MV light trap, M. Bashford, G. Lászlo, M. Talani, A. Volynkin (5♂, 2♀); Mbomo Headquarters 28.IX–1.X.2024, MV light trap, M. Bashford, G. László, A. Volynkin (2♂, 3♀) (ANHRT).Total. April: 1♀; September: 7♂, 5♀.Phenology. Four males (57.1%) collected in September had a spermatophore.Distribution. Common and widespread in west-central tropical Africa.

*Arantia* (*Euarantia*) *excelsior* Karsch, 1889Karsch, 1889[1888], Berlin Ent. Z. 32: 434, 43; type locality: Sierra Leone; depository: MZPW (♂ holotypus).Material examined. Republic of Congo, Odzala-Kokoua NP, Kokoua base 5–13.IX.2024, MV light trap, M. Bashford, G. Lászlo, M. Talani, A. Volynkin (1♂); Mbomo Headquarters 28–30.IX.2024, Actinic light trap, M. Bashford, G. László, M. Talani, A. Volynkin, S. Yaba Ngouma (1♂) (ANHRT).Phenology. One of the two males collected in September had a spermatophore.Distribution. Uncommon in west-central tropical Africa.

*Arantia* (*Euarantia*) *syssamagalei* Massa & Annoyer, 2020Massa & Annoyer, 2020, In: Massa, Annoyer, Perez, Danflous & Duvot, Orthoptera Tettigoniidae (Conocephalinae, Hexacentrinae, Phaneropterinae, Mecopodinae, Hetrodinae) from some protected areas of Central African Republic. Zootaxa 4780(3): 423; type locality: Dzanga-Sangha Special Reserve (Central African Republic); depository: MCSN, Genoa (♂ holotypus).Material examined. Republic of Congo, Odzala-Kokoua NP, Kokoua base 5–13.IX.2024, MV light trap, M. Bashford, G. Lászlo, M. Talani, A. Volynkin (8♂) (ANHRT).Distribution. Previously known from Central African Republic and Republic of Congo [4].

*Arantia* (*Euarantia*) *congensis* Griffini, 1908Griffini, 1908, Mem. Soc. entom. Belgique, Bruxelles 15:,214; type locality: Bussira (Democratic Republic of Congo); depository: RBINS, Bruxelles (♂ syntypes).Material examined. Republic of Congo, Odzala-Kokoua NP, Camp Imbalanga 13–19.IX.2024, Actinic Light trap, M. Bashford, G. László, A. Volynkin, S. Yaba Ngouma (1♂); Imbalanga Camp 13–20.IX.2024, MV Light trap, M. Bashford, G. László, M. Talani, A. Volynkin (2♂); Bongassou Forest near Lubo 23–26.IX.2024, Actinic Light trap, M. Bashford, G. László, M. Talani, A. Volynkin, S. Yaba Ngouma (1♂); Mbomo Headquarters 28–30.IX.2024, Actinic light trap, M. Bashford, G. László, M. Talani, A. Volynkin, S. Yaba Ngouma (1♂); Mbomo Headquarters 28.IX–1.X.2024, MV light trap, M. Bashford, G. László, A. Volynkin (3♂) (ANHRT).Total. September: 8♂.Distribution. Uncommon in central tropical Africa.

*Arantia* (*Euarantia*) *incerata* Karsch, 1893Karsch, 1893, Berlin Ent. Z. 38: 128; type locality: Sierra Leone; depository: MfN, Berlin (♂ holotype).Material examined. Republic of Congo, Odzala-Kokoua NP, Mbomo Headquarters 28.IX–1.X.2024, MV light trap, M. Bashford, G. László, A. Volynkin (1♂) (ANHRT).Total. September: 1♂.Distribution. Uncommon species, previously known from west-central tropical Africa (Sierra Leone, Togo, Democratic Republic Congo and Cameroon) [27].

*Dapanera genuteres* Karsch, 1889Karsch, 1889, Berlin Ent. Z. 32: 441; type locality: Accra (Ghana); depository: MfN (♂ syntypes).Material examined. Republic of Congo, Odzala-Kokoua NP, Imbalanga Camp 5–9.IV.2024, MV Light trap, M. Bashford, G. László, M. Talani, S. Yaba Ngouma (3♂, 5♀); Mbomo Headquarters 5–11.IV.2024, Actinic Light trap, M. Bashford, G. László, M. Talani, S. Yaba Ngouma (1♂); Imbalanga Camp 5–11.IV.2024, Actinic Light trap, M. Bashford, G. László, M. Talani, S. Yaba Ngouma (3♂); Kokoua base 5–13.IX.2024, MV light trap, M. Bashford, G. Lászlo, M. Talani, A. Volynkin (5♂, 6♀); Kokoua Base 6–13.IX.2024, Actinic Light trap, M. Bashford, G. László, M. Talani, A. Volynkin, S. Yaba Ngouma (1♀); Lobo Res. Camp 20–27.IX.2024, MV light trap, M. Bashford, G. László, A. Volynkin (2♂, 1♀); Lekoli river near Mboko 21–23.IX.2024, Lepiled Light trap, M. Bashford, G. László, M. Talani, A. Volynkin, S. Yaba Ngouma (1♂); Lobo Res. Camp 25–26.IX.2024, Lepiled light trap, M. Bashford, G. László, A. Volynkin, S. Yaba Ngouma (1♂); Mbomo Headquarters 28.IX–1.X.2024, MV light trap, M. Bashford, G. László, A. Volynkin (1♂); Mboko 18–21.XI.2024 MV light trap, M. Bashford, I. Elliott, A. Kirk-Spriggs (1♂); Lobo Res. Camp 22–30.XI.2024, MV light trap, M. Bashford, I. Elliott, A. Kirk-Spriggs (5♂) (ANHRT).Total. April: 7♂, 5♀; September: 10♂, 8♀; November: 6♂.Phenology. One male collected in September and one in November had a spermatophore.Distribution. Widespread in west-central tropical Africa.

*Dapanera irregularis* Karsch, 1890Karsch, 1890, Entom. Nachricht. 16: 258; type locality: Kribi (Cameroon); depository: MfN (♂,♀ syntypi).Material examined. Republic of Congo, Odzala-Kokoua NP, Lobo Res. Camp 16–17.IV.2024, Lepiled light trap, M. Bashford, G. László, M. Talani, S. Yaba Ngouma (1♂); Imbalanga Camp 5–11.IV.2024, Actinic Light trap, M. Bashford, G. László, M. Talani, S. Yaba Ngouma (3♂); Imbalanga Camp 7–8.IV.2024, Lepiled light trap, M. Bashford, G. László, M. Talani, S. Yaba Ngouma (1♂); Moba 11–12.IV.2024, Actinic Light trap, M. Bashford, G. László, M. Talani, S. Yaba Ngouma (4♂); Kokoua base 5–13.IX.2024, MV light trap, M. Bashford, G. Lászlo, M. Talani, A. Volynkin (8♂, 14♀); Imbalanga Camp 13–20.IX.2024, MV Light trap, M. Bashford, G. László, M. Talani, A. Volynkin (1♂); Moba 19–20.IX.2024, Lepiled light trap, M. Bashford, G. Lászlo, M. Talani, A. Volynkin (1♀); Mbomo Headquarters 28–30.IX.2024, Actinic light trap, M. Bashford, G. László, M. Talani, A. Volynkin, S. Yaba Ngouma (1♂); Mbomo Headquarters 28.IX–1.X.2024, MV light trap, M. Bashford, G. László, A. Volynkin (4♂); Mbomo Headquarters 28.IX–1.X.2024, Lepiled light trap, M. Bashford, G. László, M. Talani, A. Volynkin, S. Yaba Ngouma (1♂, 2♀) (ANHRT).Total. April: 9♂; September: 15♂, 17♀.Phenology. One male collected in September had a spermatophore.Distribution. Widespread in west-central tropical Africa.

*Dapanera occulta* Massa, 2015Massa, 2015, ZooKeys 524: 34; type locality: Dzanga-Ndoki National Park (Central African Republic); depository: MCSN, Genoa (♂ holotypus).Material examined. Republic of Congo, Odzala-Kokoua NP, Imbalanga Camp 5–9.IV.2024, MV Light trap, M. Bashford, G. László, M. Talani, S. Yaba Ngouma (3♂); Mbomo Headquarters 19–22.IV.2024, MV Light trap, M. Bashford, G. László, M. Talani, S. Yaba Ngouma (4♂); Imbalanga Camp 5–11.IV.2024, Actinic Light trap, M. Bashford, G. László, M. Talani, S. Yaba Ngouma (1♂); Kokoua base 5–13.IX.2024, MV light trap, M. Bashford, G. László, M. Talani, A. Volynkin (3♂, 1♀); Kokoua base 11–13.IX.2024, Lepiled light trap, M. Bashford, G. Lászlo, M. Talani, A. Volynkin, S. Yaba Ngouma (1♂); Imbalanga Camp 13–20.IX.2024, MV Light trap, M. Bashford, G. László, M. Talani, A. Volynkin (1♂); Camp Imbalanga (540 m) 15–18.IX.2024, Lepiled Light trap, M. Bashford, G. László, A. Volynkin, S. Yaba Ngouma (1♂) (ANHRT).Total. April: 8♂; September: 6♂, 1♀.Phenology. One male collected in September had a spermatophore.Distribution. Presently known from Central African Republic and Republic of Congo.

*Dapanera brevistylata* Massa, 2020Massa, 2020, In: Massa, Annoyer, Perez, Danflous & Duvot, Orthoptera Tettigoniidae (Conocephalinae, Hexacentrinae, Phaneropterinae, Mecopodinae, Hetrodinae) from some protected areas of Central African Republic. Zootaxa 4780(3): 425; type locality: Dzanga-Ndoki National Park (Central African Republic); depository: MCSN, Genoa (♂ holotypus).Material examined. Republic of Congo, Odzala-Kokoua NP, Kokoua base 5–13.IX.2024, MV light trap, M. Bashford, G. László, M. Talani, A. Volynkin (3♂); Kokoua Base 6–13.IX.2024, Actinic Light trap, M. Bashford, G. László, M. Talani, A. Volynkin, S. Yaba Ngouma (1♂); Kilo Forest 23–24.IX.2024, Lepiled Light trap, M. Bashford, G. László, M. Talani, A. Volynkin, S. Yaba Ngouma (1♂); Mbomo Headquarters 28–30.IX.2024, Actinic light trap, M. Bashford, G. László, M. Talani, A. Volynkin, S. Yaba Ngouma (5♂); Mbomo Headquarters 28.IX–1.X.2024, MV light trap, M. Bashford, G. László, A. Volynkin (2♂); Mbomo Headquarters 28.IX–1.X.2024, MV light trap, M. Bashford, G. László, A. Volynkin (2♂, 1♀); Mbomo Headquarters 28.IX–1.X.2024, Lepiled light trap, M. Bashford, G. László, M. Talani, A. Volynkin, S. Yaba Ngouma (3♂) (ANHRT).Total. April: 3♂; September: 14♂, 1♀.Phenology. Five males (35.7%) collected in September had a spermatophore.Distribution. Previously known from Central African Republic, Gabon, Liberia, and Nouabalé-Ndoki National Park in the Republic of Congo [4].

Tribe Plangiopsidini Cadeña-Castaneda, 2015*Plangiopsis adeps* Karsch, 1896Karsch, 1896, Stett. Entomol. Z. 57: 338; type locality: Cameroon, Lolodorf; depository: MfN, Berlin (♀ syntypus).Material examined. Republic of Congo, Odzala-Kokoua NP, Imbalanga Camp 5–9.IV.2024, MV Light trap, M. Bashford, G. László, M. Talani, S. Yaba Ngouma (2♂, 2♀); Imbalanga Camp 5–11.IV.2024, Actinic Light trap, M. Bashford, G. László, M. Talani, S. Yaba Ngouma (2♂); Kokoua base 5–13.IX.2024, MV light trap, M. Bashford, G. Lászlo, M. Talani, A. Volynkin (9♂, 11♀); Lobo Res. Camp 20–27.IX.2024, MV light trap, M. Bashford, G. László, A. Volynkin (3♀); Camp Imbalanga 14–15 and 17–18.XI.2024, Lepiled light trap, M. Bashford, I. Elliott, A. Kirk-Spriggs (1♀) (ANHRT).Total. April: 4♂, 2♀; September: 9♂, 14♀; November: 1♀.Distribution. Widespread from western to central and southern (Zambia) tropical Africa.

*Plangiopsis nouabalensis* Massa, 2023Massa, 2023, Zootaxa, 5331 (1): 16; type locality: Nouabalé-Ndoki National Park (Republic of Congo); depository: ANHRT (♂ holotypus).Material examined. Republic of Congo, Odzala-Kokoua NP, Imbalanga Camp 5–9.IV.2024, MV Light trap, M. Bashford, G. László, M. Talani, S. Yaba Ngouma (2♂, 2♀); Imbalanga Camp 5–11.IV.2024, Actinic Light trap, M. Bashford, G. László, M. Talani, S. Yaba Ngouma (1♂); Moba 11–12.IV.2024, Actinic Light trap, M. Bashford, G. László, M. Talani, S. Yaba Ngouma (2♂); Kokoua base 5–13.IX.2024, MV light trap, M. Bashford, G. Lászlo, M. Talani, A. Volynkin (2♀); Mbomo Headquarters 28–30.IX.2024, Actinic light trap, M. Bashford, G. László, M. Talani, A. Volynkin, S. Yaba Ngouma (1♂); Camp Imbalanga 14–18.XI.2024, MV light trap, M. Bashford, I. Elliott, A. Kirk-Spriggs (1♀) (ANHRT).Total. April: 5♂, 2♀; September: 1♂, 2♀; November: 1♀.Distribution. Recently described from Nouabalé-Ndoki National Park, now also recorded in Odzala-Kokoua National Park, Republic of Congo.

*Plangiopsis foraminata* Karsch, 1892Karsch, 1892, Berlin Ent. Z. 36(2): 324; type locality: Cameroon, Barombi Station; depository: MfN, Berlin (♂ holotypus).Material examined. Republic of Congo, Odzala-Kokoua NP, Imbalanga Camp 5–9.IV.2024, MV Light trap, M. Bashford, G. László, M. Talani, S. Yaba Ngouma (1♂, 3♀); Kokoua base 5–13.IX.2024, MV light trap, M. Bashford, G. László, M. Talani, A. Volynkin (2♂, 1♀); Camp Imbalanga 14–18.XI.2024, MV light trap, M. Bashford, I. Elliott, A. Kirk-Spriggs (1♀); Mboko 18–21.XI.2024 MV light trap, M. Bashford, I. Elliott, A. Kirk-Spriggs (1♀) (ANHRT).Total. April: 1♂, 3♀; September: 2♂, 1♀; November: 2♀.Distribution. Common in west-central tropical Africa.

*Plangiopsis semiconchata* Karsch, 1889Karsch, 1889, Berlin Ent. Z. 32: 460; type locality: Barombi Station (Cameroon); depository: MfN (♀ holotypus).Material examined. Republic of Congo, Odzala-Kokoua NP, Mbomo Headquarters 19–22.IV.2024, MV Light trap, M. Bashford, G. László, M. Talani, S. Yaba Ngouma (5♂); Lobo Res. Camp 13–18.IV.2024, MV Light trap, M. Bashford, G. László, M. Talani, S. Yaba Ngouma (1♂); Kokoua base 5–13.IX.2024, MV light trap, M. Bashford, G. László, M. Talani, A. Volynkin (10♂); Imbalanga Camp 13–20.IX.2024, MV Light trap, M. Bashford, G. László, M. Talani, A. Volynkin (2♂); Kilo Forest 23–24.IX.2024, Lepiled Light trap, M. Bashford, G. László, M. Talani, A. Volynkin, S. Yaba Ngouma (1♂); Mbomo Headquarters 28–30.IX.2024, Actinic light trap, M. Bashford, G. László, M. Talani, A. Volynkin, S. Yaba Ngouma (1♂); Mbomo Headquarters 28.IX–1.X.2024, MV light trap, M. Bashford, G. László, A. Volynkin (3♂); Mbomo Headquarters 28.IX–1.X.2024, Lepiled light trap, M. Bashford, G. László, M. Talani, A. Volynkin, S. Yaba Ngouma (2♂) (ANHRT).Total. April: 6♂; September: 19♂.Phenology. Three males (15.8%) collected in September had a spermatophore.Distribution. Common in west-central tropical Africa.

Tribe Morgeniini Karsch, 1890*Morgenia plurimaculata* Massa & Moulin, 2018Massa & Moulin, 2018, in Massa, Heller, Warchalowska-Sliwa & Moulin, Dtsch. Entomol. Z. 65(2): 165; type locality: Dzanga-Ndoki National Park (Central African Republic); depository: MCSN, Genoa (♂ holotypus).Material examined. Republic of Congo, Odzala-Kokoua NP, Imbalanga Camp 5–9.IV.2024, MV Light trap, M. Bashford, G. László, M. Talani, S. Yaba Ngouma (3♂, 1♀); Moba 11–12.IV.2024, Actinic light trap, M. Bashford, G. László, M. Talani, S. Yaba Ngouma (1♂); Mbomo Headquarters 19–22.IV.2024, MV Light trap, M. Bashford, G. László, M. Talani, S. Yaba Ngouma (1♂, 1♀); Lobo Res. Camp 16–17.IV.2024, Lepiled light trap, M. Bashford, G. László, M. Talani, S. Yaba Ngouma (2♂); Lobo Res. Camp 13–18.IV.2024, MV Light trap, M. Bashford, G. László, M. Talani, S. Yaba Ngouma (1♂); Kokoua base 5–13.IX.2024, MV light trap, M. Bashford, G. László, M. Talani, A. Volynkin (34♂, 1♀); Kokoua base 11–13.IX.2024, Lepiled light trap, M. Bashford, G. Lászlo, M. Talani, A. Volynkin, S. Yaba Ngouma (1♂); Camp Imbalanga 13–20.IX.2024, MV Light trap, M. Bashford, G. László, A. Volynkin, S. Yaba Ngouma (1♀); Lobo Res. Camp 20–27.IX.2024, MV light trap, M. Bashford, G. László, A. Volynkin (1♂); Kilo Forest 23–24.IX.2024, Lepiled Light trap, M. Bashford, G. László, M. Talani, A. Volynkin, S. Yaba Ngouma (1♂); Mbomo Headquarters 28.IX–1.X.2024, MV light trap, M. Bashford, G. László, A. Volynkin (4♀); Mboko 18–21.XI.2024 MV light trap, M. Bashford, I. Elliott, A. Kirk-Spriggs (2♂) (ANHRT).Total. April: 8♂, 2♀; September: 37♂, 6♀; November: 2♂.Phenology. One male collected in September had a spermatophore.Distribution. Known from Central African Republic, Gabon, Liberia, Guinea, Ghana, Ivory Coast, and Republic of Congo.

*Morgenia melica* Karsch, 1893Karsch, 1893, Entom. Nachricht. 19(13): 196; type locality: Victoria (Cameroon); depository: MfN (♂ holotypus).Material examined. Republic of Congo, Odzala-Kokoua NP, Imbalanga Camp 5–9.IV.2024, MV Light trap, M. Bashford, G. László, M. Talani, S. Yaba Ngouma (8♂); Imbalanga bai 6–9.IV.2024, Actinic Light trap, M. Bashford, G. László, M. Talani, S. Yaba Ngouma (1♂, 1♀); Mbomo Headquarters 19–22.IV.2024, MV Light trap, M. Bashford, G. László, M. Talani, S. Yaba Ngouma (1♂); Mbomo Headquarters 5–11.IV.2024, Actinic Light trap, M. Bashford, G. László, M. Talani, S. Yaba Ngouma (1♂); Kokoua base 5–13.IX.2024, MV light trap, M. Bashford, G. Lászlo, M. Talani, A. Volynkin (12♂); Kokoua Base 6–13.IX.2024, Actinic Light trap, M. Bashford, G. László, M. Talani, A. Volynkin, S. Yaba Ngouma (1♂); Imbalanga Camp 13–20.IX.2024, MV Light trap, M. Bashford, G. László, M. Talani, A. Volynkin (2♂); Lobo Res. Camp 20–27.IX.2024, MV light trap, M. Bashford, G. László, A. Volynkin (1♂); Bongassou Forest near Lubo 23–26.IX.2024, Actinic Light trap, M. Bashford, G. László, M. Talani, A. Volynkin, S. Yaba Ngouma (2♂); Lobo Res. Camp 25–26.IX.2024, Lepiled light trap, M. Bashford, G. László, A. Volynkin, S. Yaba Ngouma (2♂); Lobo Res. Camp 25–26.IX.2024, Lepiled Light trap, M. Bashford, G. László, M. Talani, A. Volynkin, S. Yaba Ngouma (1♂); Mbomo Headquarters 28.IX–1.X.2024, MV light trap, M. Bashford, G. László, A. Volynkin (2♂); Mbomo Headquarters 28.IX–1.X.2024, Lepiled light trap, M. Bashford, G. László, M. Talani, A. Volynkin, S. Yaba Ngouma (2♂); Mboko 18–21.XI.2024 MV light trap, M. Bashford, I. Elliott, A. Kirk-Spriggs (2♂); Lekoli river, near Mboko 23–24.XI.2024, Lepiled light trap M. Bashford, I. Elliott, A. Kirk-Spriggs (1♂); Lobo Res. Camp 23–27.XI.2024, Actinic light trap M. Bashford, I. Elliott, A. Kirk-Spriggs (1♂); Lobo Res. Camp 23–25.XI.2024, Lepiled light trap, M. Bashford, I. Elliott, A. Kirk-Spriggs (1♂) (ANHRT).Total. April: 11♂, 1♀; September: 24♂; November: 5♂.Distribution. Common and widespread in west-central tropical Africa.

*Morgenia hamuligera* Karsch, 1890Karsch, 1890, Entom. Nachricht. 16: 263; type locality: Kribi (Cameroon); depository: MfN, Berlin (♂ holotype).Material examined. Republic of Congo, Odzala-Kokoua NP, Kokoua base 11–13.IX.2024, Lepiled light trap, M. Bashford, G. Lászlo, M. Talani, A. Volynkin, S. Yaba Ngouma (1♀); Moba 19–20.IX.2024, Lepiled light trap, M. Bashford, G. Lászlo, M. Talani, A. Volynkin (1♂); Lekoli river near Mboko 21–23.IX.2024, Lepiled Light trap, M. Bashford, G. László, M. Talani, A. Volynkin, S. Yaba Ngouma (3♂); Lobo Res. Camp 20–27.IX.2024, MV light trap, M. Bashford, G. László, A. Volynkin (2♂); Lobo Res. Camp 25–26.IX.2024, Lepiled light trap, M. Bashford, G. László, A. Volynkin, S. Yaba Ngouma (1♂); Mbomo Headquarters 28.IX–1.X.2024, general collection, M. Bashford, G. László, M. Talani, A. Volynkin, S. Yaba Ngouma (2♂); Mboko 18–21.XI.2024 MV light trap, M. Bashford, I. Elliott, A. Kirk-Spriggs (5♂) (ANHRT).Total. September: 9♂, 1♀; November: 5♂.Distribution. Widespread in west-central tropical Africa (presently known from Cameroon, Equatorial Guinea, Democratic Republic of Congo, Central African Republic, Gabon, Ivory Coast, Liberia, and Togo: Massa [3]).Remarks. One male collected at Mbomo Headquarters and two collected at Lobo Res. camp have some small black dots on their tegmina, similar to those of *M. plurimaculata*, but other characteristics are those of *M. hamuligera*.

*Morgenia rubricornis* Sjöstedt, 1913Sjöstedt, 1913, Ark. Zool. 8(6): 4; type locality: Mukimbungu (Democratic Republic of Congo); depository: NHRS Stockholm (♂ holotypus).Material examined. Republic of Congo, Odzala-Kokoua NP, Imbalanga Camp 5–9.IV.2024, MV Light trap, M. Bashford, G. László, M. Talani, S. Yaba Ngouma (1♂); Lobo Res. Camp 16–17.IV.2024, Lepiled light trap, M. Bashford, G. László, M. Talani, S. Yaba Ngouma (3♂); Lobo Res. Camp 13–18.IV.2024, MV Light trap, M. Bashford, G. László, M. Talani, S. Yaba Ngouma (2♂); Moba 11–12.IV.2024, Actinic Light trap, M. Bashford, G. László, M. Talani, S. Yaba Ngouma (1♂); Kokoua base 5–13.IX.2024, MV light trap, M. Bashford, G. Lászlo, M. Talani, A. Volynkin (8♂); Lobo Res. Camp 20–27.IX.2024, MV light trap, M. Bashford, G. László, A. Volynkin (1♂); Lobo Res. Camp 25–26.IX.2024, Lepiled Light trap, M. Bashford, G. László, M. Talani, A. Volynkin, S. Yaba Ngouma (2♂); Lobo Res. Camp 25–26.IX.2024, Lepiled light trap, M. Bashford, G. László, A. Volynkin, S. Yaba Ngouma (2♂); Mbomo Headquarters 28.IX–1.X.2024, MV light trap, M. Bashford, G. László, A. Volynkin (9♂); Mbomo Headquarters 28.IX–1.X.2024, Lepiled light trap, M. Bashford, G. László, M. Talani, A. Volynkin, S. Yaba Ngouma (2♂); Lobo Res. Camp 22–23.XI.2024, MV light trap, M. Bashford, I. Elliott, A. Kirk-Spriggs (1♂); Lobo Res. Camp 23–25.XI.2024, Lepiled light trap, M. Bashford, I. Elliott, A. Kirk-Spriggs (1♂) (ANHRT).Total. April: 7♂; September: 24♂; November: 2♂.Distribution. Even if not common, it is widespread in west-central tropical Africa.Remarks. One ind. collected at Kokoua base (September) and one at Lobo Res. camp (November) have small dots on their tegmina, similar to those of *M. plurimaculata*.

*Morgenia angustipinnata* Massa, 2018Massa, 2018, in Massa, Heller, Warchalowska-Sliwa & Moulin, Dtsch. Entomol. Z. 65(2): 167; type locality: Dzanga-Ndoki National Park, Mboki (Central African Republic); depository: MCSN, Genoa (♂ holotypus).Material examined. Republic of Congo, Odzala-Kokoua NP, Imbalanga Camp 5–9.IV.2024, MV Light trap, M. Bashford, G. László, M. Talani, S. Yaba Ngouma (6♂, 1♀); Mbomo Headquarters 19–22.IV.2024, MV Light trap, M. Bashford, G. László, M. Talani, S. Yaba Ngouma (5♂); Mbomo Headquarters 21–22.IV.2024, MV Light trap, M. Bashford, G. László, M. Talani, S. Yaba Ngouma (1♀); Kokoua base 5–13.IX.2024, MV light trap, M. Bashford, G. László, M. Talani, A. Volynkin (11♂); Lobo Res. Camp 20–27.IX.2024, MV light trap, M. Bashford, G. László, A. Volynkin (1♂); Mboko 18–21.XI.2024, MV light trap, M. Bashford, I. Elliott, A. Kirk-Spriggs (2♂) (ANHRT).Total. April: 11♂, 2♀; September: 12♂, 1♀; November: 2♂.Distribution. Described from Dzanga-Ndoki National Park (Central African Republic), also recorded in Gabon and Republic of Congo (Nouabalé-Ndoki National Park, and presently Odzala-Kokoua National Park).Remarks. Rarely, some specimens lacking a particular spur at the mid-tibia apex.

*Morgenia modulata* Karsch, 1896Karsch, 1896. Stett. Entomol. Z. 57: 340; type locality: Lolodorf (Cameroon); depository: MfN (syntypes ♂ and ♀).Material examined. Republic of Congo, Odzala-Kokoua NP, Kokoua base 5–13.IX.2024, MV light trap, M. Bashford, G. László, M. Talani, A. Volynkin (12♂); Mbomo Headquarters 28.IX–1.X.2024, MV light trap, M. Bashford, G. László, A. Volynkin (3♂); Mbomo Headquarters 28.IX–1.X.2024, Lepiled light trap, M. Bashford, G. László, M. Talani, S. Yaba Ngouma (1♂); Mboko 18–21.XI.2024 MV light trap, M. Bashford, I. Elliott, A. Kirk-Spriggs (1♀); Mboko 19–22.XI.2024, Malaise trap, M. Bashford, I. Elliott, A. Kirk-Spriggs (1♀) (ANHRT).Total. September: 16♂; November: 2♀.Distribution. Uncommon in west-central tropical Africa.

Tribe Poreuomenini Brunner von Wattenwyl, 1878*Poreuomena magnicerca* (Massa, 2013)Massa, 2013, Jour. Orth. Res. 22(2): 142 (*Cestromoecha magnicerca*); type locality: Dzanga-Ndoki National Park (Central African Republic); depository: MCSN, Genoa (♂ holotypus).Material examined. Republic of Congo, Odzala-Kokoua NP, Imbalanga Camp 5–9.IV.2024, MV Light trap, M. Bashford, G. László, M. Talani, S. Yaba Ngouma (1♂); Imbalanga Camp 7–8.IV.2024, Lepiled light trap, M. Bashford, G. László, M. Talani, S. Yaba Ngouma (1♂); Moba 11–12.IV.2024, Actinic Light trap, M. Bashford, G. László, M. Talani, S. Yaba Ngouma (3♂); Mbomo Headquarters 19–22.IV.2024, MV Light trap, M. Bashford, G. László, M. Talani, S. Yaba Ngouma (5♂); Imbalanga Camp 5–11.IV.2024, Actinic Light trap, M. Bashford, G. László, M. Talani, S. Yaba Ngouma (3♂); Kokoua base 11–13.IX.2024, Lepiled light trap, M. Bashford, G. Lászlo, M. Talani, A. Volynkin, S. Yaba Ngouma (2♂); Camp Imbalanga 15–18.IX.2024, Lepiled Light trap, M. Bashford, G. László, A. Volynkin, S. Yaba Ngouma (1♂, 1♀); Mbomo Headquarters 28–30.IX.2024, Actinic light trap, M. Bashford, G. László, M. Talani, S. Yaba Ngouma (3♂); Mbomo Headquarters 28.IX–1.X.2024, MV light trap, M. Bashford, G. László, A. Volynkin (5♂); Mbomo Headquarters 28.IX–1.X.2024, Lepiled light trap, M. Bashford, G. László, M. Talani, S. Yaba Ngouma (1♂); Imbalanga camp 14–17.XI.2024, Actinic light trap M. Bashford, I. Elliott, A. Kirk-Spriggs (1♂); Camp Imbalanga 14–18.XI.2024, MV light trap, M. Bashford, I. Elliott, A. Kirk-Spriggs (3♂); Mboko 18–21.XI.2024, Lepiled light trap, M. Bashford, I. Elliott, A. Kirk-Spriggs (1♂); Mboko 18–21.XI.2024, MV light trap, M. Bashford, I. Elliott, A. Kirk-Spriggs (1♂) (ANHRT).Total. April: 13♂; September: 12♂, 1♀; November: 6♂.Phenology. One male in November with spermatophore.Distribution. Previously known from Central African Republic and also recorded in Republic of Congo (Nouabalé-Ndoki National Park, and presently Odzala-Kokoua National Park).

*Poreuomena sanghensis* Massa, 2013Massa, 2013, J. Orth. Res. 22: 140; type locality: Dzanga-Ndoki National Park (Central African Republic); depository: MSNG, Genoa (♂ holotypus).Material examined. Republic of Congo, Odzala-Kokoua NP, Imbalanga bai 6–9.IV.2024, Actinic Light trap, M. Bashford, G. László, M. Talani, S. Yaba Ngouma (1♂); Lobo Res. Camp 16–17.IV.2024, Lepiled light trap, M. Bashford, G. László, M. Talani, S. Yaba Ngouma (1♂); Kokoua base 5–13.IX.2024, MV light trap, M. Bashford, G. László, M. Talani, A. Volynkin (1♂); Lekoli river near Mboko 21–23.IX.2024, Lepiled Light trap, M. Bashford, G. László, M. Talani, A. Volynkin, S. Yaba Ngouma (4♂) (ANHRT).Total. April: 2♂; September: 5♂.Distribution. Known from Central African Republic (Dzanga-Ndoki National Park and Dzanga-Sangha Special Reserve), Cameroon, Gabon, and Republic of Congo (Nouabalé-Ndoki National Park, and presently Odzala-Kokoua National Park).

*Cestromoecha tenuipes* (Karsch, 1890)Karsch, 1890, Entom. Nachricht. 16: 363 (*Poreuomena*); type locality: Barombi Station (Cameroon); depository: MfN, Berlin (♀♀ syntypes).Material examined. Republic of Congo, Odzala-Kokoua NP, Lekoli river near Mboko 22–23.IX.2024, Actinic light trap, M. Bashford, G. László, M. Talani, A. Volynkin, S. Yaba Ngouma (2♂); Mbomo Headquarters 28.IX–1.X.2024, MV light trap, M. Bashford, G. László, A. Volynkin (2♂); Lobo Res. Camp 22–23.XI.2024, MV light trap, M. Bashford, I. Elliott, A. Kirk-Spriggs (1♂) (ANHRT).Total. September: 4♂; November: 1♂.Distribution. Uncommon in west-central tropical Africa.

Tribe Preussiini Karsch, 1890*Brycoptera lobata* Ragge, 1981Ragge, 1981, J. Nat. Hist. 15(2): 327; type locality: Dala Tango (Angola); depository: NHM, London (♂ holotypus).Material examined. Republic of Congo, Odzala-Kokoua NP, Kokoua base 5–13.IX.2024, MV light trap, M. Bashford, G. László, M. Talani, A. Volynkin (2♂, 1♀) (ANHRT).Total. September: 2♂, 1♀.Distribution. Uncommon, but widespread; previously known from Angola, Cameroon, Central African Republic, Equatorial Guinea, Ivory Coast, and Uganda [21].

*Preussia lobatipes* Karsch, 1890Karsch, 1890, Entom. Nachricht. 16(23): 365; type locality: Barombi Station (Cameroon); depository: MfN (♀ holotypus). Material examined. Republic of Congo, Odzala-Kokoua NP, Mbomo Headquarters 19–22.IV.2024, MV Light trap, M. Bashford, G. László, M. Talani, S. Yaba Ngouma (1♂); Kokoua base 5–13.IX.2024, MV light trap, M. Bashford, G. László, M. Talani, A. Volynkin (7♂) (ANHRT).Total. April: 1♂; September: 7♂.Phenology. Three males collected in September had a spermatophore.Distribution. Uncommon, but widespread in west-central tropical Africa.

*Enochletica ostentatrix* Karsch, 1896Karsch, 1896, Stett. Entomol. Z. 57: 337; type locality: Lolodorf (Cameroon); depository: MfN (♂ holotypus).Material examined. Republic of Congo, Odzala-Kokoua NP, Mbomo Headquarters 5–11.IV.2024, Actinic Light trap, M. Bashford, G. László, M. Talani, S. Yaba Ngouma (1♂); Moba 11–12.IV.2024, Actinic Light trap, M. Bashford, G. László, M. Talani, S. Yaba Ngouma (1♂); Kokoua base 5–13.IX.2024, MV light trap, M. Bashford, G. László, M. Talani, A. Volynkin (12♂); Lobo Res. Camp 20–27.IX.2024, MV light trap, M. Bashford, G. László, A. Volynkin (1♂); Mbomo Headquarters 28.IX–1.X.2024, MV light trap, M. Bashford, G. László, A. Volynkin (1♂); Camp Imbalanga 14–18.XI.2024, MV light trap, M. Bashford, I. Elliott, A. Kirk-Spriggs (1♂); Mboko 18–21.XI.2024 MV light trap, M. Bashford, I. Elliott, A. Kirk-Spriggs (1♂) (ANHRT).Total. April: 2♂; September: 14♂; November: 2♂.Distribution. Common and widespread in west-central tropical Africa. Recently Gorochov [28] described *E. simulata* from Uganda.

*Weissenbornia p. praestantissima* Karsch, 1888Karsch, 1888, Entom. Nachricht. 14: 66; type locality: Kribi (Cameroon); depository: MfN (♂ holotypus). Material examined. Republic of Congo, Odzala-Kokoua NP, Imbalanga Camp 5–11.IV.2024, Actinic Light trap, M. Bashford, G. László, M. Talani, S. Yaba Ngouma (1♂); Imbalanga Camp 7–8.IV.2024, Lepiled light trap, M. Bashford, G. László, M. Talani, S. Yaba Ngouma (1♂); Kokoua base 5–13.IX.2024, MV light trap, M. Bashford, G. Lászlo, M. Talani, A. Volynkin (5♂); Imbalanga camp 14–18.XI.2024, MV light trap, M. Bashford, I. Elliott, A. Kirk-Spriggs (1♂); Mboko 18–21.XI.2024, MV light trap, M. Bashford, I. Elliott, A. Kirk-Spriggs (1♂) (ANHRT).Total. April: 2♂; September: 5♂; November: 2♂.Distribution. Uncommon and local in west-central tropical Africa; recently, the subspecies *aurea* Gorochov, 2023 has been described from Uganda.

Tribe Vossiini Cadena-Castañeda, 2015*Vossia obesa* Brunner von Wattenwyl, 1891Brunner von Wattenwyl, 1891, Verh. der Zoologisch-Botanischen Gesellsch. Wien 41: 140; type locality: Cameroon; depository: NMW (♀ holotypus).Material examined. Republic of Congo, Odzala-Kokoua NP, Kokoua base 5–13.IX.2024, MV light trap, M. Bashford, G. László, M. Talani, A. Volynkin (14♂, 5♀); Lobo Res. Camp 20–27.IX.2024, MV light trap, M. Bashford, G. László, A. Volynkin (1♂); Mbomo Headquarters 28.IX–1.X.2024, MV light trap, M. Bashford, G. László, A. Volynkin (8♂); Mbomo Headquarters 28.IX–1.X.2024, Lepiled light trap, M. Bashford, G. László, M. Talani, A. Volynkin, S. Yaba Ngouma (1♂); Camp Imbalanga 14–18.XI.2024, MV light trap, M. Bashford, I. Elliott, A. Kirk-Spriggs (1♂); Mboko 18–21.XI.2024 MV light trap, M. Bashford, I. Elliott, A. Kirk-Spriggs (4♂); Lobo Res. Camp 22–30.XI.2024, MV light trap, M. Bashford, I. Elliott, A. Kirk-Spriggs (2♂, 1♀) (ANHRT).Total. September: 24♂, 5♀; November: 7♂, 1♀.Distribution. Widespread and common in west-central tropical Africa.

*Azamia biplagiata* Bolívar, 1906Bolívar, 1906, Mem. Soc. espan. Hist. nat. 1: 341; type locality: Cameroon; depository: MNCN (♂ holotypus).Material examined. Republic of Congo, Odzala-Kokoua NP, Kokoua 5–13.IX.2024, MV light trap, M. Bashford, G. László, M. Talani, A. Volynkin (3♂); Imbalanga Camp 13–20.IX.2024, MV Light trap, M. Bashford, G. László, M. Talani, A. Volynkin (1♂); Mbomo Headquarters 28.IX–1.X.2024, MV light trap, M. Bashford, G. László, A. Volynkin (5♂); Mboko 18–21.XI.2024 MV light trap, M. Bashford, I. Elliott, A. Kirk-Spriggs (1♂); Lobo Res. Camp 22–30.XI.2024, MV light trap, M. Bashford, I. Elliott, A. Kirk-Spriggs (1♂) (ANHRT).Total. September: 9♂; November: 2♂.Distribution. Widespread in west-central tropical Africa.

Tribe Terpnistrini Brunner von Wattenwyl, 1878*Gelotopoia bicolor* Brunner von Wattenwyl, 1891Brunner von Wattenwyl, 1891, Verh. der Zoologisch-Botanischen Gesellsch. Wien 41: 11; type locality: Sierra Leone; depository: MZPW Warsaw (♂ holotypus).Material examined. Republic of Congo, Odzala-Kokoua NP, Imbalanga Camp 5–9.IV.2024, MV Light trap, M. Bashford, G. László, M. Talani, S. Yaba Ngouma (3♂); Imbalanga Camp 5–11.IV.2024, Actinic Light trap, M. Bashford, G. László, M. Talani, S. Yaba Ngouma (1♂); Imbalanga Camp 4–5.IX.2024, Actinic Light trap, M. Bashford, G. László, M. Talani, A. Volynkin (1♂); Kokoua base 5–13.IX.2024, MV light trap, M. Bashford, G. Lászlo, M. Talani, A. Volynkin (8♂, 3♀); Lobo Res. Camp 20–27.IX.2024, MV light trap, M. Bashford, G. László, A. Volynkin (2♂); (ANHRT).Total. April: 4♂; September: 11♂, 3♀.Distribution. Uncommon, but widespread from west to central and southern (Zambia) tropical Africa.

*Tropidophrys amydra* Karsch, 1896Karsch, 1896, Stett. Entomol. Z. 57: 341; type locality: Victoria (Cameroon); depository: MfN (♂ holotypus).Material examined. Republic of Congo, Odzala-Kokoua NP, Lobo Res. Camp 20–27.IX.2024, MV light trap, M. Bashford, G. László, A. Volynkin (1♀) (ANHRT).Total. September: 1♀.Distribution. Uncommon in central tropical Africa.

Tribe Zeuneriini Karsch, 1890*Zeuneria longicercus* Sjöstedt, 1929Sjöstedt, 1929, in: Schouteden, Rev. Zool. Bot. Afr. 17: 39; type locality: Inkisi (Democratic Republic of Congo); depository: MRAC, Tervuren (♂ holotypus).Material examined. Republic of Congo, Odzala-Kokoua NP, Lobo Res. Camp 17–18.IV.2024, Lepiled light trap, M. Bashford, G. László, M. Talani, S. Yaba Ngouma (1♂); Kokoua base 6–12.IX.2024, carrion bait, M. Bashford, G. Lászlo, M. Talani, A. Volynkin, S. Yaba Ngouma (2♂); Lekoli river near Mboko 21–26.IX.2024, carrion bait, M. Bashford, G. László, M. Talani, A. Volynkin, S. Yaba Ngouma (1♂) (ANHRT).Total. April: 2♂; September: 3♂.Distribution. Uncommon in central tropical Africa.

*Zeuneria biramosa* Sjöstedt, 1929Sjöstedt, 1929, In: Schouteden, Rev. Zool. Bot. Afr. 17: 40; type locality: Bas Uele (Democratic Republic of Congo); depository: MRAC, Tervuren (♂ holotype).Material examined. Republic of Congo, Odzala-Kokoua NP, Kokoua base 6–12.IX.2024, carrion bait, M. Bashford, G. Lászlo, M. Talani, A. Volynkin, S. Yaba Ngouma (1♂) (ANHRT).Total. September: 1♂.Distribution. Uncommon in central tropical Africa.

*Zeuneria melanopeza* Karsch, 1889Karsch, 1889[1888], Berlin Ent. Z. 32: 443; type locality: Barombi Station (Cameroon); depository: MfN, Berlin (♀ holotype).Material examined. Republic of Congo, Odzala-Kokoua NP, Kokoua base 6–12.IX.2024, carrion bait, M. Bashford, G. Lászlo, M. Talani, A. Volynkin, S. Yaba Ngouma (2♂) (ANHRT).Total. September: 2♂.Distribution. Uncommon in central tropical Africa.Remarks. The use of carrion baits enabled the collection of three species of *Zeuneria* in the same site (Kokoua base).

Tribe Otiaphysini Karsch, 1889Genus *Tetraconcha* Karsch, 1890

Presently, the number of species belonging to the genus *Tetraconcha* is 20; all the species but one are known for the male sex; *T. longipes* (Bolívar, 1893) is known only for the female sex. The female sex is very rarely captured in light, and consequently it is known only from four species (cf. Table 1); morphologically, the females of *Tetraconcha* are very different from the males, so much so that Bolívar [29] described the genus *Tellidia* for the species *longipes*, thinking that he was dealing with a genus different from *Tetraconcha*. With the new material obtained in recent years, it was possible to update the measures in Table 1 and Table 2, which may help the species identification. Indeed, it emerged that there are some undescribed species with one or two windows in the tegmina in the series collected in the Ivory Coast; in the present paper, four new species, only based on males, are described, two from the Republic of Congo and two from the Ivory Coast. It is possible that one of them could belong to *T. longipes*, but until males and females of this species are collected together, it remains impossible to understand which is the male for *T. longipes*.

*Tetraconcha danielae* n. sp. (Figure 18)urn:lsid:zoobank.org:act:A4441F6D-CBB7-4894-8D15-C2359E424593

Material examined. Republic of Congo, Odzala-Kokoua NP, Imbalanga Camp, 00°45′47″ N, 15°15′39″ E, 5–9.IV.2024, MV Light trap, M. Bashford, G. László, M. Talani, S. Yaba Ngouma (♂ holotype in ANHRT); Kokoua base (540 m) 01°28′39″ N, 15°16′42″ E, 5–13.IX.2024, MV light trap, M. Bashford, G. László, M. Talani, A. Volynkin (2♂ paratypes in ANHRT and BMPC); Mbomo Headquarters (540 m) 00°26′13″ N, 14°42′01″ E, 28.IX–1.X.2024, MV light trap, M. Bashford, G. László, A. Volynkin (1♂ paratype) (ANHRT).

Description. Male (Figure 18A). *Colour*. Head, antennae, legs, and abdomen brown, pronotum green, tegmina green with brown apex and stridulatory area, hind wings roseate; a black spot at the base of each tegmen. *Head and antennae*. Fastigium of vertex narrow, sulcate above, separated from fastigium of frons. Eyes rounded, projecting prominently. Antennae longer than tegmina. *Legs*. Fore coxae armed with a small spine. Fore tibiae furrowed dorsally, distinctly widening above tympanum, conchate on both sides. Fore femora armed on inner ventral side with 9 spines, fore tibiae with 6 spines + 1 spur on inner and on outer ventral sides, 3 spines + 1 spur on outer dorsal side, mid femora armed with 8 spines on outer ventral side, mid-tibiae with 12–13 spines on outer and inner ventral sides + 1 spur on each side, and 4–5 spines + 1 spur on inner dorsal side, hind femora armed with 4 small spines on outer and on inner ventral sides, hind tibiae with many spines on ventral and dorsal sides + 3 spurs on each side. *Thorax*. Pronotum narrowing anteriorly, flat above, anterior margin incurved, posterior margin rounded, humeral sinus well developed, lobes of pronotum rounded. *Tegmina* narrow with rounded apices. Wings longer than tegmina. Stridulatory area and veinlets of left tegmen shown in Figure 18B; stridulatory file angularly arched and composed of ca. 50 evenly spaced teeth, preceded by 3 longer, stouter and spaced out teeth (Figure 18C). *Abdomen*. Epiproct flat and concave, with distal part more or less straight, covered by dense yellow pilosity (Figure 18D). Subgenital plate ending with a narrow V-shaped concavity and two processes similar to styli, cerci slender and incurved with rounded apex (Figure 18D,E).

Female. Unknown.Measurements. Body length: 19.0–21.0; length of pronotum: 5.1–5,2; pronotum depth: 3.2–3.3; length of hind femora: 24.9–25.5; length of tegmina: 36.0–36.9; width of tegmina: 5.5–5.7.

Diagnosis. *T. danielae* n. sp. is characterized mainly by the presence of blackish spots among the veinlets of tegmina (Figure 18A,B), its stridulatory area of the left tegmina (Figure 18B), its peculiar stridulatory file (Figure 18C), its epiproct (Figure 18D), and its cerci and subgenital plate (Figure 18D,E).

Etymology. This species is dedicated with much affection to Daniela Patti, a librarian from Palermo University. She has helped me on many occasions by finding old papers that are difficult to consult.

Remarks. *T. danielae* n. sp. has stridulatory area, epiproct, and cerci tip similar to those of *T. bicolor* Gorochov, 2023 from Uganda, but the stridulatory file is clearly different (Figure 23); also, the subgenital plate is different (see Gorochov [28]).

Distribution. Presently, it is known only from the Odzala-Kokoua National Park in the Republic of Congo.

**Figure 18 insects-16-00241-f018:**
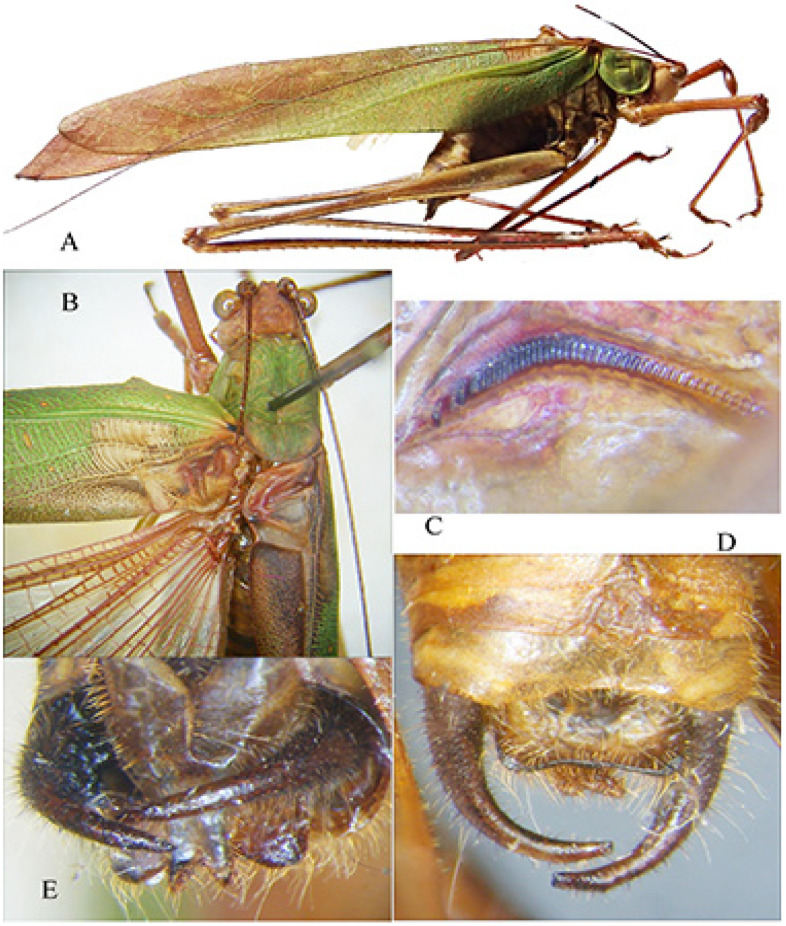
*Tetraconcha danielae* n. sp., holotype (**A**): habitus of male; (**B**): stridulatory area of left tegmen; (**C**): stridulatory file on underside of left tegmen; (**D**): last tergites with epiproct and cerci; (**E**): ventral view of subgenital plate and cerci.

*Tetraconcha fuscopunctata* n. sp. (Figure 19)urn:lsid:zoobank.org:act:91474B0B-1395-4455-A98D-15144329ED75

Material examined. Republic of Congo, Odzala-Kokoua NP, Moba, 00°48′52″ N, 15°05′15″ E, 11–12.IV.2024, Actinic Light trap, M. Bashford, G. László, M. Talani, S. Yaba Ngouma (♂ holotype in ANHRT, 1♂ paratype in BMPC); Bongassou Forest near Lubo (400 m) 00°32′50″ N, 14°51′47″ E, 23–26.IX.2024, Actinic Light trap, M. Bashford, G. László, M. Talani, A. Volynkin, S. Yaba Ngouma (1♂ paratype); Mbomo Headquarters (540 m) 00°26′13″ N, 14°42′01″ E, 28–30.IX.2024, Actinic light trap, M. Bashford, G. László, M. Talani, S. Yaba Ngouma (1♂ paratype); Mbomo Headquarters (540 m) 00°26′13″ N, 14°42′01″ E, 28.IX–1.X.2024, MV light trap, M. Bashford, G. László, A. Volynkin (2♂ paratypes); Mbomo Headquarters (540 m) 00°26′13″ N, 14°42′01″ E, 28.IX–1.X.2024, Lepiled light trap, M. Bashford, G. László, M. Talani, S. Yaba Ngouma (2♂ paratypes); Imbalanga camp 14–18.XI.2024, MV light trap, M. Bashford, I. Elliott, A. Kirk-Spriggs (1♂ paratype) (ANHRT).

Description. Male (Figure 19A). *Colour*. Brownish, with black spots among veinlets of tegmina, antennae black, except for brown scapus, stridulatory area blackish. *Head and antennae*. Fastigium of vertex narrow, sulcate above, separated from fastigium of frons. Eyes rounded, projecting prominently. Antennae long. *Legs*. Fore coxae armed with a small spine. Fore tibiae furrowed dorsally, distinctly widening above tympanum, conchate on both sides. Fore femora armed on inner ventral side with 8–9 spines, fore tibiae with 6–7 spines + 1 spur on inner and on outer ventral sides, 3 spines + 1 spur on outer dorsal side, mid femora armed with 8–10 spines on outer ventral side, mid-tibiae with 15–16 spines on outer and inner ventral sides + 1 spur on each side, and 4–5 spines + 1 spur on inner dorsal side, hind femora armed with 3 small spines on outer and on inner ventral sides, hind tibiae with many spines on ventral and dorsal sides + 3 spurs on each side. *Thorax*. Pronotum narrowing anteriorly, flat above, anterior margin incurved, posterior margin rounded, humeral sinus well developed, lobes of pronotum rounded. *Tegmina* narrow with rounded apices. Wings longer than tegmina. Stridulatory area and veinlets of left tegmina shown in Figure 19B; stridulatory file arched and composed by ca. 30 evenly spaced teeth in the proximal part, preceded by a longer, stouter tooth and a smaller one, and ca. 30 denser teeth in the distal part, (Figure 19C). *Abdomen*. Subgenital plate narrow and long with an apical V-shaped concavity and two processes similar to styli, cerci slender and incurved (Figure 19D,E).

Female. Unknown.

Measurements. Body length: 17.7–18.1; length of pronotum: 4.6–4.7; pronotum depth: 3.4–3.5; length of hind femora: 24.8–27.0; length of tegmina: 32.4–33.8; width of tegmina: 4.7–4.8.

Diagnosis. *T. fuscopunctata* n. sp. is characterized mainly by the presence of blackish spots among the veinlets of brown tegmina (Figure 19A,B), its stridulatory area of its left tegmina (Figure 19B), its peculiar stridulatory file (Figure 19C), and its subgenital plate (Figure 19E).

Etymology. From Latin *fuscus* (=dark) and *punctatus* (=with scattered dots); it is named for its dark colour and the black dots present among the veinlets of tegmina.

Remarks. *T. fuscopunctata* n. sp. has a stridulatory area similar to that of *T. unicolor* Gorochov, 2023 from Uganda, but the stridulatory file is different (Figure 23); in *T. unicolor,* the subgenital plate is concave, while in *T. fuscopunctata* n. sp., it is V-shaped; the cerci are similar in the two species. In addition, *T. unicolor* also differs from *T. laszloi* Massa, 2023 mainly in their stridulatory file, which, in *T. unicolor*, has an almost straight sublateral part with a few larger and more sparsely placed teeth than the others (three of these teeth are larger), while in *T. laszloi*, it is 2.1 long, arched, and composed of ca. 50 evenly spaced teeth (Figure 23).

Distribution. Presently, it is known only from the Odzala-Kokoua National Park in the Republic of Congo.

**Figure 19 insects-16-00241-f019:**
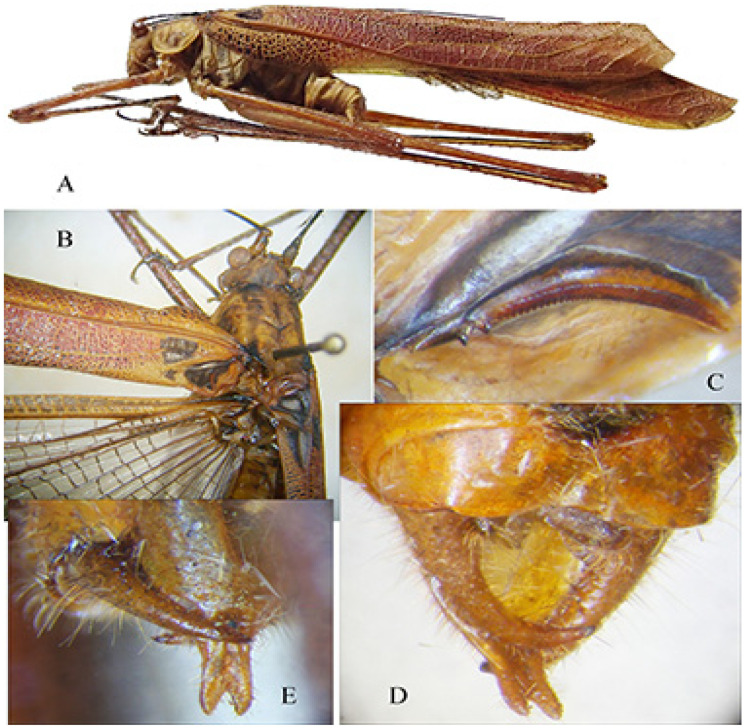
*Tetraconcha fuscopunctata* n. sp, holotype. (**A**): habitus of male; (**B**): stridulatory area of left tegmen; (**C**): stridulatory file on underside of left tegmen; (**D**): last tergites and cerci; (**E**): ventral view of subgenital plate and cerci.

*Tetraconcha alberti* n. sp. (Figure 20)urn:lsid:zoobank.org:act:AB9C3FF7-2DCD-4439-BC3D-148FBC58893DMaterial examined. Ivory Coast, Taï National Park, Res. Station (174 m) 13–21.III.2017 (light), B. Massa (♂ holotype, ♂ paratype) (BMPC); Ivory Coast, Mt. Tonkoui (1171 m) 7.XII.2018, P. Moretto (1♂ paratype) (BMPC).

Description. Male (Figure 20A). *Colour*. Green, with black spots at the base of the tegmina, antennae yellowish, four black spots on inner fore femora and at the base of spines on fore tibiae. *Head and antennae*. Fastigium of vertex narrow, flat and bifid, sulcate above, separated from fastigium of frons. Eyes rounded, projecting prominently. Antennae long. *Legs*. Fore coxae armed with a small spine. Fore tibiae furrowed dorsally, distinctly widening above tympanum, conchate on both sides. Fore femora armed on inner ventral side with 2–3 spines, fore tibiae with 6–7 spines + 1 spur on inner and on outer ventral sides, 3 spines + 1 spur on outer dorsal side, mid femora armed with 7–8 spines on outer ventral side, mid-tibiae with 16–17 spines on outer and inner ventral sides + 1 spur on each side, and 2–3 small spines + 1 spur on inner dorsal side; hind femora armed with 1–2 small spines on outer and on inner ventral sides, hind tibiae with many spines on ventral and dorsal sides + 3 spurs on each side. *Thorax*. Pronotum narrowing anteriorly, flat above, anterior margin incurved, posterior margin rounded, humeral sinus well developed, lobes of pronotum rounded. *Tegmina* wide with rounded apices. An oval window with posterior margin concave present between upper and lower cubital areas of left tegmen, where veinlets are more raised (Figure 20B); on right tegmen, oval mirror is present (Figure 20C). The specimen from Mt. Tonkoui is bigger, with slightly wider tegmina. Wings longer than tegmina. Stridulatory area of left tegmina shown in Figure 20B,C; stridulatory file arched and composed of ca. 200 evenly spaced dense teeth (Figure 20D). *Abdomen*. Subgenital plate narrow and long with an apical V-shaped concavity and two processes similar to styli; cerci stout and incurved (Figure 20E).

Female. Unknown.

Measurements. Body length: 17.0–18.5; length of pronotum: 4.4–5.5; pronotum depth: 3.6–4.0; length of hind femora: 25.8–26.1; length of tegmina: 26.7–31.0; width of tegmina: 7.1–10.0.

Diagnosis. *T. alberti* n. sp. is characterized mainly by the presence of black spots on the fore femora and tibiae, its stridulatory area of left and right tegmina has an oval window (Figure 20A–C); its stridulatory file has dense teeth (Figure 20D).

Etymology. This species is dedicated to my very nice young grandson Alberto Cigna Massa, with all my affection.

Remarks. *T. alberti* n. sp. has a stridulatory area similar to that of *T. ruzzieri* Massa, 2017 from the Ivory Coast, which inhabits the same habitat and localities; however, the stridulatory area of *T. ruzzieri* is more protruding on the back, and the stridulatory file is different (Figure 23).

Distribution. Presently, it is known only from Taï National Park and Mt. Tonkoui, Ivory Coast.

**Figure 20 insects-16-00241-f020:**
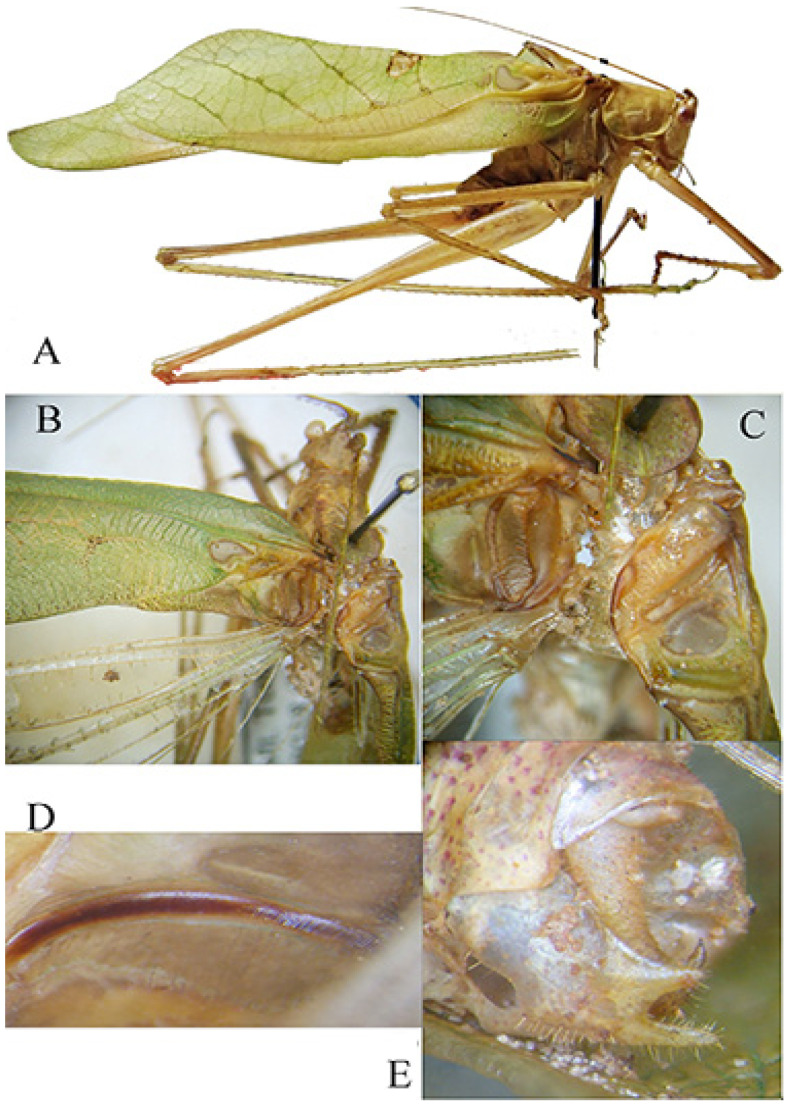
*Tetraconcha alberti* n. sp. (**A**): habitus of male; (**B**): stridulatory area of left tegmen; (**C**): detailed view of stridulatory area of left and right tegmina; (**D**): stridulatory file on underside of left tegmen; (**E**): subgenital plate and cerci in ventro-lateral view.

*Tetraconcha tonkouiensis* n. sp. (Figure 21)urn:lsid:zoobank.org:act:16CD03D2-B6E9-4CBB-8365-E7644A770757Material examined. Ivory Coast, Mt. Tonkoui (1171 m) 31.X.2018, P. Moretto (♂ holotype); Ivory Coast, Mt. Tonkoui (1171 m) 12.III.2018, P. Moretto (♂ paratype) (BMPC).

Description. Male (Figure 21A). *Colour*. Green, with 7–8 black spots on fore femora, antennae brown, stridulatory area blackish with yellowish stridulatory files. *Head and antennae*. Fastigium of vertex narrow, flat, sulcate above, separated from fastigium of frons. Eyes rounded, projecting prominently. Antennae long. *Legs*. Fore coxae armed with a small spine. Fore tibiae furrowed dorsally, distinctly widening above tympanum, conchate on both sides. Fore femora armed on inner ventral side with 5–6 spines, fore tibiae with 3–4 spines + 1 spur on inner and on outer ventral sides, 3 spines + 1 spur on outer dorsal side, mid femora armed with 11–12 spines on outer ventral side, mid-tibiae with 16–17 spines on outer and inner ventral sides + 1 spur on each side, and 2–3 small spines + 1 spur on inner dorsal side, hind femora armed with 3 small spines on outer and on inner ventral sides, hind tibiae with many spines on ventral and dorsal sides + 3 spurs on each side. *Thorax*. Pronotum narrowing anteriorly, flat above, anterior margin incurved, posterior margin rounded, humeral sinus well developed, lobes of pronotum rounded. *Tegmina* wide with rounded apices. An oval window present on upper cubital areas of left tegmen, and another one more or less triangular with a rounded side on the left of the raised stridulatory files (Figure 21B). Mirror of the right tegmen wide and oval (Figure 21B). Wings longer than tegmina. Stridulatory area of left and right tegmina shown in Figure 21B; stridulatory file arched and composed by two parts: on the proximal part, about 150 dense and evenly spaced teeth, and on the distal, more arched part, ca. 80–90 bigger evenly spaced teeth (Figure 21C). *Abdomen*. Cerci stout and incurved, subgenital plate narrow and long with an apical concavity and two processes similar to styles, (Figure 21D,E).

Female. Unknown.

Measurements. Body length: 17.2–19.6; length of pronotum: 4.8–5.3; pronotum depth: 3.2–3.3; length of hind femora: 24.5–25.5; length of tegmina: 31.9–34.2; width of tegmina: 5.9–6.0.

Diagnosis. *T. tonkouiensis* n. sp. is characterized mainly by the presence of black spots on the fore femora and tibiae; its stridulatory area of the left and right tegmina exhibits double windows (Figure 21A,B); its stridulatory file with dense teeth is divided into two parts (Figure 21C).

Etymology. *T. tonkouiensis* is named after Mount Tonkoui in the Ivory Coast. For the naturalistic importance of this tropical mountain, see Moretto et al. [30].

Remarks. *T. tonkouiensis* n. sp. has a stridulatory area similar to that of *T. ruzzieri* Massa, 2017 from the Ivory Coast, which inhabits the same habitat and locality; however, the stridulatory area of *T. tonkouiensis* n. sp. has two windows, of which that on the cubital area is oval (without any concavity, as in *T. ruzzieri*), and the stridulatory file is different (Figure 23).

Distribution. Presently, it is known only from Mt. Tonkoui, Ivory Coast.

**Figure 21 insects-16-00241-f021:**
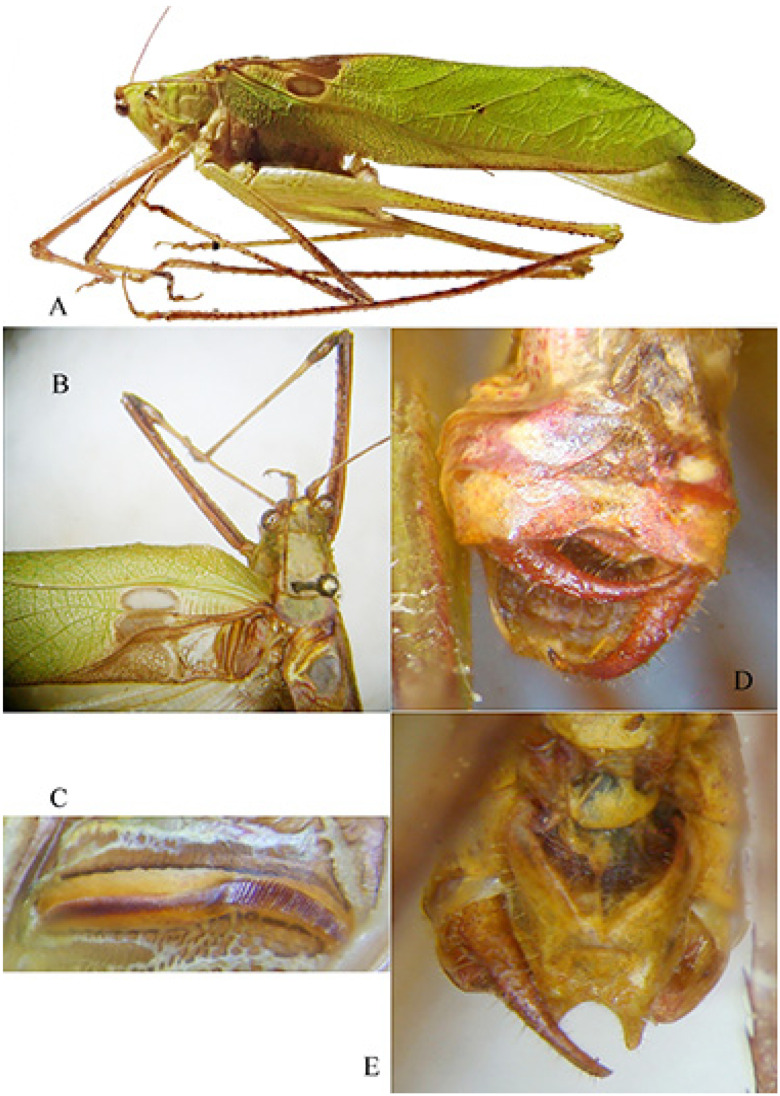
*Tetraconcha tonkouiensis* n. sp. (**A**): habitus of male; (**B**): stridulatory area of left tegmen; (**C**): stridulatory file on underside of left tegmen; (**D**): last tergites and cerci; (**E**): ventral view of subgenital plate and cerci.

*Tetraconcha fijalkowskii* Massa, 2017Massa, 2017, J. Orth. Res. 26(2): 224; type locality: Dzanga-Ndoki National Park (Central African Republic); depository: BMPC, Palermo (♂ holotypus).Material examined. Republic of Congo, Odzala-Kokoua NP, Imbalanga Camp 5–9.IV.2024, MV Light trap, M. Bashford, G. László, M. Talani, S. Yaba Ngouma (1♂); Kokoua base 5–13.IX.2024, MV light trap, M. Bashford, G. László, M. Talani, A. Volynkin (5♂); Mbomo Headquarters 28.IX–1.X.2024, MV light trap, M. Bashford, G. László, A. Volynkin (2♂); Mbomo Headquarters 28.IX–1.X.2024, Lepiled light trap, M. Bashford, G. László, M. Talani, S. Yaba Ngouma (1♂) (ANHRT).Total. April: 1♂; September: 8♂.Distribution. Considered endemic to the Dzanga-Ndoki (Central African Republic) and Nouabalé-Ndoki (Republic of Congo) National Parks; is also present in the Odzala-Kokoua National Park (Republic of Congo).

*Tetraconcha omonomai* Massa, 2017Massa, 2017, J. Orth. Res. 26(2): 224; type locality: Dzanga-Ndoki National Park (Central African Republic); depository: MCSN (♂ holotypus).Material examined. Republic of Congo, Odzala-Kokoua NP, Kokoua base 5–13.IX.2024, MV light trap, M. Bashford, G. László, M. Talani, A. Volynkin (4♂); Kokoua base 6–13.IX.2024, Actinic light trap, M. Bashford, G. Lászlo, M. Talani, A. Volynkin, S. Yaba Ngouma (1♂); Kokoua base 11–13.IX.2024, Lepiled light trap, M. Bashford, G. Lászlo, M. Talani, A. Volynkin, S. Yaba Ngouma (1♂); Camp Imbalanga 13–19.IX.2024, Actinic Light trap, M. Bashford, G. László, A. Volynkin, S. Yaba Ngouma (1♂); Imbalanga camp 14–18.XI.2024, MV light trap, M. Bashford, I. Elliott, A. Kirk-Spriggs (1♂) (ANHRT).Total. September: 7♂; November: 1♂.Phenology. Two males with spermatophore in November.Distribution. Described from Dzanga-Ndoki National Park (Central African Republic), later also found in Nouabalé-Ndoki National Park (Republic of Congo) (Massa 2023), and now also recorded in the Odzala-Kokoua National Park.

*Tetraconcha loubesi* Massa, 2017Massa, 2017, J. Orth. Res. 26(2): 219; type locality: Dzanga-Ndoki National Park (Central African Republic); depository: MCSN, Genoa (♂ holotypus).Material examined. Republic of Congo, Odzala-Kokoua NP, Kokoua base 5–13.IX.2024, MV light trap, M. Bashford, G. László, M. Talani, A. Volynkin (2♂) (ANHRT).Distribution. Described from Dzanga-Ndoki National Park (Central African Republic), later also found in Gabon and in Nouabalé-Ndoki National Park (Republic of Congo) [4], and now also recorded in the Odzala-Kokoua National Park.

*Tetraconcha ndokiensis* Massa, 2017Massa, 2017, J. Orth. Res. 26(2): 221; type locality: Dzanga-Ndoki National Park (Central African Republic); depository: MCSN, Genoa (♂ holotype). Material examined. Republic of Congo, Odzala-Kokoua NP, Imbalanga bai 6–9.IV.2024, Actinic Light trap, M. Bashford, G. László, M. Talani, S. Yaba Ngouma (1♂); Camp Imbalanga 13–19.IX.2024, Actinic Light trap, M. Bashford, G. László, A. Volynkin, S. Yaba Ngouma (1♂); Lekoli river near Mboko 21–23.IX.2024, Lepiled Light trap, M. Bashford, G. László, M. Talani, A. Volynkin, S. Yaba Ngouma (1♂); Lekoli river near Mboko 23–24.XI.2024, Lepiled light trap, M. Bashford, I. Elliott, A. Kirk-Spriggs (1♂) (ANHRT).Total. April: 1♂; September: 2♂; November: 1♂.Distribution. Previously known from Central African Republic, Gabon, Cameroon, and Republic of Congo (Nouabalé-Ndoki National Park) [4].

*Tetraconcha annoyeri* Massa, 2017Massa, 2017, J. Orth. Res. 26(2): 222; type locality: Dzanga-Ndoki National Park (Central African Republic); depository: MCSN (♂ holotypus).Material examined. Republic of Congo, Odzala-Kokoua NP, Mbomo Headquarters 28.IX–1.X.2024, MV light trap, M. Bashford, G. László, A. Volynkin (2♂) (ANHRT).Distribution. Previously reported from the Dzanga-Ndoki National Park (Central African Republic) and Nouabalé-Ndoki National Park (Republic of Congo) [4].

*Tetraconcha morettoi* Massa, 2017Massa, 2017, J. Orth. Res. 26(2): 221; type locality: Czanga-Ndoki National Park (Central African Republic); depository: MCSN, Genoa (♂ holotype). Material examined. Mbomo Headquarters 28.IX–1.X.2024, Lepiled light trap, M. Bashford, G. László, M. Talani, S. Yaba Ngouma (1♂); Imbalanga camp 14–18.XI.2024, MV light trap, M. Bashford, I. Elliott, A. Kirk-Spriggs (1♂) (ANHRT).Total. September: 1♂; November: 1♂.Phenology. One male in November, with spermatophore.Distribution. Previously recorded from the Central African Republic and Gabon [18].

*Tetraconcha stichyrata* Karsch, 1890Karsch, 1890, Entom. Nachricht. 16(23): 360; type locality: Barombi Station (Cameroon); depository: MfN, Berlin (♂ holotype).Material examined. Republic of Congo, Odzala-Kokoua NP, Mbomo Headquarters 28.IX–1.X.2024, MV light trap, M. Bashford, G. László, A. Volynkin (1♂) (ANHRT).Distribution. Quite widespread in west-central tropical Africa.

**Figure 22 insects-16-00241-f022:**
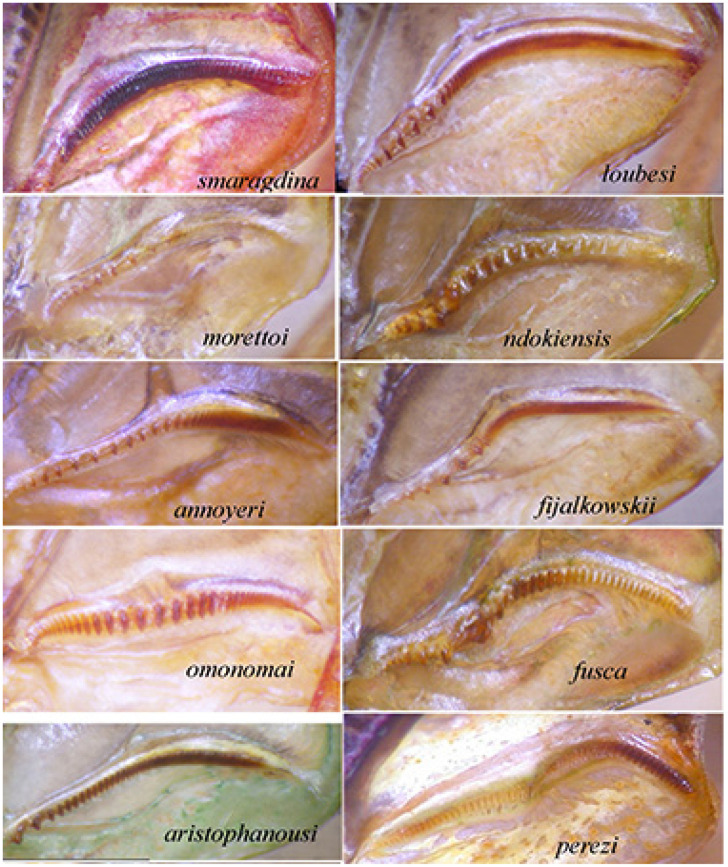
Stridulatory file on underside of left tegmen of ten species of genus *Tetraconcha*.

**Figure 23 insects-16-00241-f023:**
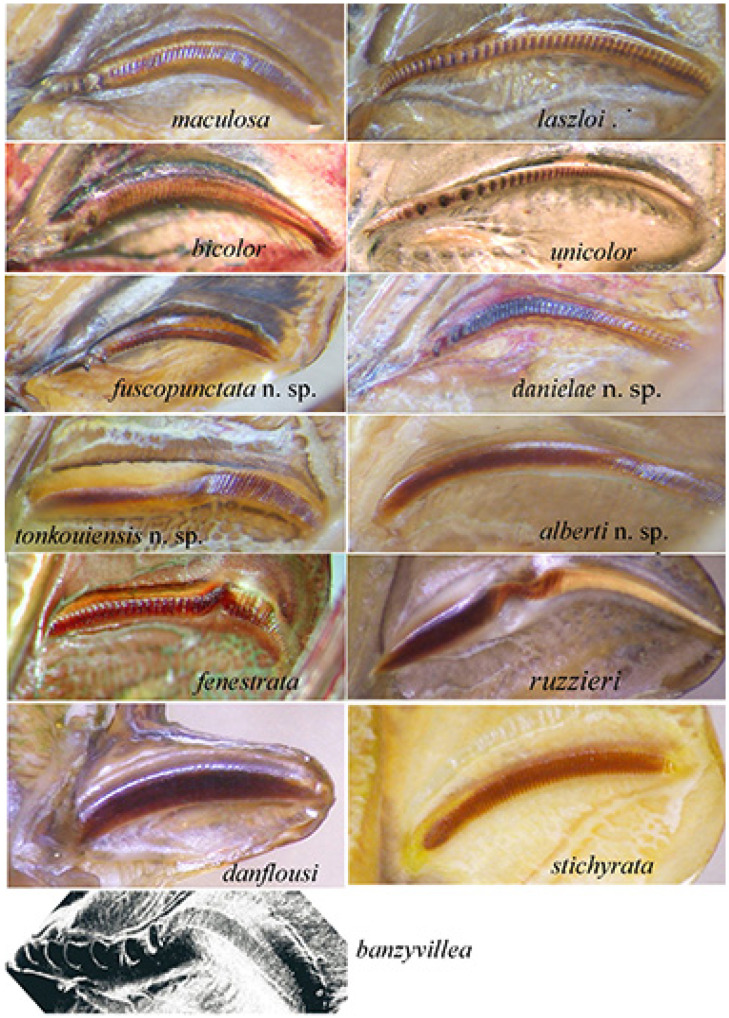
Stridulatory file on underside of left tegmen of other thirteen species of genus *Tetraconcha*. *T. banzyvillea* after Leroy [31].

**Table 1 insects-16-00241-t001:** Measurements (in mm) of all the species of the genus *Tetraconcha* Karsch, 1890.

	Species	Body Length	Length of Tegmina	Width of Tegmina	Length of Hind Femur
1	*T. fenestrata* Karsch, 1890	♂: 16.5 (15.2–17.8); ♀: 29.1 (24.0–31.9)	♂: 30.6 (29.8–32.3); ♀: 30.7 (28.4–31.9)	♂: 8.2 (7.9–9.1); ♀: 10.4 (9.2–11.6)	♂: 23.1 (22.4–25.3); ♀: 25.1 (23.3–26.9)
2	*T. ruzzieri* Massa, 2017	♂: 18.3 (18.0–18.5)	♂: 29.8 (27.3–32.4)	♂: 6.0 (5.7–6.5)	♂: 25.2 (24.9–25.5)
3	*T. danflousi* Massa, 2017	♂: 16.6	♂: 30.3	♂: 4.6	♂: 27.2
4	*T. stichyrata* Karsch, 1890	♂: 16.0 (15.2–16.5); ♀: 15.5–16.8	♂: 22.0 (21.5–25.5); ♀: 24.2–25.5	♂: 5.2 (4.9–5.5); ♀: 5.0–5.5	♂: 18.2 (17.8–19.0); ♀: 18.1–19.0
5	*T. banzyvilliana* Griffini, 1809	♂: 18.0–20.0	♂: 27.0–32.0	♂: 4.2–5.0	♂: 23.4–25.5
6	*T. smaragdina* Brunner von Wattenwyl, 1891	♂: 19.1 (16.4–22.5)	♂: 34.1 (30.0–36.7)	♂: 5.1 (4.1–5.9)	♂: 25.4 (23.7–26.5)
7	*T. perezi* Massa, 2017	♂: 19.2–20.7	♂: 34.0–34.3	♂: 3.9–5.0	♂: 25.6–25.8
8	*T. longipes* (Bolívar, 1893)	♀: 25.0	♀: 27.0	♀: 8.0	♀: 24.0
9	*T. loubesi* Massa, 2017	♂: 16.8 (16.0–18.0)	♂: 31.2 (27.9–32.9)	♂: 5.1 (4.7–5.8)	♂: 25.2 (22.9–26.9)
10	*T. morettoi* Massa, 2017	♂: 17.6 (16.0–19.5); ♀: 22.0	♂: 32.2 (29.0–34.5); ♀: 35.5	♂: 5.1 (4.6–5.9); ♀: 7.1	♂: 24.0 (20.8–26.0); ♀: 24.8
11	*T. ndokiensis* Massa, 2017	♂: 17.5 (14.5–20.5)	♂: 32.0 (28.6–37.2)	♂: 5.4 (4.5–6.5)	♂: 23.1 (20.1–26.0)
12	*T. annoyeri* Massa, 2017	♂: 17.5 (16.5–19.4)	♂: 32.4 (31.6–32.7)	♂: 5.0 (4.3–5.9)	♂: 23.8 (22.8–24.8)
13	*T. fijalkowskii* Massa, 2017	♂: 15.4–19.4	♂: 29.3–30.1	♂: 4.0–4.9	♂: 20.1–21.0
14	*T. omonomai* Massa, 2017	♂: 16.4 (15.5–19.4)	♂: 30.9 (29.6–32.1)	♂: 4.7 (4.1–5.1)	♂: 23.6 (21.2–25.5)
15	*T. aristophanousi* Massa, 2017	♂: 16.7 (15.6–18.0); ♀: 26.0	♂: 35.9 (33.5–36.9); ♀: 37.1	♂: 5.3 (4.9–5.8); ♀: 11.2	♂: 26.1 (23.8–27.5); ♀: 25.6
16	*T. fusca* Massa, 2021	♂: 14.8	♂: 30.7	♂: 4.4	♂: 22.6
17	*T. maculosa* Massa, 2023	♂: 16.8–17.8	♂: 31.5–32.7	♂: 4.0–4.1	♂: 24.5–25.6
18	*T. laszloi* Massa, 2023	♂: 17.8	♂: 33.0	♂: 5.8	♂: 24.8
19	*T. unicolor* Gorochov, 2023	♂: 14.7	♂: 33.0	n. a.	♂: 22.0
20	*T. bicolor* Gorochov, 2023	♂: 15.0–17.0	♂: 32.5–34.0	n. a.	♂: 22.5–23.5
21	*T. danielae* n. sp.	♂: 21.0	♂: 36.9	♂: 5.7	♂: 24.9
22	*T. fuscopunctata* n. sp.	♂: 17.7–18.1	♂: 32.4–33.8	♂: 4.7–4.8	♂: 24.8–27.0
23	*T. alberti* n. sp.	♂: 17.0–18.5	♂: 26.7–31.0	♂: 7.1–10.0	♂: 25.8–26.1
24	*T. tonkouiensis* n. sp.	♂: 17.2–19.6	♂: 31.9–34.2	♂: 5.9–6.0	♂: 24.5–25.5

**Table 2 insects-16-00241-t002:** Measurements of the length of the stridulatory file, the distance between the left tegmen base and the maximum width of the lower cubital area, and the size of the upper and lower cubital areas of all the species of the genus *Tetraconcha* Karsch, 1890, except for *T. longipes* (Bolívar, 1893), known only from the female sex. n. a.: not available.

	*Species*	Length of Stridulatory File (mm)	Distance Between Left Tegmen Base and Max Width of Lower Cubital Area (mm)	Size of Upper and Lower Cubital Areas (mm)
1	*T. fenestrata* Karsch, 1890	1.8	5.4–5.6	1.1–1.2, 0.5–0.6
2	*T. ruzzieri* Massa, 2017	2.0–3.0	3.6	0.7, 0.2
3	*T. danflousi* Massa, 2017	1.0	cubital area occupied by the window	cubital area occupied by the window
4	*T. stichyrata* Karsch, 1890	1.7	7.0–9.0	1.0–1.9, 1.9–2.5
5	*T. banzyvilliana* Griffini, 1909	1.8	4.5–5.0	1.0–1.0
6	*T. smaragdina* Brunner von Wattenwyl, 1891	1.6–1.8	4.9–6.5	0.6–1.0, 0.6–0.9
7	*T. perezi* Massa, 2017	1.8	4.9–5.6	0.6–0.7, 0.6–0.7
8	*T. loubesi* Massa, 2017	1.4–1.6	3.5–5.2	0.6–0.8, 0.6–0.9
9	*T. morettoi* Massa, 2017	1.5–1.6	4.2–5.4	0.7–0.9, 0.6–0.8
10	*T. ndokiensis* Massa, 2017	1.4–1.6	4.0–6.0	0.5–0.9, 0.5–0.9
11	*T. annoyeri* Massa, 2017	1.5–1.7	4.0–6.0	0.5–0.9, 0.6–1.0
12	*T. fijalkowskii* Massa, 2017	1.3–1.4	2.7–3.8	0.7–0.9, 0.5–0.8
13	*T. omonomai* Massa, 2017	1.5–1.6	3.0–4.5	0.2–0.6, 0.2–0.5
14	*T. aristophanousi* Massa, 2017	1.8–2.0	3.0–3.9	0.6–0.7, 0.6–0.8
15	*T. fusca* Massa, 2021	1.5	4.5	0.9, 0.8
16	*T. maculosa* Massa, 2023	1.8–2.0	4.2–4.4	0.6, 0.5
17	*T. laszloi* Massa, 2023	2.1	5.0	1.1, 1.3
18	*T. unicolor* Gorochov, 2023	n. a.	n. a.	n. a.
19	*T. bicolor* Gorochov, 2023	n. a.	n. a.	n. a.
20	*T. danielae* n. sp.	2.7	4.0	1.5, 1.3
21	*T. fuscopunctata* n. sp.	2.1	2.7–2.8	1.2–1.4, 0.8–0.9
22	*T. alberti* n. sp.	3.4–3.7	2.5–3.1	2.2–2.9, 0.9–1.5
23	*T. tonkouiensis* n. sp.	3.2–3.3	3.5–3.8	1.5–1.6, 1.0–1.1


**Concluding remarks on the genus *Tetraconcha* (Figure 22, Figure 23 and Figure 24, Table 1 and Table 2)**


Figure 22 and Figure 23 show the stridulatory files on the underside of the left tegmen of all the presently known species of *Tetraconcha* males for the necessary comparisons. Table 1 lists all the available measurements of the 24 presently known species of the genus. Table 2 reports the available measurements of the length of the stridulatory file, the distance between the base of the left tegmen and the max width of the lower cubital area, and the size of the lower and upper cubital areas of the left tegmen.

To better highlight the objective differences between the different species, the length of the stridulatory file was plotted against the distance between the base of the left tegmen and the maximum width of the lower cubital area; the following species were excluded: *T. bicolor*, *T. unicolor* (measurements not available), *T. danflousi* (unique tegmina structure: cf. Figure 23), and *T. longipes* (only one female known). Figure 24 shows the results, in which the different position of each species can be observed. On the far right at the bottom is *T. stichyrata*, while in the upper left-hand corner is *T. alberti* n. sp.; the ‘*smaragdina*’ group is at the bottom distributed between the centre and the left; the others are higher up.

**Figure 24 insects-16-00241-f024:**
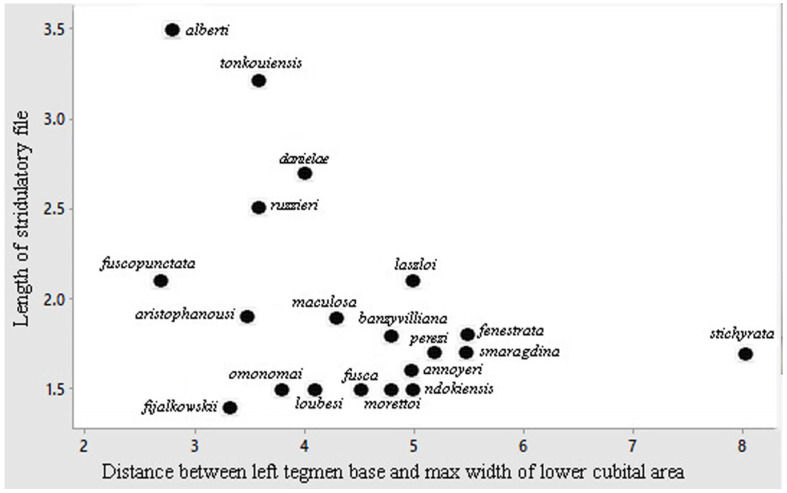
The length of the stridulatory file of *Tetraconcha* species plotted against the distance between the left tegmen base and the maximum width of the lower cubital area.

According to Massa [32], the increase in morphological disparity and taxonomic diversity in the ‘*smaragdina* group’ is very likely the effect of an evolutionary radiation, which may depend on ‘adaptive’ changes to micro-habitats within the wide tropical forest environments of west-central Africa. On the basis of new findings, it seems that all the groups of *Tetraconcha* species, also including those with a window in the tegmina, have undergone an evolutionary radiation in central as well as in western Africa. Many species here treated were collected in the same site and on the same dates and certainly live close by. It is very likely that they occupy different layers of vegetation, but live in the same forest site and have more or less the same phenology, co-occurring and being active in the same months, with small differences. It is very probable that a deeper examination of specimens preserved in Natural History Museums will result in the discovery of further undescribed species (cf. Gorochov [28]; present data). One of the morphological characteristics used to discriminate the species is the stridulatory system: in these taxa, differences in the shape and number of teeth of the stridulatory system certainly result in a different sound, which, according to Heller [33], in most Orthoptera is a very important species-specific barrier. Tropical forest canopy is known as one of the most diversified environments, allowing numerous species of insects and other animals to be adapted and evolved, realizing different ecological niches [34]. Following Simões et al. [35], there are various types of evolutionary radiation; exaptive radiation is the increase in the rate of speciation driven by a previously acquired trait becoming advantageous under a new selective regime (cf. Gould & Vbra [36]). *Tetraconcha* species very likely evolved traits primarily linked to sound communication. Changes in this kind of phenotypic trait through time may have occurred randomly, possibly driven by genetic drift. However, sound traits play an important role in isolation and evolutionary radiation [33]; thus, exaptive radiation may have occurred in the case of the *Tetraconcha* species. Speciation events are often correlated with humid and dry periods; forest expansion during humid periods and retraction during dry periods are considered the best explanation for the patterns of geographical species distribution found on East African mountains [37,38]. Thus, we may hypothesize that in tropical African forests, the ancestors of *Tetraconcha* species could have remained isolated in patches of forest during a dry period and derived populations could have met each other when the climate shifted to a warmer regime (African humid period). They could have undergone multiple episodes of allopatric speciation, this more probable than sympatric radiation [39]. Bioacoustic differences allowed them to remain separated.

Tribe Acrometopini Brunner von Wattenwyl, 1878Genus and species?

Material examined. Republic of Congo, Odzala-Kokoua NP, Imbalanga Camp 14–19.IX.2024, general collection, M. Bashford, G. László, M. Talani, A. Volynkin, S. Yaba Ngouma (♀ nymph) (ANHRT).

Remarks. This single nymph specimen, very distinctive in colour (Figure 25), may belong to the tribe Acrometopini, but the ovipositor is very atypical, lacking the characteristic apical presence of teeth. In the absence of an adult specimen, it is not possible to make more complete hypotheses.

**Figure 25 insects-16-00241-f025:**
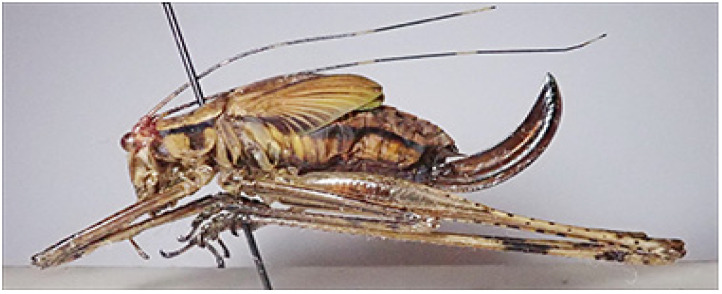
A female nymph of an unknown taxon likely belonging to the tribe Acrometopini Brunner von Wattenwyl, 1878.

Subfamily Mecopodinae Walker, 1871*Anoedopoda lamellata* (Linnaeus, 1758)Linnaeus, 1758, Systema Naturae per Regna tria naturae (10th ed.) 1: 429; type locality: Southern Africa; depository: UZIU, Uppsala (unspecified primary type).Material examined. Republic of Congo, Odzala-Kokoua NP, Lobo Res. Camp 20–27.IX.2024, MV light trap, M. Bashford, G. László, A. Volynkin (1♂) (ANHRT).Distribution. Southern and eastern tropical Africa, now also recorded in the Republic of Congo.

*Macroscirtus k. kanguroo* Pictet, 1888Pictet, 1888, Mem. Soc. Phys. Hist. Nat. Geneve 30(6): 14; type locality: Gabon; depository: MHNG, Geneva (♂ syntypes).Material examined. Republic of Congo, Odzala-Kokoua NP, Camp Imbalanga 14–18.XI.2024, MV light trap, M. Bashford, I. Elliott, A. Kirk-Spriggs (1♀); Bongassou Forest near Lobo 24–28.XI.2024, Malaise trap, M. Bashford, I. Elliott, A. Kirk-Spriggs (1♂) (ANHRT).Other material: Central African Republic, Dzanga-Sangha Special Reserve 12.X.2008 and 17–18.X.2008, P. Annoyer (1♂, 2♀) (BMPC).Remarks. Following the key by Simeu-Noutchom et al. [40] the above specimens were identified as *M. kanguroo*; three subspecies have been described for this species, mainly based on th size: (1) *kanguroo*; (2) *joannis* Bólivar, 1906 from Cabo San Juan (Equatorial Guinea); (3) *insularis* Griffini, 1906 from Bioko Island (Equatorial Guinea).Distribution. Known from Gabon, Cameroon and Equatorial Guinea, now also recorded in the Central African Republic and Republic of Congo.

*Leproscirtus granulosus* (Karsch, 1886)Karsch, 1886, Entom. Nachricht. 12: 316; type locality: between Kwako and Kimpoko (Democratic Republic of Congo); depository: MfN, Berlin (♀ holotype, lost).Material examined. Republic of Congo, Odzala-Kokoua NP, Mbomo Headquarters 28.IX–1.X.2024, MV light trap, M. Bashford, G. László, A. Volynkin (1♀) (ANHRT).Distribution. Uncommon in central Africa, previously unreported in the Republic of Congo.

*Corycoides karschi* (Krauss, 1890)Krauss, 1890, Zoologische Jahrbücher. Abt. Syst. Geogr. und Biol. der Tiere 5(2): 352, 355; type locality: Barombi Station (Cameroon); depository: MfN (♂ holotype).Material examined. Republic of Congo, Odzala-Kokoua NP, Mboko 24.IX.2024, general collection, M. Bashford, G. László, M. Talani, A. Volynkin, S. Yaba Ngouma (1♂) (ANHRT).Distribution. Previously known from Cameroon, the Central African Republic, and Nouabalé-Ndoki National Park (Republic of Congo).

Subfamily Hetrodinae Brunner von Wattenwyl, 1878*Cosmoderus femoralis* (Sjöstedt, 1902)Sjöstedt, 1902, Bihang Kungl. Svenska Vet. Akad. Handl. 27(3): 41; type locality: Cameroon; depository: NHRS, Stockholm (♀ holotype).Material examined. Republic of Congo, Odzala-Kokoua NP, Kokoua 5–13.IX.2024, MV light trap, M. Bashford, G. Lászlo, M. Talani, A. Volynkin (1♂) (ANHRT).Distribution. Presently known from Cameroon and Republic of Congo [4].

## 4. Discussion

### 4.1. Phenology of Tettigoniidae

Sampling in the most representative vegetation areas within the Odzala-Kokoua National Park resulted in a list of at least 108 species. The richest period in terms of the number of individuals collected was September, with 46.2 specimens/day, while the average daily number was lowest in April (21.8) and November (15.6) (Figure 26). Sampling in at least three different seasons of the year increased the number of species; if monitoring had been limited to only one of the seasons, the presence of a few species would certainly have been missed. Interestingly, most species exhibited mating in September, some in November, but no specimens with spermatophores were found in April.

**Figure 26 insects-16-00241-f026:**
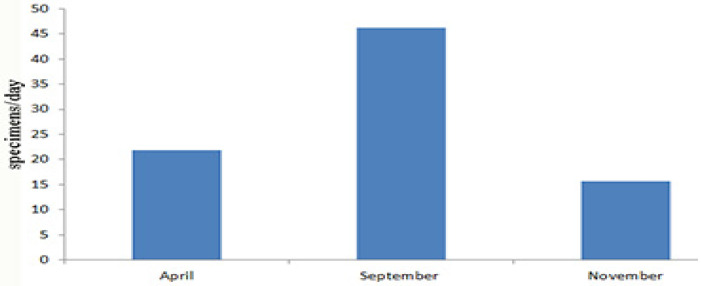
Number of specimens of Tettigoniidae collected every day in Odzala-Kokoua National Park in April, September, and November.

### 4.2. Endemism Rate and Conservation

Because I had the opportunity to study the Orthoptera Tettigonidae of the Nouabalé-Ndoki National Park (Republic of Congo), a comparison of the present results with those previously found in that park has been carried out. In addition, the lists are compared with that of the Dzanga-Ndoki National Park (Central African Republic), which, together with Nouabalé-Ndoki N.P., Dzanga-Ndoki N.P., Dzanga-Sangha Special Reserve (Central African Republic), and Lobéké National Park (Cameroon), have since 2012 been included in the Sangha Trinational Protected Area, a UNESCO World Heritage Site.

Since the late 1960s, efforts have been underway in all west-central tropical African countries to establish more strictly controlled and effectively managed protected areas, but the area of forest strictly protected is still small compared to the forest area remaining in each country. To ensure the survival of many forest species, it is imperative that remaining forest areas be given more adequate protection and that as many existing forest reserves as possible be elevated to National Park status and managed accordingly [1,3,6]. However, it is also imperative to assure the human populations of a better standard of living than the present one, and this is possible only with the intervention of richer countries. Sustainability can only be achieved through human actions based on the qualities of our existence, where we are capable of fully respecting the biophysical limits and dynamic balances of the Earth. Regrettably, richer countries are generally quite indifferent to environmental issues, and this is one of the main causes of the continued exploitation of Africa’s primary forests or their eradication to make way for crops. Perhaps the only way for us researchers to contribute to the conservation of these exceptional forest environments is to continue to talk about them in scientific journals that can have an impact on public opinion.

Generally, biodiversity hotspots are established from a botanical point of view. In fact, to qualify as a biodiversity hotspot, an area must meet two strict criteria: (a) it contains at least 1500 species of vascular plants found nowhere else on Earth (endemic species); (b) it has lost at least 70% of its primary native vegetation. According to Myers [41], mass-extinction is largely centred on tropical forests, which contain at least half of all living species and are being depleted faster than any other biome. Myers identified 10 areas characterized by (a) very high levels of species number and high levels of endemism and (b) unusual rapid rates of depletion. These “hotspot” areas comprise less than 3.5% of remaining primary forests, and they harbour over 34,000 endemic plant species (27% of all plant species in tropical forests and 13% of all plant species worldwide). They also harbour at least 700,000 endemic animal species. He concluded that by concentrating on such areas where needs are greatest and where the pay-off from safeguarding measures would also be greatest, ‘*conservationists can engage in a more systematised response to the challenge of largescale extinctions impending in tropical forests*’.

How can the value of the biodiversity of these places be assessed? Titley et al. [42] have noticed a considerable taxonomic weighting towards vertebrates and an under-representation of invertebrates in the published literature. They observed that this discrepancy is more pronounced in highly cited papers, and in tropical regions, with only 43% of biodiversity research in the tropics including invertebrates; overall, tropical countries were understudied compared to temperate countries. Thus, studies focusing on invertebrates are certainly useful from the conservation point of view. However, the number of species alone cannot be considered an appropriate index of conservation priority, while high levels of endemism occurrence are certainly more informative. We must conserve the areas with high levels of endemism because they contain species that cannot be saved elsewhere [3,6]. Based initially on plant endemism, these hotspots have been confirmed as priority regions for the conservation of biodiversity.

According to Mittermeier et al. [6], concurrent to the development of the hotspot concept was the recognition of the importance of conserving the least-threatened highly diverse regions of the globe, defined on the basis of retaining at least 70% of their original habitat cover, harbouring at least 1500 plant species as endemics, and having a human population density of <5 people/km^2^; they are Amazonia, Congo Forests, Miombo-Mopane Woodlands and Savannas, New Guinea, and North American Deserts, which, on the whole, hold 28% of the world’s mammals and 20% of the world’s amphibians, including 7% of mammals and 11% of amphibians as endemics, in about 7.9% of the world’s land surface (6.1% when including only intact habitats).

The area examined here, together with the Nouabalé-Ndoki National Park, hosts, as far as it has been possible to study, 181 species of Tettigoniidae, which I consider to be good ecological indicators of environmental quality. Of these, 29 (16.0%) are taxa endemic to an area that also includes the Sangha Trinational Protected Area (see Table 3). If one considers that these values only refer to a small group of insects amounting to around 30,000 species worldwide (all Orthoptera: Cigliano et al. [11]), and that the total number of insects is well over a million species, Orthoptera alone certainly account for less than 3% of the overall quantity of insects. Therefore, to have found such a high percentage of endemic species (16.0%) is a demonstration of the importance of this area as a biodiversity hotspot and of the need to continue all efforts to ensure the conservation and care of these tropical forests.

**Table 3 insects-16-00241-t003:** Alphabetical list of known species of Tettigoniidae (divided in subfamilies) in Dzanga-Ndoki National Park (Central African Republic), Odzala-Kokoua National Park (Republic of Congo), and Nouabalé-Ndoki National Park (Republic of Congo). * = species known only from this geographical area (considered as endemic taxa).

	Dzanga-Ndoki National Park (Central African Republic)	Odzala-Kokoua National Park (Republic of Congo)	Nouabalé-Ndoki National Park (Republic of Congo)
Subfamily Pseudophyllinae Burmeister, 1838			
*Adapantus* (*Adapantus*) *longipennis* Beier, 1954		X	X
*Adapantus* (*Adapantus*) *brunneus* Beier, 1957			X
*Adapantus* (*Adapantus*) *osorioi* (Bolívar, 1886)			X
*Chondrodera notatipes* Karsch, 1890			X
*Chondrodera ocellata* Beier, 1954	X		
*Chondrodera subvitrea* Karsch, 1891	X		
*Cymatomera argillata* Karsch, 1891	X	X	X
*Desaulcya ampulla* Brunner von Wattenwyl, 1895	X		
*Habrocomes marmoratus* (Bolívar, 1906)			X
*Habrocomes p. personatus* (Sjöstedt, 1901)	X		
*Habrocomes piotri* n. sp. *		X	
*Lagarodes facetus* Karsch, 1891	X		X
*Lichenochrus congicus* Rehn, 1914			X
*Lichenochrus crassipes* Karsch, 1890		X	
*Lichenochrus marmoratus* Sjöstedt, 1901			X
*Liocentrum aduncum* Karsch, 1891		X	X
*Mormotus montesi* (Bolívar, 1886)	X		X
*Mustius afzelii* Stål, 1873	X		
*Mustius eurypterus* Karsch, 1896	X		
*Mustius superbus* Sjöstedt, 1902	X		X
*Opisthodicrus cochlearistylus* Karsch, 1891	X	X	X
*Oxyaspis congensis* Brunner von Wattenwyl, 1895		X	
*Rhinodera spinifrons* Beier, 1955	X		
*Stenampyx annulicornis* Karsch, 1891	X	X	X
*Tomias* (*Tomias*) *stenopterus* Karsch, 1891	X	X	
*Tympanocompus acclivis* Karsch, 1891	X	X	
*Zabalius* cf. *albifasciatus* (Karsch, 1896)	X	X	X
*Zabalius apicalis* (Bolívar, 1886)			X
*Zabalius lineolatus* (Stål, 1873)	X	X	X
Subfamily Conocephalinae Burmeister, 1838			
*Anthracopsis gigliotosi* Karny, 1907			X
*Conocephalus* (*Conocephalus*) *conocephalus* (Linnaeus, 1767)	X	X	X
*Conocephalus* (*Anisoptera*) *iris* (Serville, 1838)	X		
*Conocephalus* (*Anisoptera*) *maculatus* (Le Guillou, 1841)	X	X	X
*Conocephalus* (*Anisoptera*) *mollyae* n. sp. *		X	
*Lanista africana* (Walker, 1871)	X		
*Plastocorypha nigrifrons* (Redtenbacher, 1891)		X	X
*Ruspolia differens* (Serville, 1838)	X		X
*Ruspolia fuscopunctata* (Karny, 1907)	X		X
*Ruspolia* sp.		X	
*Thyridorhoptrum senegalense* (Krauss, 1877)	X		
Subfamily Hexacentrinae Karny, 1925			
*Hexacentrus dorsatus* Redtenbacher, 1891	X		X
Subfamily Phaneropterinae Burmeister, 1838, with open tympana			
*Catoptropteryx ambigua* Huxley, 1970	X	X	X
*Catoptropteryx apicalis* Bolívar, 1893	X	X	X
*Catoptropteryx capreola* Karsch, 1896	X	X	X
*Catoptropteryx extensipes* Karsch, 1896	X	X	X
*Catoptropteryx guttatipes* Karsch, 1890	X	X	X
*Catoptropteryx nana* Huxley, 1970	X	X	X
*Catoptropteryx naevia* Huxley, 1970	X	X	X
*Catoptropteryx neutralipennis* Karsch, 1896	X		
*Catoptropteryx occidentalis* Huxley, 1970	X	X	
*Catoptropteryx punctulata* (Karsch, 1890)	X	X	X
*Corycomima camerata* (Karsch, 1889)	X	X	X
*Dannfeltia nana* Sjöstedt, 1902	X	X	
*Diogena fausta* (Burmeister, 1838)	X		X
*Ducetia loosi* Griffini, 1908	X		
*Eurycoplangiodes sanghaensis* Massa, 2020	X		
*Eurycorypha canaliculata* Karsch, 1890	X	X	X
*Eurycorypha montana* Sjöstedt, 1902		X	
*Eurycorypha ndokiensis* Massa, 2016	X		
*Eurycorypha ornatipes* Karsch, 1890	X	X	X
*Eurycorypha spinulosa* Karsch, 1889	X		
*Eurycorypha* sp. 1	X		X
*Eurycorypha* sp. 2	X		X
*Eurycorypha* sp. 3		X	X
*Eurycorypha* sp. 4	X		
*Eurycorypha* sp. 5		X	
*Eurycorypha* sp. 6		X	
*Gelotopoia bicolor* Brunner von Wattenwyl, 1891	X	X	X
*Griffinipteryx mukonja* (Griffini, 1908)		X	
*Monteiroa nigricauda* Ragge, 1980	X	X	X
*Oxygonatium huxleyi* Ragge, 1980	X		X
*Paraeulioptera emitflesti* Massa, 2020 *	X	X	
*Paraeurycorypha ocellata* Massa et Annoyer, 2020 *	X		
*Phaneroptera maculosa* Ragge, 1956	X		X
*Phaneroptera sparsa* Stål, 1857	X	X	X
*Phanreticula fenestrata* n. gen. n. sp. *		X	
*Plangia astylata* Massa, 2021 *	X	X	
*Plangia deminuta* Griffini, 1908	X	X	X
*Plangia nebulosa* Karsch, 1890	X	X	
*Pleothrix conradti* (Bolívar, 1906)		X	
*Pseudoeurycorypha civilettorum* Massa, 2023	X	X	X
*Pseudoplangia laminifera* (Karsch, 1896)	X	X	X
Subfamily Phaneropterinae Burmeister, 1838, with closed tympana			
*Arantia* (*Arantia*) *gretae* Massa, 2020 *	X		
*Arantia* (*Arantia*) *manca* Bolívar, 1906	X		X
*Arantia* (*Arantia*) *marginata* Massa, 2021	X	X	X
*Arantia* (*Arantia*) *quinquemaculata Hemp & Massa*, 2017	X	X	X
*Arantia* (*Arantia*) *simplicinervis* Karsch, 1889	X	X	X
*Arantia* (*Euarantia*) *congensis* Griffini, 1908	X	X	X
*Arantia* (*Euarantia*) *excelsior* Karsch, 1889	X	X	X
*Arantia* (*Euarantia*) *griffinii Hemp & Massa*, 2017 *	X		
*Arantia* (*Euarantia*) *incerata* Karsch, 1893		X	
*Arantia* (*Euarantia*) *marmorata* Karsch, 1889	X		X
*Arantia* (*Euarantia*) *melanota* Sjöstedt, 1902	X		X
*Arantia* (*Euarantia*) *rectifolia* Brunner von Wattenwyl, 1878	X	X	X
*Arantia* (*Euarantia*) *regina* Karsch, 1889	X	X	X
*Arantia* (*Euarantia*) *retinervis* Karsch, 1889	X	X	X
*Arantia* (*Euarantia*) *scurra* Karsch, 1896	X		
*Arantia* (*Euarantia*) *syssamagalei* Massa et Annoyer, 2020	X	X	X
*Arantia* (*Goetia*) *dimidiata* (Bolívar, 1906)	X		X
*Arantia* (*Goetia*) *galbana* (Karsch, 1891)	X		
*Arantia* (*Goetia*) *purpurea* (Massa, 2013) *	X		
*Azamia biplagiata* Bolívar, 1906	X	X	X
*Bongeia puncticollis* Sjöstedt, 1902	X		X
*Brycoptera lobata* Ragge, 1981		X	
*Buettneria maculiceps* Karsch, 1889		X	X
*Cestromoecha tenuipes* (Karsch, 1890)	X	X	X
*Cestromoecha longicerca* (Massa, 2013)	X	X	X
*Dapanera brevistylata* Massa, 2020	X	X	X
*Dapanera genuteres* Karsch, 1889	X	X	X
*Dapanera irregularis* Karsch, 1890	X	X	X
*Dapanera occulta* Massa, 2015 *	X	X	X
*Drepanophyllum marmoratum* Karsch, 1890	X		X
*Enochletica ostentatrix* Karsch, 1896	X	X	X
*Itokiia sylvarum* Sjöstedt, 1902	X		
*Leiodontocercus condylus* Ragge, 1962	X	X	X
*Leiodontocercus spinicercatus* Massa, 2020 *	X		
*Leiodontocercus vicentae* Massa, 2023 *			X
*Leiodontocercus vicii* Massa, 2020 *	X	X	
*Morgenia angustipinnata* Massa, 2018	X	X	X
*Morgenia hamuligera* Karsch, 1890	X	X	X
*Morgenia melica* Karsch, 1893	X	X	X
*Morgenia modulata* Karsch, 1896	X	X	X
*Morgenia plurimaculata* Massa et Moulin, 2018	X	X	X
*Morgenia rubricornis* Sjöstedt, 1913	X	X	X
*Morgenia spathulifera* Griffini, 1908	X		
*Myllocentrum stigmosum* (Karsch, 1896)	X		X
*Myllocentrum raggei* Massa, 2013 *	X		X
*Paraporeuomena signata* Massa, 2018 *	X		
*Phlaurocentrum armatum* n. sp. *		X	
*Phlaurocentrum dentatum* Massa, 2023 *		X	X
*Phlaurocentrum elegans* Massa, 2013 *	X		X
*Phlaurocentrum latevittatum* Karsch, 1889	X	X	X
*Phlaurocentrum lobatum* Ragge, 1962 (=*P. morettoi* Massa, 2013)	X		
*Phlaurocentrum maculatum* Ragge, 1962	X		X
*Phlaurocentrum mecopodoides* Karsch, 1891	X	X	
*Phlaurocentrum paratuberosum* Massa, 2013 *	X	X	X
*Phlaurocentrum tuberosum* Ragge, 1962	X	X	X
*Plangiola herbacea* Bolívar, 1906	X		
*Plangiopsis adeps* Karsch, 1896	X	X	X
*Plangiopsis nouabalensis* Massa, 2023 *		X	X
*Plangiopsis foraminata* Karsch, 1891	X	X	X
*Plangiopsis semiconchata* Karsch, 1889	X	X	X
*Poreuomena crassipes* Karsch, 1890	X		X
*Poreuomena derozierae* Massa, 2023 *			X
*Poreuomena forcipata* Sjöstedt, 1902	X		
*Poreuomena huxleyi* Massa, 2013	X		
*Poreuomena magnicerca* (Massa, 2013) *	X	X	X
*Poreuomena sanghensis* Massa, 2013	X	X	X
*Preussia lobatipes* Karsch, 1890	X	X	X
*Stenamblyphyllum dilutum* Karsch, 1896	X		
*Tetraconcha annoyeri* Massa, 2017 *	X	X	X
*Tetraconcha danielae* n. sp. *		X	
*Tetraconcha fijalkowskii* Massa, 2017 *	X	X	X
*Tetraconcha fuscopunctata* n. sp. *		X	
*Tetraconcha laszloi* Massa, 2023 *			X
*Tetraconcha loubesi* Massa, 2017	X	X	X
*Tetraconcha maculosa* Massa, 2023 *			X
*Tetraconcha morettoi* Massa, 2017	X		X
*Tetraconcha ndokiensis* Massa, 2017	X	X	X
*Tetraconcha omonomai* Massa, 2017 *	X	X	X
*Tetraconcha perezi* Massa, 2017 *	X		
*Tetraconcha smaragdina* Brunner von Wattenwyl, 1878	X		
*Tetraconcha stichyrata* Karsch, 1890	X	X	X
*Tylopsis irregularis* Karsch, 1893		X	
*Tropidophrys amydra* Karsch, 1896	X	X	
*Vossia obesa* Brunner von Wattenwyl, 1891	X	X	X
*Weissenbornia p. praestantissima* Karsch, 1888	X	X	X
*Zeuneria biramosa* Sjöstedt, 1929		X	X
*Zeuneria longicercus* Sjöstedt, 1929	X	X	X
*Zeuneria melanopeza* Karsch, 1889	X	X	
Genus and species unknown		X	
Subfamily Mecopodinae Walker, 1871			
*Acridoxena hewaniana* Smith, 1865	X		X
*Afromecopoda frontalis* (Walker, 1871)	X		X
*Anoedopoda erosa* Karsch, 1891	X		
*Anoedopoda lamellata* (Linnaeus, 1758)		X	
*Apteroscirtus denudatus* Karsch, 1891	X		X
*Corycoides karschi* (Krauss, 1890)	X	X	X
*Macroscirtus kanguroo* Pictet, 1888 (Reported erroneously as *Euthypoda acutipennis* (Karsch, 1886))	X	X	
*Leproscirtus granulosus* (Karsch, 1886)		X	
Subfamily Hetrodinae Brunner von Wattenwyl, 1878			
*Cosmoderus erinaceus* (Fairmaire, 1858)	X		
*Cosmoderus femoralis* (Sjöstedt, 1902)		X	X
Total No. of taxa (n = 181)	140	108	114

X means presence of species.

## Data Availability

The raw data supporting the conclusions of this article will be made available by the authors without undue reservation.

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
