# Peer review of "Diversity of Leaf Katydids of Odzala-Kokoua National Park, Republic of Congo, Central Tropical Africa (Insecta: Orthoptera: Tettigoniidae)"

_insects, 2025, doi:10.3390/insects16030241_

Round 1
Reviewer 1 Report
Comments and Suggestions for Authors
The ms of Bruno Massa presents wonderful data on the diversity of tettigoniids in an unexplored African rainforest. There are, however, a few serious problems. The most important one must be solved by the editors of the journal (could possibly done also by the author; unusual solution). Since Insects is an online-only journal, papers containing important (new taxa!) taxonomic information must be registered at zoobank (see below) in addition to registrations of the new taxa (see papers e.g,.in Zootaxa, Zookeys..). An example for taxa registration is found in Insects 2024, 15, 787, 11 of 16 , line 4 - paper registration missing! Other examples below; https://zoobank.org/References/19E46430-5E96-42AF-8E7E-6053D7DADCE1https://zoobank.org/DB7A1C07-354F-475F-BF2A-B6BC818E0800
Typically this is done by the journal -see example below
"All papers with taxonomic information are registered in the Official Register of Zoological Nomenclature (ZooBank) and issues of the journal are archived in eLibrary and Zenodo (according to the amendment of Articles 8, 9, 10, 21 and 78 of the International Code of Zoological Nomenclature, 2012: https://www.iczn.org/the-code/electronic-publication-made-available-with-amendment-to-the-code/). Thus, all nomenclature acts published in xxx become valid immediately after the online publication. "
Minor problems:
The Introduction is dealing with Guinean forests (lines 42ff, eastwards to Sanaga River in Cameroon ), but the paper is focusing on forests from Congo (as part the Northwestern Congolian lowland rainforest ecoregion). Please adjust.
During mating male bush-crickets produce and tranfer a spermatophore to the _female_ where it is visible for several to many hours. It cannot be seen in males except in very rare cases in problems during mating!
Please separate the sections dealing with a species by empty lines (editorial decision?)
Please change sub-genital (occasionally in ms )-> subgenital
Please check capitulizatiion in
23. Lieberman, B.S. Adaptive Radiations in the
Author Response
Comment
The ms of Bruno Massa presents wonderful data on the diversity of tettigoniids in an unexplored African rainforest. There are, however, a few serious problems. The most important one must be solved by the editors of the journal (could possibly done also by the author; unusual solution). Since Insects is an online-only journal, papers containing important (new taxa!) taxonomic information must be registered at zoobank (see below) in addition to registrations of the new taxa (see papers e.g,.in Zootaxa, Zookeys..). An example for taxa registration is found in Insects 2024, 15, 787, 11 of 16 , line 4 - paper registration missing! Other examples below; https://zoobank.org/References/19E46430-5E96-42AF-8E7E-6053D7DADCE1https://zoobank.org/DB7A1C07-354F-475F-BF2A-B6BC818E0800
Typically this is done by the journal -see example below
"All papers with taxonomic information are registered in the Official Register of Zoological Nomenclature (ZooBank) and issues of the journal are archived in eLibrary and Zenodo (according to the amendment of Articles 8, 9, 10, 21 and 78 of the International Code of Zoological Nomenclature, 2012: https://www.iczn.org/the-code/electronic-publication-made-available-with-amendment-to-the-code/). Thus, all nomenclature acts published in xxx become valid immediately after the online publication. "
Reply
The author is aware of the need to register new taxa on ZooBank, but intended to do so after acceptance by the journal. The new version also contains the registered codes of the new taxa on ZooBank.
Comment
The Introduction is dealing with Guinean forests (lines 42ff, eastwards to Sanaga River in Cameroon ), but the paper is focusing on forests from Congo (as part the Northwestern Congolian lowland rainforest ecoregion). Please adjust.
Reply
Introduction ahs been modified
Comment
During mating male bush-crickets produce and tranfer a spermatophore to the _female_ where it is visible for several to many hours. It cannot be seen in males except in very rare cases in problems during mating!
Reply
Since most of the specimens were collected at light, most likely some of them were caught during mating and the spermatophore was stuck in the male genital apparatus. The spermatophores of some species, demonstrating this fact, were preserved in envelopes for possible future studies.
Comment
Please separate the sections dealing with a species by empty lines (editorial decision?)
Reply
Done
Comment
Please change sub-genital (occasionally in ms )-> subgenital
Reply
Done
Comment
Please check capitulizatiion in
23. Lieberman, B.S. Adaptive Radiations in the
Reply
I reported the exact title of the original paper (Adaptive Radiations in the Context of Macroevolutionary Theory: A Paleontological Perspective)
Reviewer 2 Report
Comments and Suggestions for Authors
Very interesting, especially the conclusion chapter on importance of Nature protection.
I suggest to add an empty line between one species chapter and the following.
Author Response
Comment: I suggest to add an empty line between one species chapter and the following.
Reply: done
Reviewer 3 Report
Comments and Suggestions for Authors
Review "Diversity of Leaf Katydids insects-3491285"
An interesting review of the Tettigoniidae fauna of central tropical Africa. It consists of two main parts: (1) a description of the current situation of land and habitats as experienced by the authors during three expeditions to three central African National Parks, well illustrated by photos of the landscape and a report of the species found during the authors' visits. The second part deals with faunistics and taxonomy and gives an overview on the former and actual Tettigoniidae species living in that area. It is largely based on formerly described and on undescribed species found in museum collections. The latter are described as new. The manuscript thus gives an actualised overview of the Tettigoniidae species found in that area. Six species are described as new. Species names, Localities, collectors, type status, number of specimens and depositories are listed. The study is a good base for future studies and will allow to detect eventual changes in faunal composition due to change in land use or climate.
The manuscript is written in good and understandable English. A few minor typos are summarised in a separate list.
Proposal: accept as is after correction of typos and other minor points.

Author Response
I followed all the suggestions kindly listed by the referee on the pdf
Reviewer 4 Report
Comments and Suggestions for Authors
This paper deals with the study of Tettigoniidae found in Odzala-Kokoua National Park, Republic of Congo. The results show the high diversity of species that may be found in the tropical Africa. It's a good job. I do not see any problems with the paper.
Author Response
Thank you very much
Round 2
Reviewer 1 Report
Comments and Suggestions for Authors
OK
Perhaps to improve
Table1 and 2 beginning on new pages would be better to read